# Integrating spatial and single-cell transcriptomics data using deep generative models with SpatialScope

Xiaomeng Wan[1,12], Jiashun Xiao[2,12], Sindy Sing Ting Tam [3], Mingxuan Cai[4], Ryohichi Sugimura [5], Yang Wang[1,6,7], Xiang Wan[2], Zhixiang Lin [8] ✉, Angela Ruohao Wu [3,9,10,11] ✉ & Can Yang [1,6,7] ✉

The rapid emergence of spatial transcriptomics (ST) technologies is revolutionizing our understanding of tissue spatial architecture and biology. Although current ST methods, whether based on next-generation sequencing (seq-based approaches) or fluorescence in situ hybridization (image-based approaches), offer valuable insights, they face limitations either in cellular resolution or transcriptome-wide profiling. To address these limitations, we present SpatialScope, a unified approach integrating scRNA-seq reference data and ST data using deep generative models. With innovation in model and algorithm designs, SpatialScope not only enhances seq-based ST data to achieve single-cell resolution, but also accurately infers transcriptome-wide expression levels for image-based ST data. We demonstrate SpatialScope's utility through simulation studies and real data analysis from both seq-based and image-based ST approaches. SpatialScope provides spatial characterization of tissue structures at transcriptome-wide single-cell resolution, facilitating downstream analysis, including detecting cellular communication through ligand-receptor interactions, localizing cellular subtypes, and identifying spatially differentially expressed genes.

Single-cell RNA sequencing (scRNA-seq) characterizes the whole transcriptome of individual cells within a given organ, providing remarkable opportunities for broad and deep biological investigations of diverse cellular behaviors[1–3]. However, scRNA-seq does not capture the spatial distribution of cells due to samples having to undergo tissue dissociation[4]. As spatial information is so critical to understanding communication between cells, many scientific questions related to cellular communication cannot be fully addressed by scRNA-seq alone[5].

Current ST approaches are predominantly based on either next-generation sequencing (seq-based) or fluorescence in situ hybridization (image-based). Seq-based approaches, such as 10x Visium[6], Slide-

[1]Department of Mathematics, The Hong Kong University of Science and Technology, Hong Kong SAR, China. [2]Shenzhen Research Institute of Big Data, Shenzhen 518172, China. [3]Division of Life Science, The Hong Kong University of Science and Technology, Clear Water Bay, Kowloon, Hong Kong SAR, China. [4]Department of Biostatistics, City University of Hong Kong, Hong Kong SAR, China. [5]Li Ka Shing Faculty of Medicine, School of Biomedical Sciences, University of Hong Kong, Hong Kong SAR, China. [6]Guangdong-Hong Kong-Macao Joint Laboratory for Data-Driven Fluid Mechanics and Engineering Applications, The Hong Kong University of Science and Technology, Hong Kong SAR, China. [7]Big Data Bio-Intelligence Lab, The Hong Kong University of Science and Technology, Hong Kong SAR, China. [8]Department of Statistics, The Chinese University of Hong Kong, Hong Kong SAR, China. [9]Department of Chemical and Biological Engineering, The Hong Kong University of Science and Technology, Hong Kong SAR, China. [10]Center for Aging Science, The Hong Kong University of Science and Technology, Hong Kong SAR, China. [11]State Key Laboratory of Molecular Neuroscience, The Hong Kong University of Science and Technology, Hong Kong SAR, China. [12]These authors contributed equally: Xiaomeng Wan and Jiashun Xiao. ✉e-mail: zhixianglin@cuhk.edu.hk; angelawu@ust.hk; macyang@ust.hk

seq[7] and Stereo-seq[8], can detect transcriptome-wide gene expression within spatial spots. Among them, the Visium technology has gained considerable maturity over the years, becoming a well-established commercially available method in the field of ST. According to the database collected by the museum of spatial transcriptomic project[9], more than half of studies in the past year still utilized the Visium technology to quantify gene expression in space, accumulating a substantial amount of data[9]. However, considering the larger spot size of 55 $\mu m$, a Visium spot often contains multiple cells, which limits its usage in resolving detailed tissue structure and in characterizing cellular communications (e.g., identifying ligand-receptor interactions[10]).

Image-based approaches such as seqFISH[11] and MERFISH[12] are designed to measure thousands of genes with single-cell resolution, but they often lack whole-transcriptome coverage, resulting in only a few hundred genes in real applications. Users of these image-based methods need to have well-defined biological hypotheses to design an appropriate and useful gene panel, and it is unlikely to generate incidental discoveries in this scenario.

Ideally, the integration of single-cell and ST data should allow us to characterize the spatial distribution of the whole transcriptome at single-cell resolution, by combining their complementary information. However, existing integration methods are far from satisfactory in real data analysis[13]. There are now several cell-type deconvolution methods for ST data, including RCTD[14], Cell2location[15], CARD[16] and spatialDWLS[17]. When these deconvolution methods are applied to seq-based ST data, they only estimate the proportions of cell types in each spatial spot but cannot achieve single-cell resolution. Therefore, the aforementioned limitations of not having single-cell resolution remain unresolved. For image-based ST data, methods developed to infer unmeasured gene expressions, such as Tangram[18], gimVI[19] and SpaGE[20] are not sufficiently accurate, especially when ST expression data are sparse[13]. Therefore, there remains a need for accurate statistical and computational methods for integrating single-cell and ST datasets[4].

Herein we introduce SpatialScope, a unified approach to integrating scRNA-seq reference data and ST data generated from various experimental platforms, applicable to both seq-based ST data (e.g., 10x Visium and Slide-seq) and image-based data (e.g., MERFISH). By leveraging deep generative models, SpatialScope can resolve the spot-level data composed of multiple cells to single-cell resolution when it is applied to seq-based ST data. There are two key features of SpatialScope. First, it can greatly improve cell type identification by exploiting spatial information of cells through Potts model and properly correcting for batch effect between ST and scRNA-seq reference data. Second, unlike alignment-based methods such as Tangram[18] and CytoSPACE[21] that assign existing cells from scRNA-seq data to spatial spots, SpatialScope can generate the gene expressions of pseudo-cells using the learned deep generative model to match the observed spot-level gene expression in space. Consequently, SpatialScope can decompose the observed gene expression at each spot into the single-cell level gene expression accurately. In addition, for image-based ST data, SpatialScope can learn the distribution of gene expressions from the scRNA-seq data and then infer transcriptome-wide expression of the unmeasured genes in the sample, conditioned on the observed tens to hundreds of genes in that sample. With the above features, SpatialScope allows more in-depth and informative downstream analyses at single-cell resolution. Using ST data generated from various experimental platforms, such as 10x Visium, Slide-seq and MERFISH data, we show that the results of SpatialScope enable spatially resolved cellular communications mediated by ligand-receptor interactions and spatially differentially expressed genes expression, highlighting SpatialScope's utility in elucidating underlying biological processes. By applying SpatialScope to human heart data, ligand and receptor pairs that are essential in vascular proliferation and differentiation are detected using higher resolution ST data generated by SpatialScope. Some meaningful genes absent in MERFISH data are detected as DE

genes through the imputation of SpatialScope. Very recently, Spatial-Scope has been applied to enhance the resolution of ST data generated from human embryonic hematopoietic organoids, producing single-cell resolution ST data which was then used to detect spatially resolved cell-cell interactions and co-localization of different cell types[22]. This single-cell resolution decomposition of the original data has allowed us to identify additional biological findings that were not possible at spot-level.

## Results

### Overview of the SpatialScope method

By leveraging the deep generative model, SpatialScope enables the characterization of spatial patterns of the whole transcriptome at single-cell resolution for ST data generated from various experimental platforms. We begin our formulation with gene expression decomposition of seq-based ST data from the spot level to the single-cell level. Let $\mathbf{y} \in \mathbb{R}^G$ be the expression levels of $G$ genes (after batch effect corrections) at a spot in seq-based data. While it is important to note that each spot in ST data may contain multiple single cells with aggregated expression levels, for the sake of illustration, we consider a spot containing two cells (although our method is applicable to spots with multiple cells). To elucidate our key concept, let us assume that we already know that the spot-level gene expression $\mathbf{y}$ comes from two cells of different types, $\mathbf{y} = \mathbf{x}_1 + \mathbf{x}_2 + \boldsymbol{\varepsilon}$, where $\mathbf{x}_1$ and $\mathbf{x}_2$ are the true gene expression levels of cells 1 and 2 whose cell types are denoted as $k_1$ and $k_2$, respectively, and the independent random noise $\boldsymbol{\varepsilon}$ is assumed to be $\mathcal{N}(0, \sigma_\varepsilon^2 \boldsymbol{I})$ for convenience. We aim to decompose $\mathbf{y}$ into $\mathbf{x}_1$ and $\mathbf{x}_2$, and thus obtain the single-cell resolution gene expression at the given spot. To achieve this, we use a deep generative model[23–25] to learn the expression distributions of cell types $k_1$ and $k_2$ from the scRNA-seq reference data, denoted as $p(\mathbf{x}_1|k_1)$ and $p(\mathbf{x}_2|k_2)$. Based on Langevin dynamics[24, 26], we can obtain the decomposition by sampling $\mathbf{X} = [\mathbf{x}_1; \mathbf{x}_2]$ from the posterior distribution $p(\mathbf{X}|\mathbf{y}, k_1, k_2)$,

$$\mathbf{X}^{(t+1)} = \mathbf{X}^{(t)} + \eta \nabla_{\mathbf{X}} \log p\left(\mathbf{X}^{(t)}|\mathbf{y}, k_1, k_2\right) + \sqrt{2\eta} \boldsymbol{\varepsilon}^{(t)}, \quad (1)$$

Where $\boldsymbol{\varepsilon}^{(t)} \sim \mathcal{N}(0, I)$ and $\eta > 0$ is the step size, $t = 1, \dots, \infty$. By Bayes rule, we have $\log p(\mathbf{X}^{(t)}|\mathbf{y}, k_1, k_2) = \log p(\mathbf{y}|\mathbf{X}^{(t)}, k_1, k_2) + \log p(\mathbf{x}_1^{(t)}|k_1) + \log p(\mathbf{x}_2^{(t)}|k_2) - \log p(\mathbf{y}|k_1, k_2)$

Noting that $\nabla_X \log p(\mathbf{y}|k_1, k_2) = 0$, this makes it easy to obtain posterior samples from the Langevin dynamics as

$$X^{(t+1)} = X^{(t)} + \eta \left( \nabla_{\mathbf{X}} \log p\left(\mathbf{y}|\mathbf{X}^{(t)}, k_1, k_2\right) + \begin{bmatrix} \nabla_{\mathbf{x}_1} \log p\left(\mathbf{x}_1^{(t)}|k_1\right) \\ \nabla_{\mathbf{x}_2} \log p\left(\mathbf{x}_2^{(t)}|k_2\right) \end{bmatrix} \right) + \sqrt{2\eta} \boldsymbol{\varepsilon}^{(t)}, \quad (2)$$

where $p(\mathbf{y}|\mathbf{X}^{(t)}, k_1, k_2) = \mathcal{N}(\mathbf{y}|\mathbf{x}_1^{(t)} + \mathbf{x}_2^{(t)}, \sigma_\varepsilon^2 \boldsymbol{I})$; $\nabla_{\mathbf{x}_1} \log p(\mathbf{x}_1^{(t)}|k_1)$ and $\nabla_{\mathbf{x}_2} \log p(\mathbf{x}_2^{(t)}|k_2)$ are known as the score function which can be learned from the scRNA-seq reference data. The samples from the posterior distribution $p(\mathbf{X}^{(t)}|\mathbf{y}, k_1, k_2)$ recover gene expression levels of the two cells, achieving single-cell resolution.

To implement the key idea formulated above, SpatialScope comprises three steps of real data analysis (Fig. 1): (i) Nucleus segmentation; (ii) cell type identification; and (iii) gene expression decomposition with a score-based generative model. Specifically, we first perform nucleus segmentation on the hematoxylin and eosin (H&E)-stained histological image to count the number of cells at each spot. Second, for cell type identification (i.e., $k_1$, and $k_2$) at each spot, we develop a fast and accurate method by integrating scRNA-seq and ST data. Third, we learn the conditional score generative model (i.e., $\nabla_{\mathbf{x}_1} \log p(\mathbf{x}_1^{(t)}|k_1)$ and $\nabla_{\mathbf{x}_2} \log p(\mathbf{x}_2^{(t)}|k_2)$) in a coherent neural network to approximate the expression distribution of different cell types from scRNA-seq data (Supplementary Fig. 20), and then use the learned model to decompose gene expression from the spot level to the single-

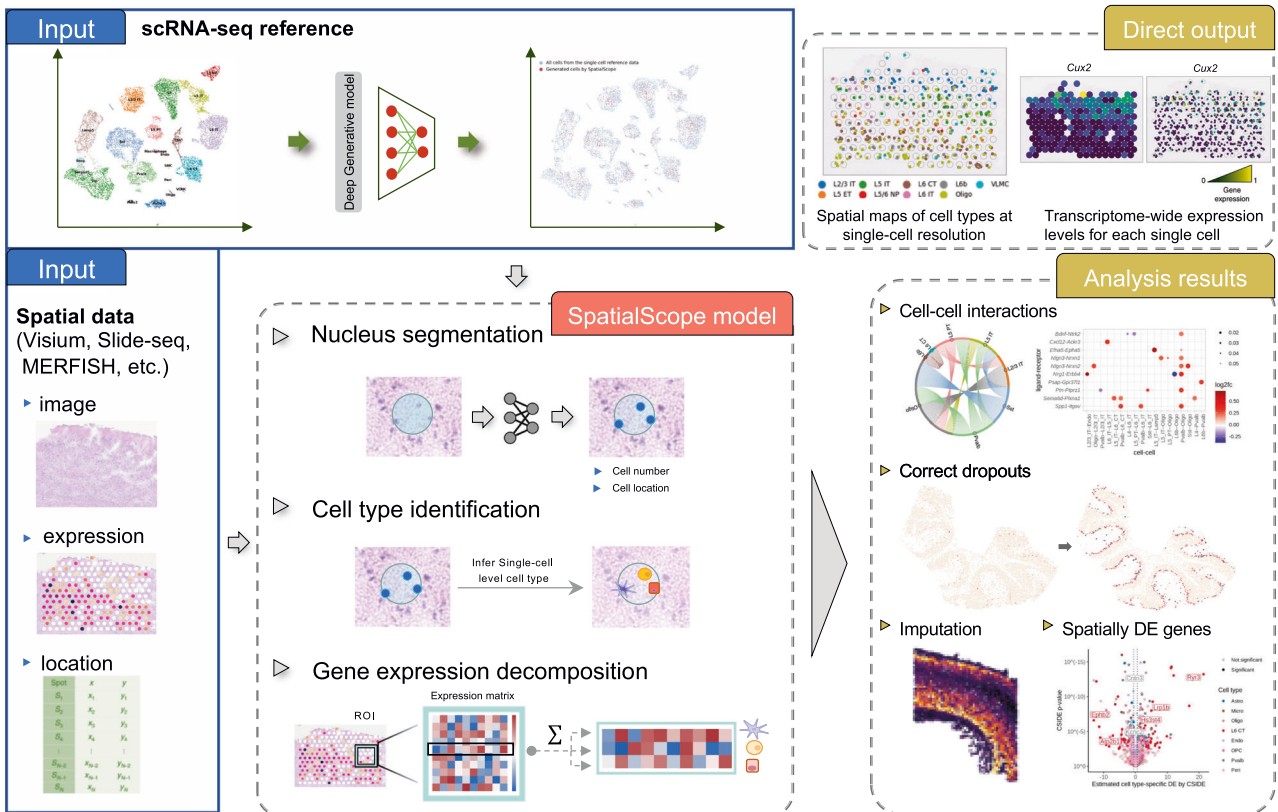

**Fig. 1 | Overview of SpatialScope.** SpatialScope is designed to infer spatially resolved single-cell transcriptomes by harnessing the capabilities of deep generative models to learn distributions from scRNA-seq reference data. The workflow of SpatialScope begins by quantifying the number of cells within each spot in low-resolution ST data, such as Visium. Subsequently, it identifies the cell type labels for individual cells within the spot. Finally, by conditioning on the inferred cell type labels, SpatialScope performs gene expression decomposition, transforming the spot-level gene expression profile into single-cell resolution. This decomposition enables more comprehensive and informative downstream analyses at the single-cell level.

cell level, as outlined above. Based on the same modeling principle, we generalize SpatialScope to infer the unmeasured gene expression for image-based ST data, conditional on the observed gene expression levels. We introduce the details of SpatialScope in the method section.

## A benchmarking study on cell type identification and gene expression decomposition

To evaluate the performance of SpatialScope in the cell type identification and gene expression decomposition steps, we conducted a benchmarking study using six simulated datasets (Supplementary Fig. 4). We compared SpatialScope with twelve existing methods, including Tangram[18], CytoSPACE[21], RCTD[14], SpatialDWLS[17], Cell2location[15], CARD[16], SpaOTsc[27], novoSpaRc[28], DestVI[29], STRIDE[30], SPOTlight[31], and DSTG[32]. Additionally, we included an alternative method called StarDist+RCTD (Supplementary Note section 2.9.3, Supplementary Fig. 43 and 44), which discretizes the results of RCTD and assigns the average expression of cell types to individual cells, as a baseline for comparison. Following the approach described in a previous benchmarking study[13], we generated simulation datasets by gridding and aggregating cells on uniform grids to create simulated spots (Fig. 2a, Supplementary Fig. 1). More details of simulated datasets in the benchmarking study can be found in Supplementary Note section 2.9.1.

To evaluate the cell type identification performance, we conducted two analyses. In Case (a), we compared the performance of SpatialScope, Tangram, CytoSPACE, and StarDist+RCTD, which are capable of inferring cell type labels at the single-cell level. We assessed their cell type identification accuracy at the single-cell resolution by calculating the misclassification error rate, which represents the proportion of cells with misclassified cell type labels. In Case (b), we considered methods that provide cell type proportions at the spot level. For these methods, we aggregated the results of SpatialScope from the single-cell level to the spot level and compared them to other methods using the Pearson correlation coefficient (PCC) and root-mean-square error (RMSE) metrics. These metrics quantified the correlation and deviation between the estimated cell type proportions obtained by each method and the ground truth values.

In Case (a), we applied SpatialScope, Tangram, CytoSPACE, and StarDist+RCTD to four single-slice datasets (Fig. 2a, b). SpatialScope consistently outperformed all other cell type identification methods, exhibiting a 50.3%, 20.6%, and 6.3% reduction in error rate compared to Tangram, CytoSPACE, and StarDist+RCTD, respectively, across all four single-slice datasets (Fig. 2c). The same trend was observed when these methods were applied to the two multiple-slice datasets (Supplementary Fig. 14a). SpatialScope remained the most accurate method for inferring cell type labels at the single-cell level, achieving a 22.9–50.0% reduction in error rate for Dataset 5 and a 4.6–48.3% reduction in error rate for Dataset 6. In Case (b), we compared SpatialScope to existing deconvolution methods that provide cell type proportions only at the spot level, using metrics such as PCC and RMSE (Fig. 2d, Supplementary Fig. 14b and Figs. 5–13). SpatialScope consistently outperformed or achieved comparable performance to other methods across all datasets in terms of PCC. It demonstrated substantial improvements in PCC, ranging from 9.4% to 157.5%, compared to Tangram, CytoSPACE, SpaOTsc, novoSpaRc, STRIDE, and SPOTlight across the six datasets. Additionally, SpatialScope exhibited a maximum improvement of 51.4% over RCTD, SpatialDWLS, Cell2location, CARD, DestVI, and DSTG for the same datasets. In terms of RMSE,

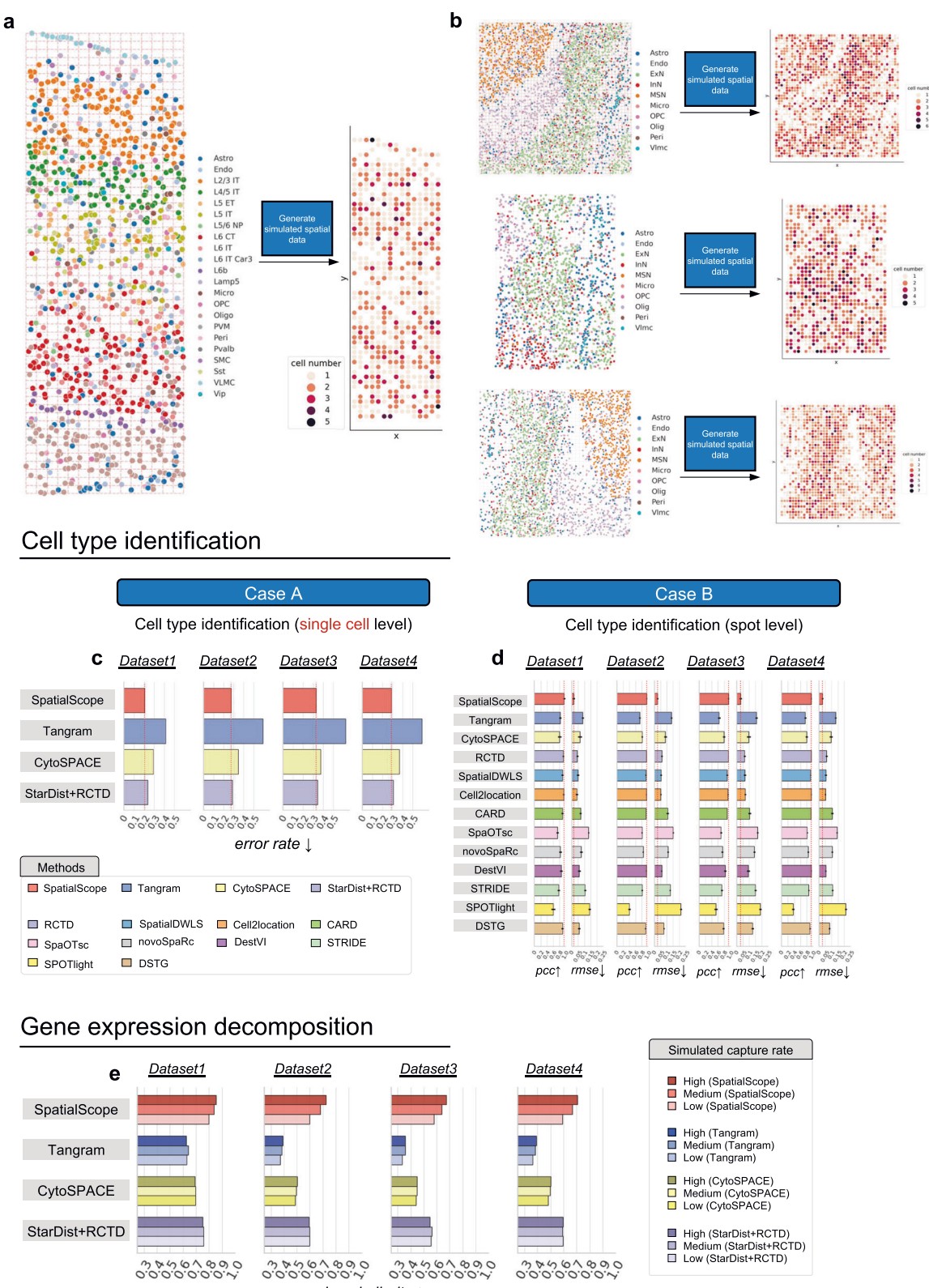

**Fig. 2 | A benchmarking study on cell type identification and gene expression decomposition at single-cell resolution. a** Dataset 1 in the benchmarking datasets. A spatial scatter plot displays the cell types at the single-cell resolution. Red dashed lines indicate the grids used for aggregating cells into spots. After aggregation, a scatter plot shows the simulated spots and the number of cells per spot. **b** Datasets 2-4 in the benchmarking datasets. **c** Bar plots showing the error rate of each method in inferring cell type labels at the single-cell level for the four single-slice benchmarking datasets (Dataset 1-4). **d** Bar plots showing the Pearson correlation coefficient (PCC) and root-mean-square error (RMSE) of each method in inferring cell type proportions at the spot level for the four single-slice benchmarking datasets (Dataset 1-4). Data are presented as mean values ± 95% confidence intervals; $n$ = 599, 1753, 901, 1359 is the number of spots for Dataset1, Dataset2, Dataset3 and Dataset4 respectively. **e** Bar plots showing the cosine similarity of each method in inferring transcriptome-wide expression levels for each single cell in the four single-slice benchmarking datasets (Dataset 1-4) at different simulated capture rates. Source data are provided as a Source Data file.

SpatialScope consistently achieved the highest deconvolution accuracy across all datasets, with improvements ranging from 25.3% to 89.7% compared to all other methods. The main reason for these improvements lies in the smoothness constraint incorporated in the cell type identification step of SpatialScope. We provide more evidence to demonstrate the role of smoothness constraint in Supplementary Note section 2.9.4.

To assess the robustness of SpatialScope in cell type identification and gene expression decomposition, we conducted evaluations using simulated spots with different grid sizes and total UMI counts per spot (Supplementary Figs. 2 and 3, Supplementary Note section 2.9.11). Specifically, we considered the configuration with a grid size of 34 × 30 μm and a total UMI count per spot of 260, and the results are presented in Supplementary Fig. 15a. SpatialScope consistently produced more accurate results compared to Tangram and CytoSPACE. Tangram missed many cells due to its inability to utilize the actual number of nuclei in the histological image to determine the output cell number. CytoSPACE exhibited inaccuracies in assigning cell type labels, particularly in the L5 IT layer. This observation is further supported by the confusion matrix, which shows a lower on-diagonal correlation for Tangram and noise with inappropriate off-diagonal correlation for CytoSPACE (Supplementary Fig. 15b). Furthermore, when the reference dataset contained missing cell types, we demonstrated that SpatialScope exhibited the highest robustness among the compared methods by predicting the cells as the most transcriptionally similar cell type in the reference (Supplementary Note section 2.9.5, Supplementary Fig. 52). Additionally, SpatialScope demonstrated computational efficiency, with the construction of spatial maps of cell types at the single-cell resolution taking only 1.5 minutes, faster than most other methods (Supplementary Fig. 15c).

Next, we assessed the performance of SpatialScope in single-cell gene expression inference. The objective was to decompose mixed reads within each spot and generate gene expression profiles at the single-cell resolution, overcoming the limitation of low-resolution spatial transcriptomics (ST) data. To illustrate this, we present an example in Supplementary Fig. 16a, where SpatialScope decomposes spot-level gene expression profiles, which are a mixture of signals from two single cells, into cell-level gene expression profiles using Dataset 1. SpatialScope accurately recovers the gene expression of the two individual cells, with a mean cosine similarity as high as 0.90, measured between the estimated gene expression and the underlying truth for each single cell. In contrast, the inferred cells generated using Tangram (purple dots) and CytoSPACE (orange dots), both of which employ alignment-based methods, exhibit greater dissimilarity to the ground truth, with mean cosine similarities of 0.44 and 0.57, respectively. By accurately decomposing gene expression, SpatialScope recaptures the higher spatial resolution offered by the original MER-FISH data, which is lost in the simulated ST data (Supplementary Fig. 16b).

To systematically assess SpatialScope's ability to infer expression levels at the single-cell level, we conducted a benchmark study using four single-slice datasets. We compared SpatialScope with Tangram, CytoSPACE, and StarDist+RCTD, which are among the few methods capable of inferring expression levels at this resolution. In order to assess the robustness of these methods to variations in data quality, we manipulated the unique molecular identifier (UMI) counts by downsampling, simulating different capture rates of spatial transcriptomics (ST) data. To quantify the accuracy of gene expression decomposition, we computed the cosine similarity between the estimated gene expression and the ground truth for each individual cell.

Across all four datasets in the benchmarking study, SpatialScope consistently outperformed other methods in inferring transcripts at the single-cell level. It achieved significant improvements in terms of cosine similarity compared to Tangram, CytoSPACE, and StarDist+RCTD, with improvements of 64.6%, 32.1%, and 11.4% respectively,

across all settings and datasets (Fig. 2e). The superior performance of SpatialScope in gene expression decomposition can be attributed to its fundamental differences from other methods. Unlike alignment-based methods such as Tangram and CytoSPACE, which assign existing cells from scRNA-seq data to spatial spots, or methods like StarDist+RCTD that assign average gene expressions of cell types, SpatialScope has the unique ability to generate pseudo-cell gene expressions using its learned deep generative model. This generation process enables SpatialScope to better match the observed spot-level gene expression in space, resulting in more accurate results.

We also examined the gene expression accuracy of different methods at distinct simulated capture rate levels of the datasets. SpatialScope exhibits a consistent pattern where the accuracy increases with higher capture rates (Fig. 2e), indicating its ability to fully leverage data quality. This pattern is not observed or not evident in the results of other methods, suggesting that they are unable to fully leverage the information contained in the data.

In real-world scenarios, generating paired scRNA-seq data for each ST profiling experiment may not be feasible due to budget constraints or sample availability. To address this, we conducted simulations on Dataset 1 to evaluate the accuracy and robustness of different tools in generating single-cell gene expression decomposition from ST data using either paired scRNA-seq data or an independently generated scRNA-seq reference of the same tissue type. SpatialScope consistently achieved significantly higher accuracy regardless of whether paired or unpaired single-cell reference data was used (Supplementary Fig. 16c). However, the performance of alignment-based methods, Tangram and CytoSPACE, notably declined when the reference data was generated from the same tissue type but different biological samples. Furthermore, considering that the nucleus segmentation step may miss some cells with weak signals, we also evaluated the performance of SpatialScope and the compared methods when the estimated cell number in the spots did not match the ground truth cell number (see Supplementary Note section 2.9.6). We observed that SpatialScope demonstrated robustness in handling inconsistent cell numbers and was able to accurately identify the remaining ground truth cells with highly matched transcriptional profiles (Supplementary Figs. 53–55). By investigating the gene expression performance in scenarios where there is significant variation in the proportions of different cell types within single-cell reference data and imbalanced cell numbers within spots, we conducted further analysis to validate the robustness of SpatialScope in handling unbalanced cell types within single-cell reference data and uneven cell numbers within spots (Methods, Supplementary Note section 2.9.9, 2.9.10).

## SpatialScope enables the integration of multiple slices and interpretation of cell-cell interactions by leveraging single-cell resolution gene expression profiles

Recently, ST data with multiple parallel slices in tissue from one or more samples has become more widely available and is being generated at an accelerated pace. Effectively capitalizing on the information present in neighboring slices and integrating information from multiple slices is crucial for enhancing the performance of ST data analysis tools when applied to ST data with multiple slices. SpatialScope utilizes spatial information by encouraging neighboring cells to belong to the same cell type either within a single slice or across slices (Supplementary Fig. 15d). When ST data have multiple slices, leveraging spatial location information enables SpatialScope to integrate information from multiple slices.

We benchmarked its performance on two multiple-slice datasets (Supplementary Note section 2.9.1, Supplementary Fig. 4, Dataset 5-6) and compared it with Tangram, CytoSPACE, and StarDist+RCTD. We prepared two settings for applying the four methods to multiple-slice datasets. In setting (i), we applied the methods to each single slice one by one in the dataset (Supplementary Fig. 14, Single slice). In setting

(ii), we applied the methods to all slices at once in the dataset (Supplementary Fig. 14, Multiple slices), where all slices were aligned in the xy-axis and evenly spaced in the z-axis (Supplementary Fig. 4e, f). If a method has the ability to integrate the information across slices, the cell type identification accuracy should improve in setting (ii) compared to setting (i). This pattern was observed for SpatialScope, where the error rate decreased under setting (ii) compared to setting (i) in all multiple-slice datasets. However, the same pattern was not observed for the error rate of other methods. For example, the error rate of Tangram decreased under setting (ii) compared to setting (i) in Dataset 5 but increased in Dataset 6 (Supplementary Fig. 14a). The accuracy improvement of SpatialScope when applied to multiple-slice data

benefits from incorporating spatial information in the model design. When measuring PCC and RMSE at spot level, PCC and RMSE did not show significant improvement when integrating multiple slices compared to using only single-slice (Supplementary Fig. 14b). This pattern differs from the improvement observed when measuring the error rate. The reason behind this is that PCC and RMSE are measured at the spot level, which is a coarse resolution that cannot capture the improvement gained from borrowing information across slices. The improvement can only be observed at a higher resolution by measuring the error rate at the single-cell level.

To further illustrate how SpatialScope is applied to multi-slices data and improves the accuracy of cell type identification by

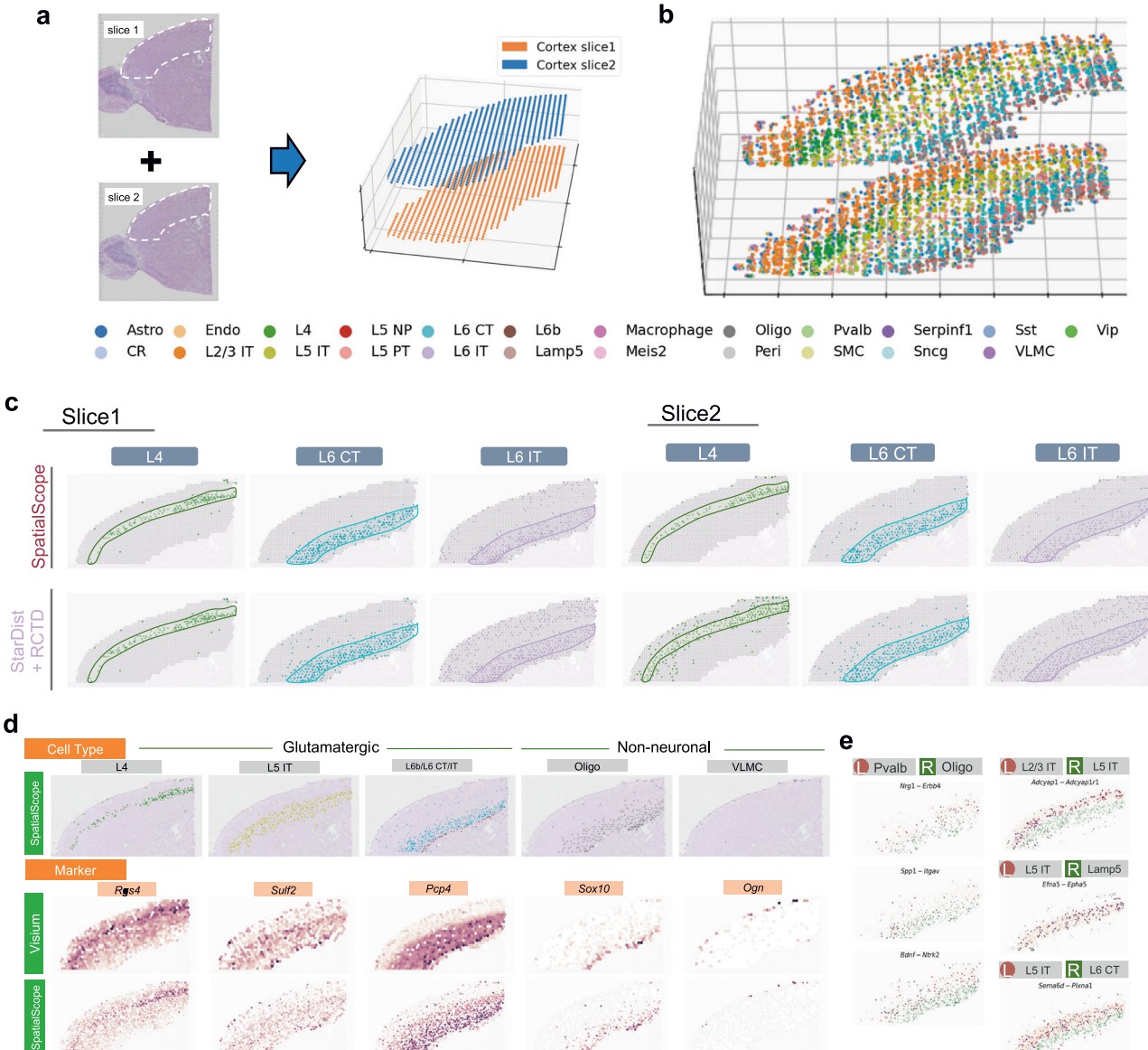

**Fig. 3 | SpatialScope enables the integration of multiple slices and interpretation of cell-cell interactions in mouse brain cortex data. a** H& E staining histology of two adjacent slices of mouse brain cortex from 10x Visium (left). White polygons indicate the region of interest. Alignment results using PASTE[34] of the two slices (right). **b** The SpatialScope identified cell type labels for the stacked 3D ST data constructed by two slices. **c** Comparison of cell type identification results (left: slice 1 of mouse brain cortex; right: slice 2 of mouse brain cortex) between SpatialScope using multiple-slice neighboring information (top) and StarDist +RCTD (bottom). Polygons indicate the region of the corresponding cell type layer, and the color represents cell types. **d** Top: spatial cell locations identified as

L4, L5 IT, L6b/L6 CT/L6 IT, Oligo, and VLMC by SpatialScope using multiple slices. Middle: spot-level expression levels of the corresponding cell-type-specific marker genes in the original Visium data. Bottom: refined single-cell resolution expression levels of the corresponding marker genes by SpatialScope.
**e** Visualization of some representative molecular interactions detected in the 3D aligned single-cell resolution spatially resolved transcriptomic data produced by SpatialScope. The scatter plot shows the expression level of ligand-receptor pairs in corresponding cell type pairs. The expression of ligands and receptors is colored orange and green, respectively. L cell type that expresses ligand genes, R cell type that expresses receptor genes.

leveraging spatial neighboring information, we considered a real two-slice dataset of spot-level mouse brain cortex data (Fig. 3a). Both of the two slices are from 10x Visium and they are adjacent slices from the mouse brain cortex, and these two-slices data serve as the spatial data. Separately, we used a published mouse brain scRNA-seq data (Smart-seq2)[33] as the single-cell reference, which is comprised of 14,249 cells across 23 cell types (Supplementary Fig. 17). We first segmented single cells independently in the two corresponding H&E-stained histological images from the same tissue sections and located 3777 and 3,034 cells within 812 and 794 spots from the brain cortex of slice 1 and slice 2, respectively. Using PASTE[34] to align multiple adjacent tissue slices, we then successfully constructed a 3D aligned ST data for the mouse brain cortex tissue (Fig. 3a). We applied SpatialScope to the 3D-aligned ST data for cell type identification. We evaluated the accuracy of the inferred cell type labels based on the known spatial organization of cell types in the brain cortex: The mouse brain cortex consists of four main layers of glutamatergic neurons (L2/3, L4, L5 and L6), and cell type labels identified by SpatialScope accurately reconstructed these multi-layer structures in both slices of mouse brain cortex (Fig. 3b, Supplementary Fig. 18). Alignment-based and deconvolution methods can only handle one slice at a time, and as a result, the tissue layer structure can be misidentified (Supplementary Fig. 19). By incorporating 3D spatial structure and borrowing information from adjacent slices, SpatialScope reduces cell mis-identification compared to StarDist +RCTD by taking into account neighboring cell types (Fig. 3c). For example, StarDist+RCTD misidentifies L4 and L6 IT cells in other layers for both slices due to a lack of spatial smoothness constraint, while SpatialScope accurately identifies them within their corresponding layers.

After inferring the cell type labels at the single-cell level for mouse brain cortex data, we utilized SpatialScope to infer transcriptome-wide expression levels of individual cells through gene expression decomposition. This step enabled us to conduct more detailed and informative analyses of cell-cell interactions at the single-cell resolution. By decomposing gene expressions from the spot-level to the single-cell level, we refined the spatial transcriptomic landscape of the mouse brain cortex while preserving accurate spatial patterns of gene expressions (Fig. 3d). In contrast, Tangram and CytoSPACE were unable to reconstruct the expected spatial expression patterns of certain marker genes at the single-cell resolution (Supplementary Fig. 19d). Furthermore, we demonstrate that the spatially resolved transcriptomic data at single-cell resolution, generated by SpatialScope with the aid of 3D alignment, allowed us to infer reliable spatially proximal cell-cell communications (Fig. 3e, Supplementary Fig. 21a). Compared to the limited ligand-receptor signaling detected in a single slice alone, we observed widespread proximity interactions between Parvalbumin-positive neurons (Pvalb) and Oligodendrocytes (Oligo) when analyzing the 3D aligned ST data (Supplementary Fig. 21d, e). The identified ligand and receptor pairs exhibited strong enrichment in multiple biological processes/pathways crucial for neuronal development in the cortex, including synapse organization and assembly, oligodendrocyte differentiation, and regulation of gliogenesis (Supplementary Fig. 21b)[35]. For instance, the cell-cell communication mediated by the interaction between *Nrg1* and *Erbb4* is well-documented, with Neuregulin ligands playing a role in the proliferation, survival, and maturation of oligodendrocytes through the *Erbb4* pathways[36]. Another example is the suggested communication and migration between oligodendrocytes and microglia mediated by *Spp1-Itgav*[37]; our analysis indicates that this molecular interaction may also occur between Pvalb neurons and oligodendrocytes, providing a potential direction for further investigation. Additionally, we detected extensive cellular communications between neuronal subtypes, such as *Adcyap1-Adcyap1r1* between L2/3 IT and L5 IT, *Efna5-Epha5* between L5 IT and Lamp5, and *Sema6d-Plxna1* between L5 IT and L6 CT (Fig. 3e). These molecular interactions have been reported to be critical for

cortical development in the brain[38, 39]. The interacting cell types identified by SpatialScope provide a more comprehensive understanding of cellular and molecular interactions in the cortex.

## SpatialScope enables high resolution identification of cell types and candidate pathways for cellular communication in human heart tissue

The human heart is a highly functionally coordinated organ, and different cell types within the same tissue must act in concert with precise feedback and control. Previous single-cell profiling of the human heart identified cellular subtypes with high levels of specialization in their gene expression, corresponding to their roles in regeneration/renewal or as fully differentiated cells that participate in blood circulation and pacing[40]. With spatial transcriptomics, there is an additional opportunity to understand these cellular specializations in the context of the complex architecture of the human heart. We applied SpatialScope to a real spatial transcriptomics (ST) dataset of adult heart tissue profiled at the spot-level[41] and demonstrated that decomposed single-cell transcriptomes enable the localization of cellular subtypes at a high resolution. Furthermore, the assessment of ligand-receptor co-expression in neighboring cells reveals candidate pathways that facilitate cellular communication in a given tissue region.

First, we segmented single cells in the corresponding H&E stained image and located 10,734 cells within 3813 spots in the whole slice (Fig. 4a, Supplementary Fig. 22). As the paired single-cell reference (produced from the same sample as the ST data) is not available, we used as reference another human heart snRNA-seq atlas[40] consisting of 10 major cell types, including cardiomyocytes to less common adipocytes and neuronal cells (Supplementary Fig. 23). SpatialScope learned the distribution of the gene expression in each cell type from this atlas via a deep generative model. The "pseudo-cells" generated using this learned model are indistinguishable from existing real cells in the reference data (Fig. 4c), laying the foundation for SpatialScope to accurately resolve spot-level ST data containing multiple cells to single-cell resolution. The overall cell-type composition across all spots identified by SpatialScope was highly consistent with that of the snRNA-seq reference from the same tissue type (the heart left ventricle) (Fig. 4b). These results further validate the performance of SpatialScope on real data beyond the simulated dataset. Alignment-based methods, on the other hand, did not provide satisfactory estimations of cell-type composition. Tangram mis-identified many cells in the left ventricle as atrial cardiomyocytes, and CytoSPACE could not identify pericytes, a major cell type in human heart tissue. The SpatialScope estimated cell-type compositions remained highly consistent even when using different non-paired human heart snRNA-seq atlases as reference (Supplementary Fig. 24), suggesting that it is robust to the choice of reference data in real data analyses, which is important during practical implementation.

That SpatialScope can construct pseudo-cells with inferred gene expressions offers a unique advantage over other methods: through deep learning we recover additional information from each spot that is missing in the original ST data due to dropouts of low expression genes, and this enables statistically meaningful analysis of relative expression between cells (Fig. 4e). To illustrate this feature, we focused on a region of interest (ROI) that shows a spatial pattern characterized by vascular cells. Figure 4d shows that the SpatialScope inferred smooth muscle cells (SMC) accurately reside in the areas containing vascular structures, as marked by the pan-SMC marker gene *MYH11* in the original ST data and by the H&E staining in the histological image (Fig. 4d, e). In comparison, alignment-based methods Tangram and CytoSPACE were unable to identify SMCs in this region (Supplementary Fig. 25a). Cell type deconvolution methods RCTD and spatialDWLS performed better and correctly identified SMCs, while CARD and Cell2location incorrectly identified many endothelial cells (EC) and atrial cardiomyocytes, respectively (Supplementary Fig. 25b). The

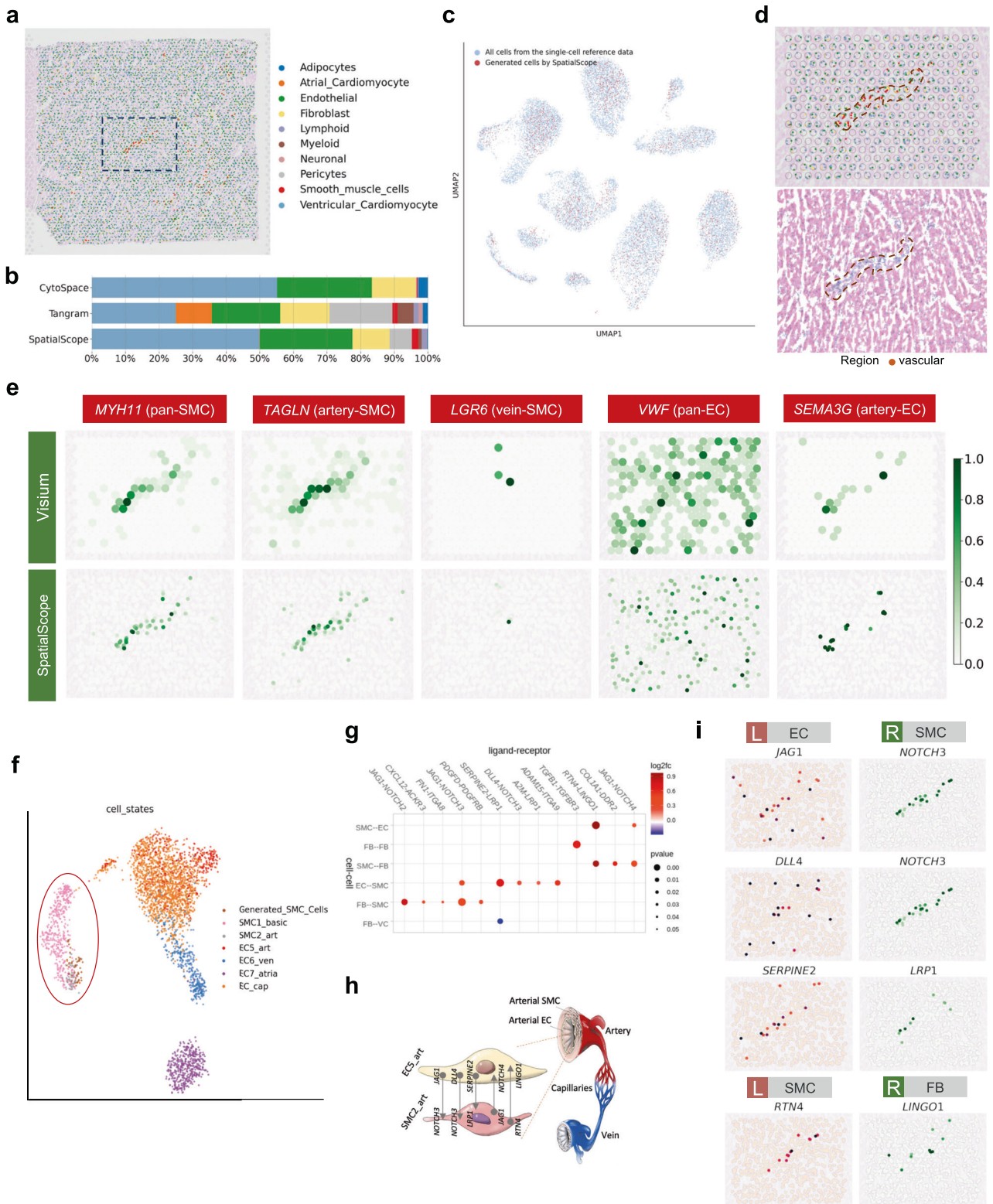

SpatialScope inferred results also indicate that this region has a much higher expression level of *TAGLN* than *LGR6* (Fig. 4e). In both brain[42] and cardiac vasculature[40], *TAGLN* was previously found to be highly expressed in arteriole SMCs while much lower in venous SMCs, and high *LGR6* expression was found to be associated with venous SMC[42]. Based on these previous atlas studies, which labeled *TAGLN*-high/*LGR6*-low SMCs as arterial, our result classifies this region as being arterial rather than venous. It is worth noting that the same conclusion could also be drawn from the raw ST data as their expression patterns were highly similar (Fig. 4e), but the single-cell resolution ST generated by SpatialScope allowed us to significantly increase the confidence of the conclusion. To see this, we projected the inferred single-cell level gene expression profiles of SMCs in this region onto the UMAP of all SMCs and ECs in the snRNA-seq reference, obtaining a global view of SMCs identified by SpatialScope. This reveals that the global gene expression of inferred SMCs are clustered with real arterial SMC rather

**Fig. 4 | Analysis of vascular region in spot-based human heart ST data. a** Cell type identification result at single-cell resolution for whole slice 10x Visium data of human heart using SpatialScope. The background is H&E staining of human heart. The black dotted line indicates ROI. **b** Inferred cell type compositions across the whole slice by SpatialScope, Tangram and CytoSPACE. **c** UMAP of scRNA-seq reference data (blue dots) and the pseudo cells (red dots) generated by the deep generative model. **d** Top, cell type identification result at single-cell resolution in ROI. Bottom, H&E staining of the heart ROI. The red dotted line indicates vascular location. **e** Expression of SMC and EC marker genes in raw Visium data (top) and single-cell transcriptomes generated by SpatialScope (bottom). **f** The UMAP plot of SMCs in refined single-cell resolution spatial data generated by SpatialScope and cells from all subgroups of SMC and EC in snRNA-seq reference. The red circle highlights the overlap of inferred SMCs with real arterial SMCs rather than venous ones. **g** Dot plot of ligand-receptor pairs that exhibit spatially resolved cell-cell communications inferred from SpatialScope generated single-cell resolution spatial data. SMC smooth muscle cells, EC endothelial cells, FB Fibroblast, VC ventricular cardiomyocyte. *p* values were calculated under the null condition in the permuted data with the two-sided test. **h** Schematic of the vascular cells and inferred cell-cell interactions between SMC and EC in the arteries. **i** Visualization of molecular interactions between EC and SMC, SMC and FB using single-cell resolution gene expression profiles generated by SpatialScope. The scatter plot shows the expression level of ligand-receptor pairs. Source data are provided as a Source Data file.

than venous ones (Fig. 4f), indicating that SpatialScope accurately identified the arterial SMC. Other methods could not distinguish these subtypes (Supplementary Fig. 25c).

Spatially resolved single-cell expression profiles inferred by SpatialScope can further facilitate downstream analysis, for example in exploring cell-cell communication between ECs and SMCs in arteries (Fig. 4h). We applied Giotto[10] to identify statistically significant ligand-receptor (LR) interactions between these two cell types when in close proximity (Fig. 4g) and found LR expression patterns that are consistent with previous studies[40]. Spatial co-expression patterns of these LR pairs was also verified by visual inspection (Fig. 4i, Supplementary Fig. 26). We further noted that the interacting ECs are arterial, marked by *SEMA3G* expression (Fig. 4e), which is concordant with our previous observation that these SMC are the arterial subtype. Among the LR-pairs we identified as significant, Notch receptor-ligand interactions (e.g., *JAG1-NOTCH3*, *DLL4-NOTCH3*) are known to be essential for regulating vascular smooth muscle proliferation and differentiation[43, 44], and *SERPINE2-LRP1* has been reported to act as a protector of vascular cells against protease activity[45, 46]. *RTN4-LINGO1* is commonly detected in brain tissue due to its importance in regulating neuronal development[47]. With the inferred gene expression profiles with single-cell resolution, here our results indicate that *RTN4-LINGO1* has a spatially strong co-expression pattern in the human heart vascular region (Fig. 4i). The RTN family of genes is also known by another name, the Nogo family, and *RTN4* protein products are widely expressed in many cell types but most highly expressed on the surface of glial cells. Both *RTN4* and *LINGO1* are found to be expressed in multiple cell types, including smooth muscle cells and endothelial cells[48]. Literature has reported the interaction of this ligand-receptor in the brain[47, 49, 50], and the Nogo-B isoform was found to be important in regulating vascular homeostasis and remodeling in mouse models[51]. Further research is needed to uncover the tissue-specific mechanisms and roles of the *RTN4-LINGO1* pair in the human heart.

## SpatialScope enables accurate correction of dropouts in spot-level ST data

Various spatial technologies differ in their resolution; as an example, Slide-seqV2 can achieve a higher spatial resolution than the Visium technology but the trade-off is a lower transcript capture rate[52]. For example, in a cerebellum Slide-seq V2 dataset with 10,975 cells within 8952 spots (Fig. 5a), 98.55% entries of the gene expression matrix are zero and the median UMI counts per spot is about 300[14]. In this dataset, some marker genes exhibit unusual sparsity (Fig. 5d, Supplementary Fig. 27), with total UMIs across all spots as low as 25 in some cases (*Klf2*). We can also leverage SpatialScope to correct the low-detection in situ transcripts, inferring the missing signals using the gene expression distribution learned from the single-cell reference (Supplementary Fig. 28). As shown in Fig. 5a, SpatialScope correctly assigned cell type labels and captured the three-layer architecture (molecular layer, Purkinje cell layer and granular layer) of the cerebellum[53, 54]; these high resolution single-cell level results are consistent with spot-level RCTD results[14]. Other methods produced noisy

results and even incorrectly estimated cell type proportions: Cell2location missed most Astrocytes; SpatialDWLS wrongly detected a large number of Fibroblasts in the Purkinje cell layer; alignment-based methods Tangram and CytoSPACE could not reconstruct the granular layer, suggesting that alignment-based methods are not robust to low capture rate data (Supplementary Fig. 29).

To evaluate the performance of dropouts correction for SpatialScope, we randomly subsampled the UMIs of existing marker genes with high-capture rates to mimic the technical dropouts, and then applied SpatialScope to check if we could accurately recover the spatial expression patterns of these marker genes (Fig. 5b). Specifically, we selected 22 marker genes with high-capture rates, where the median UMIs is about 3600 across all spots, and then subsampled their UMIs to 50, 100, 200. Notably, SpatialScope showed the best performance of the dropout correction in all settings in terms of mean absolute error (MAE) and PCC (Fig. 5c, Supplementary Fig. 30). As the subsampled UMIs increased, SpatialScope further improved the correction accuracy but the performance of Tangram plateaued. We then used SpatialScope to correct low-capture genes in Slide-seq data. A close inspection of the corrected sparse marker genes showed clear expression patterns concordant with spatial cell type organization, indicating that SpatialScope can effectively address the dropout issue in Slide-seq ST data (Fig. 5d, Supplementary Fig. 27).

Low capture rates mean that many ligand and receptor pairs are also sparsely captured, making it difficult to perform relevant downstream analyses. The SpatialScope-corrected Slide-seq data imputes genes with low-capture rates, enabling further calculation of cell-cell communications. For example, the cellular communication mediated by *Psap* and *Gpr37l1* between molecular layer interneuron type 1 (MLI1) cells and astrocytes was only detected in the corrected data (Fig. 5g). Astrocytes are reported to have neuroprotective effects on neurons through the *Gpr37l1* pathway[55, 56], supporting the cell-cell interactions we identified in the corrected data; In contrast, raw Slide-seq data was too sparse to detect this (Fig. 5h-i). We detected many cellular interactions that are concordant with existing literature (Fig. 5e). For example, basket cells (e.g., MLI1 and MLI2) in the molecular layer of the cerebellum is known to have a powerful inhibitory effect on Purkinje cells[53], and we indeed found the *Apoe-Sorl1* interaction between these two cell types (Fig. 5f). Notably, both *Apoe* and *Sorl1* are genes associated with Alzheimer's disease risk, and play roles in regulating the clearance of amyloid protein $\beta$[57]; the interacting cell types detected by SpatialScope may help to elucidate the underlying genetic etiology behind Alzheimer's disease.

## SpatialScope accurately imputes unmeasured genes on single molecule imaging-based ST dataset to enable global differential gene expression analysis

Finally, we investigated how SpatialScope could leverage deep generative models to impute unmeasured genes in image-based spatial transcriptomics data that only measures a panel of selected genes. We analyzed a MERFISH dataset, where the expression profiles for 254 genes were measured in 5,551 single cells in a mouse brain slice from

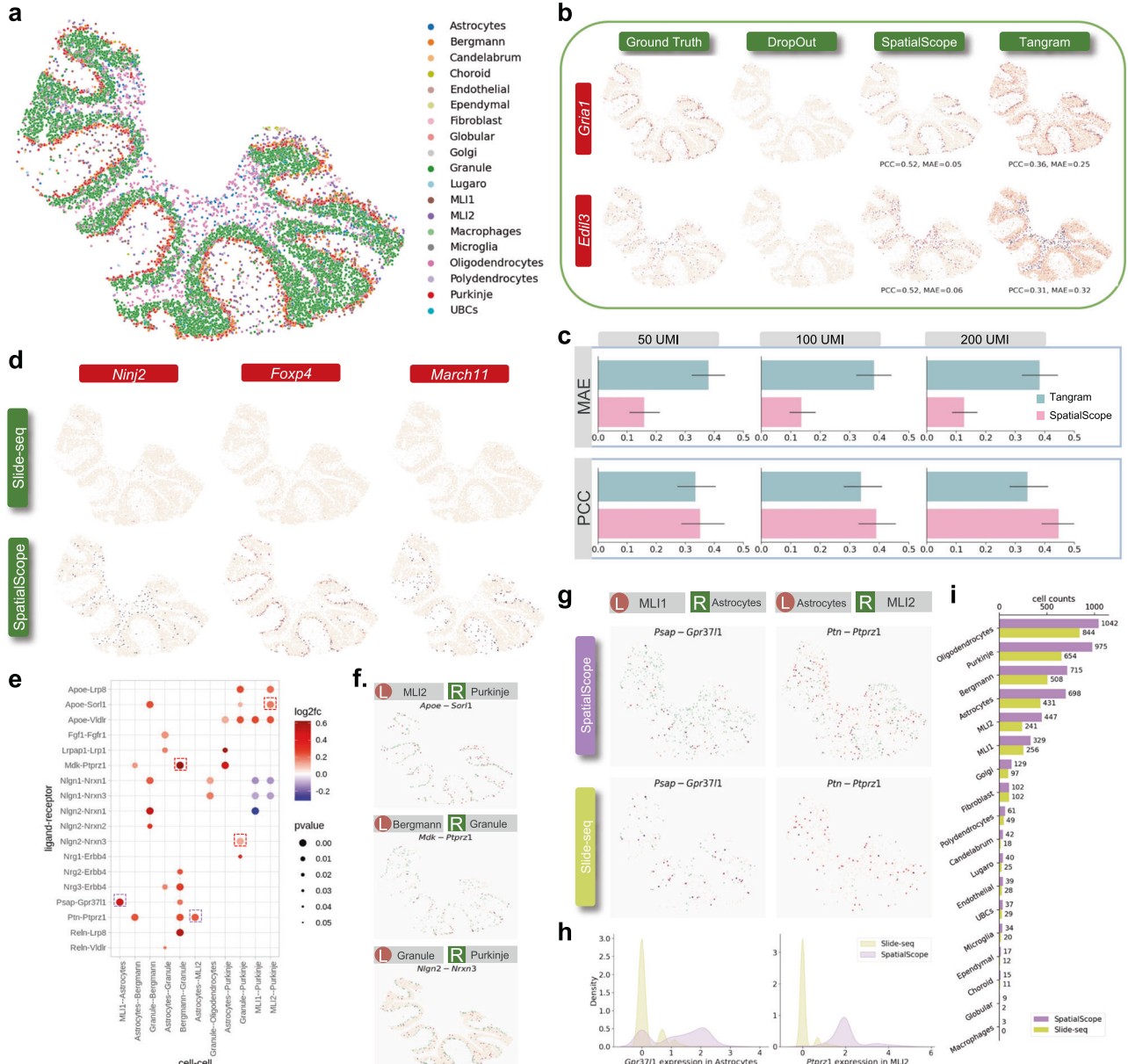

**Fig. 5 | Application of SpatialScope to Slide-seq V2 cerebellum data. a** Cell type identification results of Slide-seq V2 cerebellum data by SpatialScope. **b** Correction of two simulated low-quality spatial measurement genes. Slide-seq V2 measurements (first column), simulated low-quality spatial measurements after processing subsampling (second column), SpatialScope-corrected gene expression level (third column), and Tangram-corrected gene expression levels (Fourth column)). **c** Accuracy of dropouts correction on simulated low expression genes under different subsampling levels. Two different metrics are used to evaluate correction accuracy regarding the similarity between the corrected expression and true expression: Pearson's correlation (PCC) and Mean Absolute Error (MAE). Data are presented as mean values ± 95% confidence intervals; *n* = 22 selected marker genes. **d** Correction of low-quality spatial measurements on real data. Slide-seq measured (top), and SpatialScope corrected genes (bottom). **e** A Dot plot of ligand-receptor pairs that exhibit spatially resolved cell-cell communications inferred from corrected gene expression profiles by SpatialScope. The red frame indicates ligand-receptor pairs further visualized in **f**. Purple frame indicates ligand-receptor pairs that are newly found after dropouts correction by SpatialScope and further visualized in **g**. *p* values were calculated under the null condition in the permuted data with the two-sided test. **g** Visualization of ligand-receptor expression in both corrected Slide-seq data by SpatialScope (first row) and raw Slide-seq data (second row). The expression of ligands and receptors is colored orange and green, respectively. **h** Comparison of gene expression level for ligand/receptor between raw Slide-seq data and corrected Slide-seq data by SpatialScope, displayed by density plots. **i** Comparison of cell counts between raw Slide-seq data and corrected Slide-seq data by SpatialScope for each cell type. Source data are provided as a Source Data file.

the primary motor cortex (MOp)[58]. To perform cell type identification and learn the distribution of single-cell gene expression, we used a paired droplet-based snRNA-seq profiles from mouse MOp as the reference dataset (Fig. 6b)[18]. SpatialScope successfully learned the gene expression distribution of the single cell reference data (Supplementary Fig. 31), laying the groundwork for inferring the expressions of unmeasured genes. Using the 252 genes that were targeted by MERFISH and that overlap with snRNA-seq reference data, we assigned

cell type labels for each cell on the slice. SpatialScope successfully reconstructed the known spatial organization of cell types in the MOp of the brain cortex (Fig. 6a). Specifically, glutamatergic neuronal cells showed distinct cortical layer patterns, while GABAergic neurons and most non-neuronal cells were granularly distributed.

We compared the performance of gene expression imputation using SpatialScope with seven existing methods: Tangram[18], gimVI[19], SpaGE[20], SpaOTsc[27], novoSpaRc[28], stPlus[59], and Seurat[60]. We selected

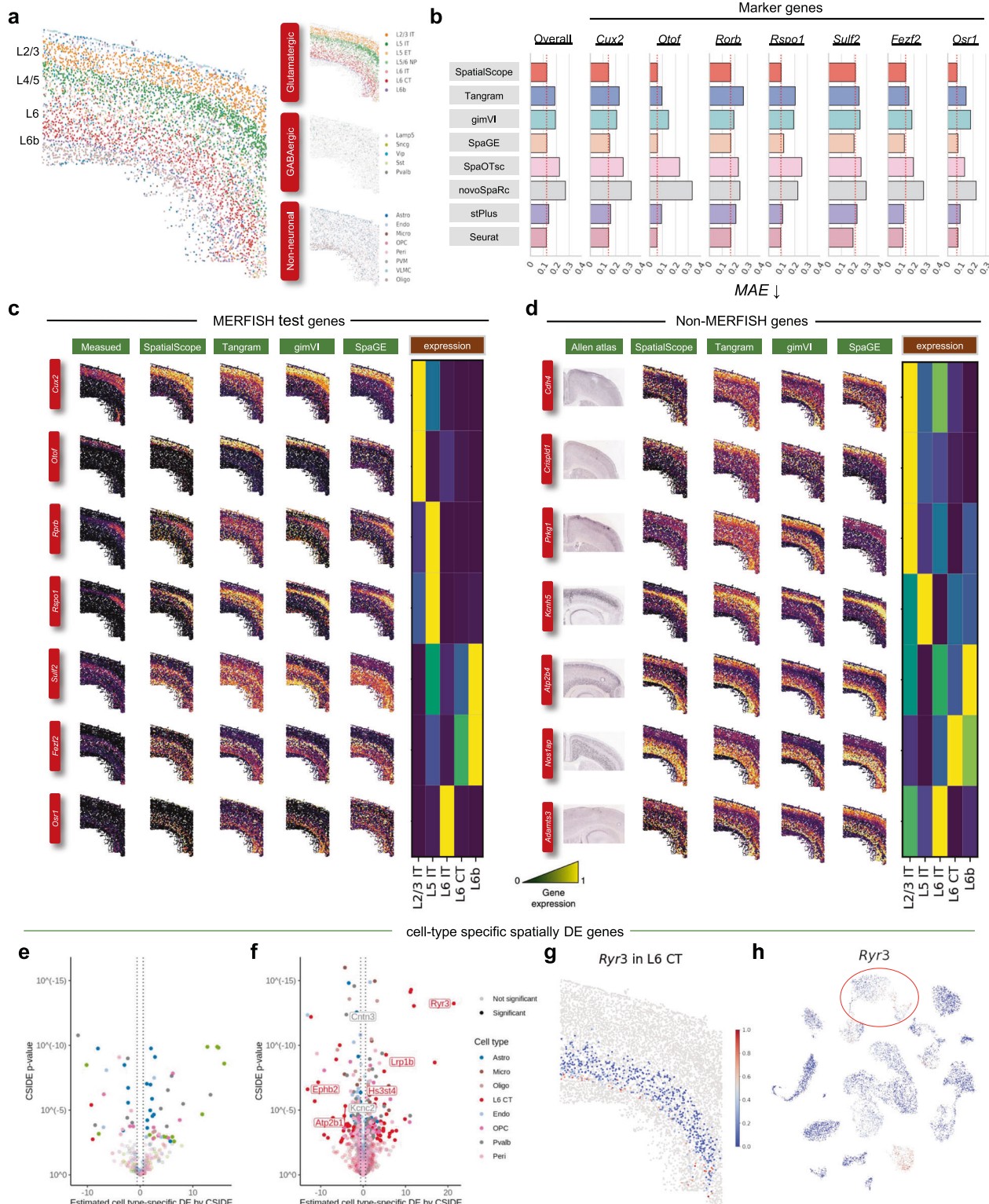

cortical layer-specific markers (*Cux2, Otof, Rorb, Rspo1, Sulf2, Fezf2,* and *Osr1*) as testing genes to visualize the predicted spatial gene expression patterns. These markers were then removed from the dataset, and the remaining genes were used as training genes, serving as input for the eight methods to predict the spatial expression pattern of the left-out marker genes. We evaluated the imputation performance by computing the mean absolute error (MAE) between the real measurements and the predicted gene expression of the testing genes. The results demonstrate a significant improvement in performance for

SpatialScope compared to Tangram, gimVI, SpaOTsc, and novoSpaRc, with improvements of 33.6%, 34.3%, 43.4%, and 53.6%, respectively. SpatialScope performs comparably to state-of-the-art methods SpaGE, stPlus, and Seurat in terms of predicting spatial gene expression of the seven cortical layer-specific markers (Fig. 6b, c, Supplementary Fig. 32).

Let's consider a marker gene, *Rspo1*, of the L5 IT layer as an example. The spatial gene expression of *Rspo1* imputed by Spatial-Scope is in accordance with the real measurement specific to the L5

**Fig. 6 | Application of SpatialScope to MERFISH data. a** Cell type identification results of MERFISH MOp data by SpatialScope. Cell type identification results in each of the three major categories are shown on the right. **b** Bar plots showing the overall mean absolute error (MAE) (first column) and MAE of seven cortical layer-specific marker genes (second to eighth columns) for each method in predicting unmeasured spatial gene expression patterns. **c** Measured and imputed expressions of known spatially patterned genes in the MERFISH dataset. Each row corresponds to a single gene. The first column from the left shows the measured spatial gene expression in the MERFISH dataset, while the second to fifth columns show the corresponding imputed expression patterns by SpatialScope, Tangram, gimVI, and SpaGE. The marker gene expression signatures in the snRNA-seq reference are displayed with a heatmap plot (sixth column). **d** Measured and imputed expressions of Non-MERFISH genes. Each row corresponds to a single gene. The first column from the left shows the ISH images from the Allen Brain Atlas, while the second to fifth columns show the corresponding imputed

expression patterns by SpatialScope, Tangram, gimVI, and SpaGE. The gene expression signatures in the snRNA-seq reference are displayed with a heatmap plot (sixth column). **e** Volcano plot of C-SIDE cell-type-specific spatial differential expression (DE) results for the MERFISH dataset, considering a total of 252 genes. Color represents cell types, a subset of significant genes is labeled, and dotted lines represent a 1.5-fold change cutoff. **f** Volcano plot of C-SIDE spatial DE results for the imputed MERFISH dataset by SpatialScope, considering a total of 1,938 genes including genes in the original MERFISH dataset and imputed Non-MERFISH genes by SpatialScope. $p$ values in **e** and **f** were calculated by $z$-statistics with the two-sided $z$-test. Benjamini-Hochberg procedure was used to control FDR in the context of multiple testing. **g** Spatial visualization of *Ryr3*, identified by C-SIDE as differentially expressed in L6 CT. Color shows the expression change of *Ryr3* across L6 CT. **h** The expression profile of *Ryr3* in the single-cell reference data. The expression level change of *Ryr3* in L6 CT cell type is outlined in the red circle. Source data are provided as a Source Data file.

layer. In contrast, Tangram, SpaGE, SpaOTsc, and novoSpaRc overestimate the expression of *Rspo1* outside the L5 layer, while gimVI and stPlus incorrectly expressed the gene in the positions of Oligo and Astro cell types. Next, we used all overlapping genes between the MERFISH data and the single-cell reference data as training genes and evaluated the imputation performance of non-MERFISH genes. Since ground truth data for these non-MERFISH genes is unavailable, we utilized the Allen ISH dataset[61] for validation purposes. We found that other methods tended to overestimate the spatial expression of some layer-specific marker genes (e.g., *Cdh4*, *Prkg1*) (Fig. 6d, Supplementary Fig. 33). SpatialScope also shows high robustness when imputing low-abundance or variable genes, and even non-brain tissue markers (see Supplementary Note sections 2.9.12, 2.9.13). It can predict expressions that are consistent with the gene expression signatures in the snRNA-seq reference, even when the expression levels measured in the MERFISH data are very low and have little spatial pattern (Supplementary Figs. 63–70). Additionally, it can predict spatial expression patterns that are consistent with the Allen ISH dataset when predicting kidney, bone, and lung marker genes (Supplementary Figs. 71–73).

SpatialScope increases the gene throughput of MERFISH from 254 to thousands of genes, enabling us to conduct wide-ranging downstream analysis such as detection of spatially differentially expressed (DE) genes. We first applied a recently developed tool, C-SIDE[62], to detect cell-type specific spatially DE genes on the imputed MERFISH dataset. As expected, compared to 63 cell-type specific spatially DE genes detected in MERFISH genes under an FDR of 1% (Fig. 6e), the number of significant genes with FDR < 1% increases to 293 by incorporating the imputed Non-MERFISH genes (Fig. 6f, Supplementary Fig. 34b). For example, *Ryr3* encodes a calcium release channel that affects cardiac contraction, insulin secretion, and neurodegeneration by altering the levels of intracellular $Ca^{2+}$[63]. The expression of *Ryr3* in L6 CT shows a spatial pattern that coincides with the L6b cell boundary (Fig. 6g and Supplementary Fig. 34a), suggesting the potential communication between L6 CT and L6b through *Ryr3*. Interestingly, the expression signature of *Ryr3* in the single-cell reference data also suggests its diverse expression in L6 CT and L6b (Fig. 6h), and the transition region between these two cell types in single-cell reference shows higher expression, which is perfectly concordant with what we observed in the imputed spatial expression pattern of *Ryr3* in MERFISH data. This concordance highlights the value of SpatialScope in integrating the merits of both single cell reference and lower throughput high precision spatial transcriptomic data such as MERFISH. Next, we considered the spatially DE genes across the entire MERFISH data instead of restricting to specific cell types. We applied SPARK-X[64] and identified 243 genes that exhibit spatially DE patterns in a global perspective, which was 2.3 times more than the number of DE genes detected in MERFISH genes (Supplementary Fig. 34c). Visualizing a few representative non-MERFISH DE genes clearly shows their significantly spatially distinct expression patterns (Supplementary Fig. 34d). For

example, *Lingo2* encodes a transmembrane protein that positively regulates synapse assembly[65], and the genetic variants of *Lingo2* have been reported to be linked to Parkinson's disease (PD) and essential tremor (ET)[66, 67]. The spatial expression pattern of *Lingo2*, highly expressed in the upper cortical layer, imputed by SpatialScope may shed light on the genetic etiology of PD/ET in brain MOp.

## Discussion

Fine-grained cell gradients are critical for understanding cellular communication within tissues, which requires that ST technologies achieve the detection of the whole transcriptome at single-cell resolution. However, existing ST technologies often have limitations either in spatial resolution, capture rate of the genes, or the number of genes that can be profiled in one experiment. Here we developed a unified framework SpatialScope to address these limitations.

SpatialScope is applicable to different ST technologies and can achieve several important functions. First, SpatialScope recovers single-cell resolution data from seq-based technologies (e.g. 10X Visium) that do not have single-cell resolution. Consequently, single-cell resolution ST data produced by SpatialScope enables the detection of spatially resolved cellular communication, which is almost impossible for ST data that does not have cellular resolution. Spatially resolved cell-cell communications between each paired cell mediated by ligand-receptor interactions can be robustly inferred and visualized, leading to decoding spatial inter-cellular dynamics in tissues. Second, SpatialScope improves the power and precision of molecular interaction by correcting for genes that has low capture rate in higher-resolution spatial data, such as Slide-seq. Some signals missing in the raw ST data can be detected after the correction for dropouts by SpatialScope. Third, SpatialScope imputes unmeasured genes for image-based ST technology that cannot measure the whole transcriptome, such as MERFISH, allowing the discovery of more biologically meaningful signals. Fourth, SpatialScope can integrate multiple slices of ST data, which enables better cell type identification and the detection of cell-cell communication by increasing the effective sample size.

SpatialScope's ability to accurately and robustly resolve the spot-level data towards higher resolution and expand from signature to transcriptome-wide scale expression comes from the fact that SpatialScope leverages the deep generative model to approximate the distribution of gene expressions accurately from the scRNA-seq reference data. Rather than directly applying learned distribution from single-cell reference data, SpatialScope accounts for the platform effects between single-cell reference and ST data. Cell type identification and gene expression decomposition results would not be satisfactory if the platform effects are not appropriately corrected. With these innovations in its model design, SpatialScope serves as a unified framework which is applicable to ST data from various platforms. In the step of cell type identification, SpatialScope leverages spatial information to improve the accuracy of cell type identification

for each single cell. The inclusion of spatial information also allows straightforward extrapolation of SpatialScope to data with multiple slices, where 3D spatial information across slices is well exploited.

SpatialScope incorporates a spatial smoothness constraint imposed by the Potts model, and has demonstrated its effectiveness through simulation studies and real data analysis (Supplementary Note section 2.9.4, Supplementary Fig. 3c). However, the assumption that neighboring cells belong to the same cell type may not always be valid. A more effective approach involves enabling the model to adaptively learn from the data and assess the similarity of cell types among neighboring cells. This can be achieved by incorporating spatial location information as a key input within the Deep Graph Infomax (DGI) framework[68, 69], or by employing GCN with attention mechanism to adaptively learn the similarity of neighboring spots/cells[70, 71]. We leave this direction for future work.

Although several widely adopted ST technologies, such as Visium, Stereoseq, and Slide-lock, presently offer paired histological images, it is anticipated that the availability of such paired images will expand in the future as these techniques become more accessible and cost-effective. However, circumstances may arise where histological images are unavailable, and alternative types of image data, such as single-channel nuclear image data (e.g., DAPI image), are provided instead. Noted that Step 1, Nucleus segmentation serves as a building block to quantify the cell count at each spot. Consequently, it is possible to utilize other types of images alongside histological images, leveraging segmentation methods such as Baysor and DeepCell, to determine the cell count within each spot. In situations where image data is unavailable due to experimental failure or other factors, the initial step of nucleus segmentation can be substituted with singlet/doublet classification for Slide-seq data, based on the assumption that a maximum of two cells coexist within a spot. For other lower-resolution ST data, it is feasible to develop alternative methods for estimating cell numbers by incorporating information of both cell type compositions and the total number of UMIs within the spots. Spots exhibiting higher UMIs and more diverse cell type compositions are more likely to contain a larger number of cells. These alternative strategies enable the estimation of cell numbers within spots, even when paired images are not available. By leveraging information about cell type compositions and UMIs, valuable insights can be gained regarding cell counts within ST datasets.

While SpatialScope has shown its superior performance, it can be time-consuming to train a generative model to approximate the distribution from single-cell reference data. In our real data analysis, the training time of the deep generative models is measured in hours or days. Although the generative model can be pre-trained using single-cell atlas data sets and we only need to train each dataset once, ST data analysis can still benefit a lot from more computationally efficient methods that reduce the computational complexity for learning deep generative models[72].

While our work focused on the analysis of cell-cell communications and spatially DE genes detection, we anticipate that refined single-cell resolution spatial transcriptomic data generated by SpatialScope can be very useful in many other downstream applications. Examples include unraveling spatiotemporal patterns of cells[73], analysis of cellular interactions between tumor and immune cells in disease or cancer tissue, and inference of differentiation trajectories[74]. We believe that SpatialScope can serve as a very useful tool in providing single-cell resolution ST data, facilitating detailed downstream cellular analysis, and generating biological insights.

## Methods

To characterize spatially-resolved transcriptome-wide gene expression at single-cell resolution, we introduce SpatialScope as a unified framework to integrate single-cell and ST data. For 10x Visium ST data, the SpatialScope method comprises of three steps: nucleus segmentation, cell type identification, and gene expression decomposition. SpatialScope can also be applied to dropout correction for Slide-seq data and transcriptome-wide gene expression imputation for image-based ST data, such as MERFISH data.

### Nucleus segmentation

Accurate segmentation of nuclei/cells in microscopy images is an important step to locate cells and count the number of cells within a spot. Considering the widespread use of 10 Visium data with H&E images, we conducted a comprehensive evaluation of several segmentation methods: StarDist[75], Cellpose[76], Baysor[77], and DeepCell[78], specifically for H&E-stained images (Supplementary Note section 2.1). Our comprehensive analysis reveals that StarDist outperforms the other methods, making it the most effective tool for nucleus segmentation in HE-stained histological images (Supplementary Figs. 35–37). On the other hand, Baysor and DeepCell exhibit inferior performance, likely due to their lack of specific design for H&E images. StarDist's exceptional performance, as evidenced by high DICE and AJI scores, underscores its robustness and reliability in accurately segmenting nuclei in H&E images. Therefore, we employ StarDist as the default tool for nucleus segmentation on H&E-stained histological images. After segmentation, we denote $M_i$ as the number of detected cells at the $i$-th spot, $i = 1, …, I$, where $I$ is the total number of spots.

### Cell type identification

Suppose we have $K$ cell types in a single-cell reference data. The expression counts of $G$ genes have been measured to capture the whole transcriptome in the scRNA-seq data. Let $k_{i,m} \in \{1, 2, …, K\}$ be the cell type of the $m$-th cell at spot $i$, where $m = 1, …, M_i$. Our goal is to infer the cell type vector $\mathbf{k}_i = \{k_{i,m}\}$ at spot $i$ by integrating scRNA-seq and ST data.

As inspired by RCTD[14], we consider the following probabilistic model for cell type identification in ST data by incorporating scRNA-seq reference data,

$$y_{i,g}|\lambda_{i,g} \sim \text{Poisson}\left(N_i \lambda_{i,g}\right), \ \log\left(\lambda_{i,g}\right) = \alpha_i + \log\left(\frac{1}{M_i}\sum_{m=1}^{M_i} \mu_{k_{i,m},g}\right) + \gamma_g + \varepsilon_{i,g}$$

(3)

where $y_{i,g}$ is the observed gene expression counts of gene $g$ at spot $i$, $N_i$ is the total number of unique molecular identifiers (UMIs) of spot $i$, $\lambda_{i,g}$ is the relative expression level of gene $g$ at spot $i$, $M_i$ is the number of cells in spot $i$ inferred from the last step, $\varepsilon_{i,g} \sim \mathcal{N}(0, \sigma_\varepsilon^2 \mathbf{I})$ is a random effect to account for additional noise, and $\mu_{kg}$ represents the mean expression level of cell type $k$ and gene $g$, which can be estimated from annotated single-cell reference data. Both $\gamma_g$ and $\alpha_i$ are designed to address the batch effect between single-cell reference and ST data. More specifically, $\gamma_g \sim \mathcal{N}(0, \sigma_\gamma^2 \mathbf{I})$ represents a gene-specific random effect to account for expression differences of a gene $g$ between single-cell and ST platforms, and $\alpha_i$ is the spot-specific effect to account for differences of a gene set across platforms.

Recall that the RCTD model is given as $\log(\lambda_{i,g}) = \alpha_i + \log(\sum_{k=1}^K \beta_{i,k}\mu_{k,g}) + \gamma_g + \varepsilon_{i,g}$, where $\beta_{i,k}$ is the proportion of cell type $k$ at spot $i$. Our model differs from RCTD in the term $\frac{1}{M_i}\sum_{m=1}^{M_i} \mu_{k_{im},g}$, which is the average of the mean expression level of cell types corresponding to the $M_i$ cells at spot $i$. In other words, our model can be viewed as a discrete version of RCTD which was developed to estimate the continuous cell type proportions. The benefits of our discrete version are two-fold. First, given the accurate number of detected cells from image segmentation, it allows us to achieve cell type identification at single-cell resolution. Second, it also enables the incorporation of spatial smoothness constraints to improve the accuracy of cell type identification. In contrast, RCTD can only impose the simplex constraint (i.e., $\sum_{k=1}^K \beta_{i,k} = 1$ and $\beta_{i,k} \geq 0$) when estimating $\beta_{i,k}$, leading to suboptimal results. To incorporate spatial smoothness

in the distribution of the cell types, we assume a prior given by the Potts model for cell types $\mathbf{K} = \{k_{i,m}\}$,

$$p(k_{im}|k_{-\{i,m\}}) = \frac{1}{Z}\exp\{-U(\mathbf{K})\}, \tag{4}$$

where $U(\mathbf{K}) = \sum_{\{i,m\}}\{\{i',m'\}\in\mathcal{N}_{i,m}\nu[1-\mathbb{I}(k_{i,m}=k_{i',m'})]$, $\mathbb{I}(\cdot)$ is the indicator function which equals to 1 when $k_{i,m}=k_{i',m'}$ and 0 otherwise, $Z$ is a normalization constant, $\mathcal{N}_{i,m}$ is the set of neighbors of the $m$-th cell in spot $i$ and $-\{i,m\}$ denote all the cells other than $(i,m)$ cell. Parameter $\nu$ controls the smoothness of cell type labels. The larger the $\nu$, the smoother the cell type labels.

Now we develop an iterative algorithm to identify cell type label $k_{i,m}$ based on maximum a posterior (MAP) estimate, where $i=1,2,...,I$, and $m=1,2,...,M_i$. Meanwhile, we are also interested in $\gamma_g$ which will used to correct gene-level batch effects between different platforms. First, we estimate $\mu_{k,g}$ by calculating mean expression of gene $g$ and cell type $k$ from single-cell reference data. Next, we follow RCTD's strategy to accurately approximate $\gamma_g$ by viewing ST data as a bulk RNA-seq data using the convenient property of Poisson distribution[14]. Other parameters, including $\alpha_i$, can be obtained accordingly (Supplementary Methods). Then we iteratively find MAP estimate of $\{k_{i,m}\}$ and the estimate of $\sigma_\varepsilon$. The derivation of the MAP estimate for $\{k_{i,m}\}$ is as follows. Let $\hat{\boldsymbol{\theta}}_c = \{\hat{\mu}_{k,g}, \hat{\gamma}_g, \hat{\alpha}_i, \hat{\sigma}_\varepsilon\}$ be the collection of these estimates in the above cell type identification model, where $k=1,2,...,K$, $g=1,2,...,G$, and $i=1,2,...,I$. Given $\hat{\boldsymbol{\theta}}_c$, we can obtain the MAP estimate for $\mathbf{K}$:

$$\begin{aligned}\hat{\mathbf{K}} &= \arg\max_{\mathbf{K}} \log p(\mathbf{K}|\mathbf{Y},\hat{\boldsymbol{\theta}}_c) \\ &= \arg\max_{\mathbf{K}} \log p(\mathbf{Y}|\mathbf{K},\hat{\boldsymbol{\theta}}_c) + \log p(\mathbf{K}),\end{aligned} \tag{5}$$

where $\mathbf{Y} = \{y_{i,g}\}$ represents all observed gene expression counts. The term $\log p(\mathbf{Y}|\mathbf{K},\hat{\boldsymbol{\theta}}_c)$ is the likelihood term in Equation (3) and $\log p(\mathbf{K})$ is the prior term given by Potts model (Equation (4)). Notably, this MAP estimate for $\mathbf{K}$ in Bayesian analysis represents the value that maximizes the combined influence of observed data (likelihood term) and prior beliefs (prior term), allowing for the incorporation of prior knowledge into the estimation process. For computational efficiency and scalability, we adopt iterative-conditional-mode-based scheme[79] to infer $\mathbf{K}$ by updating two labels $k_{i,m}, k_{i,\bar{m}}$ at a time. Then the distribution becomes,

$$\begin{aligned}\max_{\mathbf{K}} \log p((k_{i,m},k_{i,\bar{m}}|\mathbf{y}_i,\hat{\boldsymbol{\theta}}_c)) &= \max_{\mathbf{K}} \sum_{g=1}^{G} \log p(y_{i,g}|k_{im},k_{i\bar{m}},k_{-\{(i,m),(i,\bar{m})\}}) \\ &\quad + \log p(k_{im},k_{i\bar{m}}|k_{-\{(i,m),(i,\bar{m})\}}).\end{aligned} \tag{6}$$

By finding the MAP estimate, we not only use information from gene expression levels $y_{i,g}$ to determine the cell type labels $k_{i,m}$, but also incorporate information from its neighbors.

### Gene expression decomposition

We first learn a score-based generative model to approximate the expression distribution of different cell types from the single-cell reference data. Then we use the learned model to decompose gene expression from the spot level to the single-cell level, while accounting for the batch effect between single-cell reference and ST data.

### Learning conditional score-based generative models from single-cell reference data.

There are two major challenges in learning score-based generative models for the scRNA-seq data. First, while score-based generative models[23, 24, 80–82] can accurately approximate the distribution of images, the nature of the scRNA-seq count data, such as sparsity in the expression matrix, may hinder the capacity of score-

based generative models. Second, as given in Equation (2), we need to learn a conditional score function $\nabla_{\mathbf{x}} \log p(\mathbf{x}|k)$ rather than an unconditional score function $\nabla_{\mathbf{x}} \log p(\mathbf{x})$, where $k$ represents the cell type. The reason for learning a conditional score function has been demonstrated in the Supplementary Note section 2.9.8. It largely remains unknown how to learn the conditional score function across different cell types using a coherent neural network. Let's begin with the key idea of learning the unconditional score function and then show how to learn a conditional score function based on the single-cell reference data.

Consider the vanilla score matching problem which aims to find a neural network $\mathbf{s}_{\boldsymbol{\theta}}(\mathbf{x})$ to approximate $\nabla_{\mathbf{x}} \log p(\mathbf{x})$: $\min_{\boldsymbol{\theta}} \mathbb{E}_{p(\mathbf{x})}[\|\mathbf{s}_{\boldsymbol{\theta}}(\mathbf{x}) - \nabla_{\mathbf{x}} \log p(\mathbf{x})\|_2^2]$, where $\boldsymbol{\theta}$ represents the parameter set of the neural network. The challenge of vanilla score matching comes from the fact that high dimensional data $\mathbf{x}$ often tends to concentrate on a low dimensional manifold embedded of the entire ambient space. For data points not on the low-dimensional manifold, they would have zero probability, the log of which is undefined. Moreover, the score function can not be estimated accurately in the low density region. Fortunately, these challenges can be addressed by adding multiple levels of Gaussian noise to data. The perturbed data with Gaussian noise will not concentrate on the low-dimensional manifold, and the multiple levels of noise will increase training samples in the low-density region. Specifically, a sequence of data distributions perturbed by $L$ levels of Gaussian noise is given as $p_{\sigma_l}(\mathbf{x}^{(l)}) = \int p(\mathbf{x})\mathcal{N}(\mathbf{x}^{(l)}|\mathbf{x},\sigma_l^2\mathbf{I})d\mathbf{x}$, where $\mathbf{x}^{(l)}$ represents a sample perturbed by the noise level $\sigma_l^2$, $\sigma_L > \sigma_{L-1} > \cdots > \sigma_1 \approx 0$. To learn the score function $\mathbf{s}_{\boldsymbol{\theta}}(\mathbf{x},\sigma_l)$, we consider the following problem,

$$\min_{\boldsymbol{\theta}} \sum_{l=1}^{L} \lambda_l \mathbb{E}_{p(\mathbf{x})} \mathbb{E}_{\mathbf{x}^{(l)}\sim\mathcal{N}(\mathbf{x}^{(l)}|\mathbf{x},\sigma_l^2\mathbf{I})}\left[\left\|\mathbf{s}_{\boldsymbol{\theta}}(\mathbf{x}^{(l)},\sigma_l) - \nabla_{\mathbf{x}^{(l)}} \log p_{\sigma_l}(\mathbf{x}^{(l)}|\mathbf{x})\right\|_2^2\right], \tag{7}$$

where $\lambda_l = \sigma_l^2$ is the weight for noise level $l$ and $\nabla_{\mathbf{x}^{(l)}} \log p_{\sigma_l}(\mathbf{x}^{(l)}|\mathbf{x}) = -\frac{\mathbf{x}^{(l)}-\mathbf{x}}{\sigma_l^2}$. Based on Equation (7), the score function can be estimated by stochastic gradient methods. Let $\mathbf{x}^{(l,t)}$ be the $t$-sample at level $l$. We run Langevin dynamics in Equation (8) from $l=L$ to $l=1$ with initialization $\mathbf{x}^{(l,t=1)} = \mathbf{x}^{(l+1,t=T)}$. In the meanwhile, we progressive reduce of noise level $\sigma_l$ and decrease the step size $\eta$:

$$\mathbf{x}^{(l,t+1)} = \mathbf{x}^{(l,t)} + \eta\mathbf{s}_{\boldsymbol{\theta}}(\mathbf{x}^{(l,t)},\sigma_l) + \sqrt{2\eta}\boldsymbol{\varepsilon}^{(l,t)}, \tag{8}$$

where $\boldsymbol{\varepsilon}^{(l,t)} \sim \mathcal{N}(\mathbf{0},\mathbf{I})$. Then the obtained samples $\mathbf{x}^{(l,t)}$ at level $l=1$, $t=1,...,T$, will approximately follow the target distribution $p(\mathbf{x})$ because $\sigma_1 \approx 0$.

Now we consider learning our score function conditional on cell types based on scRNA-seq data. For computational stability, we transform the count data to its log scale and remove the mean expression level of each cell type. Specifically, let $\mathbf{x}_n^{\text{count}}$ be the gene expression counts corresponding to cell $n$ of cell type $k$. The transformation is given as $\mathbf{x}_n \leftarrow \log(\mathbf{x}_n^{\text{count}}+1) - \boldsymbol{\mu}_k$, where $\boldsymbol{\mu}_k \in \mathbb{R}^G$ is the mean expression level of cell type $k$. Later on, we will learn the conditional score function based on the transformed expression level. To incorporate cell type information, we consider the following optimization problem:

$$\min_{\boldsymbol{\theta}} \sum_{l=1}^{L} \lambda_l \mathbb{E}_{p(k)} \mathbb{E}_{p(\mathbf{x}|k)} \mathbb{E}_{\mathbf{x}^{(l)}\sim\mathcal{N}(\mathbf{x}^{(l)}|\mathbf{x},\sigma_l^2\mathbf{I})} \\ \left[\left\|\mathbf{s}_{\boldsymbol{\theta}}(\mathbf{x}^{(l)},\sigma_l,k) - \nabla_{\mathbf{x}^{(l)}} \log p_{\sigma_l}(\mathbf{x}^{(l)}|\mathbf{x},k)\right\|_2^2\right], \tag{9}$$

where the score function $\mathbf{s}_{\boldsymbol{\theta}}(\mathbf{x}^{(l)},\sigma_l,k)$ explicitly takes cell type $k \in \{1,2,...,K\}$ as its input. In principle, the score function $\mathbf{s}_{\boldsymbol{\theta}}(\mathbf{x}^{(l)},\sigma_l,k)$ can be estimated by solving the optimization problem given in

Equation (9). In practice, however, the learning process often tends to largely ignore the cell type information because the neural network will naturally focus on $\mathbf{x}^{(l)} \in \mathbb{R}^{G}$ rather than the scalar $k$. To successfully incorporate cell type information, our key idea is to embed cell type information in a vector whose dimension is comparable to $\mathbf{x}^{(l)}$. Therefore, we propose to learn the score function $\mathbf{s}_{\boldsymbol{\theta}}(\mathbf{x}^{(l)}, \sigma_l, \boldsymbol{\mu}_k)$ which takes the mean expression level of cell type $k$ as input. The benefits are two-fold. First, $\boldsymbol{\mu}_k$ provides precise information about cell type $k$. Second, $\boldsymbol{\mu}_k \in \mathbb{R}^{G}$ has the same dimension of $\mathbf{x}^{(l)}$ such that it will not be ignored. With this key idea, we can design a novel network architecture for learning the score function $\mathbf{s}_{\boldsymbol{\theta}}(\mathbf{x}^{(l)}, \sigma_l, \boldsymbol{\mu}_k)$. The details of the learning procedure are provided in Supplementary Methods. The learned score function $\mathbf{s}_{\boldsymbol{\theta}}(\mathbf{x}^{(l)}, \sigma_l, \boldsymbol{\mu}_k)$ is then used in Gene expression decomposition step (next section).

**Decomposition with a conditional score-based generative model.** Now we show how we obtain gene expression decomposition at single-cell resolution by leveraging the learned score-based generative model. One of the pertinent challenges for decomposition is the batch effects between single-cell reference and ST data. If the batch effects are not appropriately corrected, the decomposition results will not be satisfactory. Therefore, we adjust the batch effects between ST and single-cell reference data before we perform gene expression decomposition. Our batch effect correction includes two steps. Specifically, in the first step, we adjust for the gene-specific cross-platform effects using $\mathbf{y}_i = [y_{i,1} / \exp(\hat{\gamma}_1), \ldots, y_{i, G} / \exp(\hat{\gamma}_G)] \in \mathbb{R}^{G}$, where $y_{i,g}$ are the observed expression counts of gene $g$ at spot $i$ and $\hat{\gamma}_g$ is the batch effect of gene $g$ estimated under Equation (3). In the second step, we account for the difference in sequencing depth by normalizing the total count of $\mathbf{y}_i$ to the mean of the total transcript counts of individual cells from single-cell reference data. Next, we show how to decompose the normalized $\mathbf{y}_i$, which is corrected for batch effect, into single-cell resolution. Let $\mathbf{X}_i = [\mathbf{x}_{i,1}; \ldots, \mathbf{x}_{i, M_i}]$ be the expression level in the log scale, where $\mathbf{x}_{i,m}$ is the expression level of the $m$-th cell in spot $i$, and $M_i$ is the number of cells in spot $i$ inferred in the nucleus segmentation step. Our goal is to decompose gene expression from the spot-level $\mathbf{y}_i$ to the single-cell level $\mathbf{x}_{i,m}$.

Let $p(\mathbf{x} | k_{i,m})$ be the gene expression distribution of cell type $k_{i,m}$, where the cell type labels for the cells in spot $i$, $\mathbf{k}_i = \{ k_{i,1}, \ldots, k_{i, M_i} \}$ are inferred as in the Cell type identification step. As outlined in the methods overview, we consider the following probabilistic model for gene expression decomposition,

$$
\mathbf{y}_i | \mathbf{x}_{i,1}, \ldots, \mathbf{x}_{i, M_i} \sim \mathcal{N}\left( f\left(\mathbf{x}_{i,1}, \ldots, \mathbf{x}_{i, M_i}\right), \sigma_y^2 \mathbf{I} \right), \quad \mathbf{x}_{i,m} \sim p(\mathbf{x}_i | k_{i,m}) \quad m = 1, \ldots, M_i
\tag{10}
$$

where $f\left(\mathbf{x}_{i,1}, \ldots, \mathbf{x}_{i, M_i}\right) = \sum_{m=1}^{M_i}\left(\exp\left(\mathbf{x}_{i,m} + \boldsymbol{\mu}_{k_{i,m}}\right) - 1\right)$ transforms the log-scale expression level to the count scale. To obtain the decomposition, we use Langevin dynamics to get samples from the posterior $p(\mathbf{X}_i | \mathbf{y}_i, \mathbf{k}_i)$,

$$
\begin{aligned}
\mathbf{X}_i^{(t+1)} &= \mathbf{X}_i^{(t)} + \eta \nabla_{\mathbf{X}_i} \log p\left(\mathbf{X}_i^{(t)} | \mathbf{y}_i, \mathbf{k}_i\right) + \sqrt{2\eta}\,\boldsymbol{\varepsilon}^{(t)}, \\
&= \mathbf{X}_i^{(t)} + \eta\left[\nabla_{\mathbf{X}_i} \log p\left(\mathbf{y}_i | \mathbf{X}_i^{(t)}\right) + \nabla_{\mathbf{X}_i} \log p\left(\mathbf{X}_i^{(t)} | \mathbf{k}_i\right)\right] + \sqrt{2\eta}\,\boldsymbol{\varepsilon}^{(t)},
\end{aligned}
\tag{11}
$$

where $\nabla_{\mathbf{X}_i} \log p(\mathbf{y}_i | \mathbf{X}_i^{(t)}) = \nabla_{\mathbf{X}_i}(\frac{1}{2\sigma_{yl}^2} | \mathbf{y}_i - f(\mathbf{x}_{i,1}^{(t)}, \ldots, \mathbf{x}_{i, M_i}^{(t)})|^2)$ and $\nabla_{\mathbf{X}_i} \log p(\mathbf{X}_i^{(t)} | \mathbf{k}_i)$ is given by the learned score function $\mathbf{s}_{\boldsymbol{\theta}}(\mathbf{x}^{(l)}, \sigma_l, \boldsymbol{\mu}_k)$. Similar to Equation (8), we progressively reduce noise level $\sigma_l$ (from $l = L$ to $l = 1$) and initialize later stage with samples from the

previous stage $\mathbf{X}^{(l, t=1)} = \mathbf{X}^{(l+1, t=T)}$,

$$
\mathbf{X}^{(l, t+1)} = \mathbf{X}^{(l, t)} + \eta\left[\nabla_{\mathbf{X}_i} \log p\left(\mathbf{y}_i | \mathbf{X}_i^{(l, t)}\right) + \begin{bmatrix} \mathbf{s}_{\boldsymbol{\theta}}\left(\mathbf{x}_{i,1}^{(l, t)}, \sigma_l, \boldsymbol{\mu}_{k_{i,1}}\right) \\ \vdots \\ \mathbf{s}_{\boldsymbol{\theta}}\left(\mathbf{x}_{i, M_i}^{(l, t)}, \sigma_l, \boldsymbol{\mu}_{k_{i, M_i}}\right) \end{bmatrix}\right] + \sqrt{2\eta}\,\boldsymbol{\varepsilon}^{(l, t)},
\tag{12}
$$

where $\boldsymbol{\varepsilon}^{(l, t)} \sim \mathcal{N}(\mathbf{0}, \mathbf{I})$. The obtained samples $\mathbf{X}^{(l, t)}$ at level $l = 1, t = 1, \ldots, T$, will be posterior samples from $p(\mathbf{X}_i | \mathbf{y}_i, \mathbf{k}_i)$. By averaging samples from Langevin dynamics (Equation (12)), we use the posterior means as the decomposed gene expression levels at single-cell resolution. The posterior sampling process is summarized in Algorithm 1.

**Algorithm 1.** Annealed Langevin dynamics for gene expression decomposition

> **Require:** $\{\sigma_l\}_{l=1}^{L}, \{\sigma_{yl}\}_{l=1}^{L}, \eta_0, T, R$.
> Initialize $\mathbf{X}^{(0)} = \mathbf{0}, \mathbf{X}_{\text{sum}} = \mathbf{0}$
> **for** $rep = 1, 2, \ldots, R$ **do**
> **for** $l = L, L-1, \ldots, 1$ **do**
> $\eta = \eta_0 \cdot \sigma_l^2 / \sigma_1^2$.
> **for** $t = 1, 2, \ldots, T$ **do**
> Draw $\boldsymbol{\varepsilon}^{(l, t)} \sim \mathcal{N}(\mathbf{0}, \mathbf{I})$,

$$
\mathbf{X}^{(l, t)} = \mathbf{X}^{(l, t)} + \eta\left[\nabla_{\mathbf{X}_i} \log p\left(\mathbf{y}_i | \mathbf{X}_i^{(l, t)}\right) + \begin{bmatrix} \mathbf{s}_{\boldsymbol{\theta}}\left(\mathbf{x}_{i,1}^{(l, t)}, \sigma_l, \boldsymbol{\mu}_{k_{i,1}}\right) \\ \vdots \\ \mathbf{s}_{\boldsymbol{\theta}}\left(\mathbf{x}_{i, M_i}^{(l, t)}, \sigma_l, \boldsymbol{\mu}_{k_{i, M_i}}\right) \end{bmatrix}\right] + \sqrt{2\eta}\,\boldsymbol{\varepsilon}^{(l, t)},
$$

$$
\text{where } \nabla_{\mathbf{X}_i} \log p(\mathbf{y}_i | \mathbf{X}_i^{(t)}) = \nabla_{\mathbf{X}_i}\left(\frac{1}{2\sigma_{yl}^2} \left| \mathbf{y}_i - f\left(\mathbf{x}_{i,1}^{(t)}, \ldots, \mathbf{x}_{i, M_i}^{(t)}\right)\right|^2\right),
$$

$$
f\left(\mathbf{x}_{i,1}, \ldots, \mathbf{x}_{i, M_i}\right) = \sum_{m=1}^{M_i}\left(\exp\left(\mathbf{x}_{i,m} + \boldsymbol{\mu}_{k_{i,m}}\right) - 1\right),
$$

> **end for**
> $\mathbf{X}^{(0)} = \mathbf{X}^{(T)}$.
> **end for**
> $\mathbf{X}_{\text{sum}} = \mathbf{X}_{\text{sum}} + \mathbf{X}^{(0)}$
> **end for**
> **return** $\mathbf{X}_{\text{sum}} / R$

## SpatialScope for ST data from other platforms

As a unified framework, SpatialScope not only can handle low-resolution ST data with histological images (e.g., 10x Visium), but also can serve as efficient analytical tools for spatial data from other experimental platforms. In this section, we demonstrate that SpatialScope can be applied to perform dropout correction for genes with low-detection rates in Slide-seq data, and imputation for unmeasured genes in MERFISH data or other in-situ hybridization-based ST data.

**Sparse genes dropout correction for Slide-seq data.** As a high-resolution approach, the pixel size of Slide-seq can achieve single cell level (10 μm[83]) but it may still contain the mRNA from multiple cells[14]. Slide-seq data can be highly sparse. About 99.46% entries are zero for Slide-seq V1 data and 98.35% for Slide-seq V2 data, compared to about 90% zero counts for 10x Visium data[64]. The framework of SpatialScope can also be applied to correct dropouts in Slide-seq data and recover transcriptome-wide gene expressions at single-cell resolution.

Because of the high resolution of Slide-seq data and lack of histological images, nucleus segmentation step in dealing with 10x Visium

data is not practicable for Slide-seq data. Although the spot size of Slide-seq (10 μm) already matches the size of a single cell, one spot may contain fractions of several cells due to the technique limitation. To demonstrate this, we estimated the number of cell types per Slide-seq V2 cerebellum spot using the cell type deconvolution results (Supplementary Fig. 38). In total, 22.0% and 1.6% of spots were predicted to contain two and three cell types, respectively, consistent with previous estimates[14]. Simply assuming that there is only one cell in these spots may not be appropriate in this case. Consequently, to enhance the flexibility of our model and mitigate the risk of overfitting with regard to cell number estimation, we replace the first step nucleus segmentation by singlet/doublet classification, which assumes that at most two cell types co-exist within a Slide-seq spot. With this flexible assumption, we are able to yield a more elucidated and comprehensive depiction of tissue structures in real data analysis (Supplementary Fig. 39).

Next, the steps of cell type identification and gene expression decomposition can be applied similarly as those for 10x Visium data. The correction of dropout for genes with low detection rate is achieved in gene expression decomposition based on the same procedure. Let's consider the case where a pixel $i$ contains two cells, i.e., $M_i = 2$. In this case, $\mathbf{y}_i$ is the aggregated gene expression profiles from two cells, in which the expression levels of some genes are nearly zero. By the same modeling principle as that in Equation (10), we can assume that

$$\mathbf{y}_i | \mathbf{x}_{i,1}, \mathbf{x}_{i,2} \sim \mathcal{N}\left(f\left(\mathbf{x}_{i,1}, \mathbf{x}_{i,2}\right), \sigma_y^2 \mathbf{I}\right),$$

where $f\left(\mathbf{x}_{i,1}, \mathbf{x}_{i,2}\right) = \sum_{m=1}^{m=2}\left(\exp\left(\mathbf{x}_{i,m} + \boldsymbol{\mu}_{k_{i,m}}\right) - 1\right)$. Because SpatialScope first learns the distribution of gene expressions from the single-cell reference data, it can output the posterior means of $\mathbf{x}_{i,1}$ and $\mathbf{x}_{i,2}$ by running Algorithm 1. We then use the posterior means of $x_{i,1}$ and $x_{i,2}$ as the de-noised data, where the dropouts are corrected.

**Imputation for in-situ hybridization based ST data.** In-situ hybridization-based ST data can provide localizations of gene expressions at the cellular level, resulting in single-cell resolution spatial transcriptomics. However, because of the limitation of the indexing scheme[11, 58, 84], the detected spatial transcriptomics by in-situ hybridization methods tend to have limited throughput in the number of genes (e.g., tens to hundreds of genes captured by MERFISH[58]). Therefore, researchers begin to integrate in-situ hybridization-based ST data with single-cell reference data to impute the unmeasured genes, providing more complete spatial transcriptome information and cellular structures[18–20]. By learning the distribution of gene expressions from the single-cell reference data using a score-based generative model, SpatialScope can achieve accurate gene imputation as follows.

Suppose that the expression levels of $G$ genes and $G_0$ genes are measured in the single-cell reference and ST data, respectively. We assume that the set of $G_0$ genes measured in ST data is a subset of $G$ genes in the scRNA-seq data. Let $\mathbf{y}_i \in \mathbb{R}^{G_0}$ be the measured gene expression counts in ST data after batch effect correction, and $\mathbf{x}_i^{\text{count}} \in \mathbb{R}^G$ be the true expression at location $i$, respectively. Without loss of generality, we assume that the first $G_0$ genes in $\mathbf{x}_i^{\text{count}}$ are measured. Then we have

$$\mathbf{y}_i = \mathbf{I}_{\text{mask}} \mathbf{x}_i^{\text{count}} + \boldsymbol{\varepsilon},$$

where $\mathbf{I}_{\text{mask}} = \left[\mathbf{I}_{G_0}, \mathbf{0}\right] \in \mathbb{R}^{G_0 \times G}$, $\mathbf{I}_{G_0}$ is the $G_0 \times G_0$ identity matrix, and $\boldsymbol{\varepsilon} \in \mathcal{N}\left(0, \sigma_\varepsilon^2 \mathbf{I}\right)$ is the measurement noise. As the score function is estimated in the log scale, we denote $\mathbf{x}_i = \log(\mathbf{x}_i^{\text{count}} + 1) - \boldsymbol{\mu}_{k_i}$ as the log scale expression, where $\boldsymbol{\mu}_{k_i}$ is the mean expression level of cell type $k_i$. Now we have $\mathbf{y}_i | \mathbf{x}_i \sim \mathcal{N}\left(\mathbf{I}_{\text{mask}}\left(\exp\left(\mathbf{x}_i + \boldsymbol{\mu}_{k_i}\right) - 1\right), \sigma_\varepsilon^2 \mathbf{I}\right)$ and $\mathbf{x}_i \sim p(\mathbf{x}_i | k_i)$. To obtain the imputed expression, we can take samples

from posterior $p(\mathbf{x}_i | \mathbf{y}_i)$ based on the Langevin dynamics,

$$\begin{aligned}
\mathbf{x}_i^{(t+1)} &= \mathbf{x}_i^{(t)} + \eta \nabla_{\mathbf{x}_i} \log p(\mathbf{x}_i | \mathbf{y}_i) + \sqrt{2\eta} \boldsymbol{\varepsilon}^{(t)} \\
&= \mathbf{x}_i^{(t)} + \eta \left[\nabla_{\mathbf{x}_i} \log p\left(\mathbf{y}_i | \mathbf{x}_i^{(t)}\right) + \nabla_{\mathbf{x}_i} \log p\left(\mathbf{x}_i^{(t)} | k_i\right)\right] + \sqrt{2\eta} \boldsymbol{\varepsilon}^{(t)},
\end{aligned} \quad (13)$$

where $\nabla_{\mathbf{x}_i} \log p\left(\mathbf{y}_i | \mathbf{x}_i^{(t)}\right) = \frac{1}{\sigma_\varepsilon^2} \exp\left(\mathbf{x}_i + \boldsymbol{\mu}_{k_i}\right) \odot \left(\mathbf{I}_{\text{mask}}^T (\mathbf{I}_{\text{mask}} (\exp(\mathbf{x}_i + \boldsymbol{\mu}_{k_i}) - 1)))\right)$ and $\odot$ is element-wise product. Using the learned score function $\mathbf{s}_{\boldsymbol{\theta}}(\mathbf{x}, \sigma_l, \boldsymbol{\mu}_k)$ given by Equation (9), we begin with random initialization and then run the Langevin dynamics by progressively reducing noise level $\sigma_l$,

$$\mathbf{x}_i^{(l,t+1)} = \mathbf{x}_i^{(l,t)} + \eta \left[\nabla_{\mathbf{x}_i} \log p\left(\mathbf{y}_i | \mathbf{x}_i^{(l,t)}\right) + \mathbf{s}_{\boldsymbol{\theta}}\left(\mathbf{x}_i^{(l,t)}, \sigma_l, \boldsymbol{\mu}_{k_i}\right)\right] + \sqrt{2\eta} \boldsymbol{\varepsilon}^{(l,t)},$$

$$(14)$$

where the initial point at the later stage is given by the sample from the previous stage, i.e., $\mathbf{x}^{(l,t=1)} = \mathbf{x}^{(l+1, t=T)}$. The obtained samples $\mathbf{X}^{(l,t)}$ at level $l = 1, t = 1, \ldots, T$, will be posterior samples from $p(\mathbf{x}_i | \mathbf{y}_i, k_i)$. By averaging samples of Langevin dynamics in Equation (14), we use the posterior mean as the imputed gene expression.

### Spatial smoothness constraint

To better demonstrate the effectiveness of spatial smoothness constraint imposed by the Potts model, we performed simulations to assess the performance of cell type identification on six benchmarking datasets (Supplementary Fig. 4, Dataset 1-6). We varied the parameter $\nu$ in Equation (4) and compared the results with the baseline methods RCTD and StarDist+RCTD by measuring error rate at single-level or PCC and RMSE at spot-level (Supplementary Note section 2.9.4). Notably, by incorporating spatial information within the $\nu = 10 \sim 50$ range, SpatialScope demonstrated substantial improvement in accurately identifying the cell types at each location (Supplementary Fig. 45 and 49). Therefore, we use the hyperparameter $\nu = 10$ as the default setting. The details of the optimization process are given in Supplementary Methods.

### Comparison between SpatialScope and RCTD

Although Step 2: Cell type identification can be considered an extension of the discrete RCTD model, the primary advantage of our extended discrete model lies in incorporating a spatial smoothness constraint imposed by the Potts model. This constraint enhances the accuracy and robustness of cell type identification by considering the spatial context. Furthermore, Step 3: gene expression decomposition plays a crucial role in obtaining a spatially resolved cellular transcriptomic landscape by integrating ST data and single-cell reference data using deep generative models. These gene expression profiles are the foundation for understanding and exploring the underlying cellular processes and interactions. We can facilitate the interpretation of cellular downstream analyses, such as cell-cell interactions, localization, and spatial trajectories, only after obtaining gene expression profiles at the cellular level. Overall, SpatialScope and RCTD are quite different methods to analyze ST data, and we provide a detailed between SpatialScope and RCTD in terms of method utility, model, algorithm and downstream applications in supplementary (Table S2, Supplementary Note section 2.8).

### Robustness of unbalanced cell types in single-cell reference data and unbalanced cell numbers within spots

In practical applications, it is a common situation that there is a large variation in the proportions of different cell types within single-cell reference data, as well as imbalanced cell numbers within spots. For instance, the number of cells can range from tens to thousands across different cell types. In tissues with high cell density, the cell counts

within spots may range from a few to a dozen. To evaluate the robustness of SpatialScope in handling unbalanced cell types within single-cell reference data, we conducted a comparative analysis of gene expression decomposition performance across different cell types using Dataset 1. Our results demonstrate that SpatialScope exhibits robust decomposition performance despite unbalanced cell types (Supplementary Note section 2.9.9, Supplementary Fig. 60a). We also found that the decomposition accuracy is more related to the heterogeneity within a cell type rather than cell type proportion in the single cell reference data (Supplementary Fig. 60b, c). Subsequently, we quantitatively assessed gene expression decomposition performance by separately evaluating spots with different cell numbers and comparing SpatialScope with Tangram and CytoSPACE (Supplementary Note section 2.9.10, Supplementary Figs. 61, 62). As anticipated, the performance of all compared methods, as measured by cosine similarity, declined with an increasing number of cells. This can be attributed to introducing more components in a spot, which introduces greater uncertainty in the decomposition process. However, it is crucial to highlight that our method, SpatialScope, consistently achieved the highest performance across various scenarios of cell numbers, UMI subsample rates, and whether the reference data was paired or not, in the construction of simulated spots.

## Hyperparameters sensitivity analysis

One of the unique features and strengths of SpatialScope lies in its utilization of a score-based generative model to accurately approximate the distribution of gene expressions from the scRNA-seq reference data. Then SpatialScope ran the Langevin dynamics to perform posterior sampling for gene expression decomposition at each spot. We tested several key hyperparameters in Step 3: Gene expression decomposition, including the hyperparameters $epoch, L, T$, and $\sigma_{yl}$ (Supplementary Note section 2.9.7). We use the score function at 7500 training epoch for all data analysis in the main text. We also investigate the performance of SpatialScope under score-based generative models with different training epochs and recommend that the number of epochs ranges from 5000 to 10,000 due to the trade-off between the performance and the time cost (Supplementary Fig. 56). The parameter $L$ represent the number of noise level (Equation (7)), $T$ is the number of sampling steps per noise scale (Equation (8)), and $\sigma_{yl}$ (Equation (11)) is related to the distribution we assign to the count-scale spot level gene expression profile $y|x_1, x_2, …, x_M$ at each noise level, where $x_m, m = 1, 2, …, M$ represents the true count-scale gene expression levels of cells in the spot, and $M$ is the number of cells in that spot. Intuitively, the more extensive the grid of noise levels $\{\sigma_l\}_{l=1}^{L}$, the better for learning (i.e., the larger $L$, the better). For the sampling step $T$, similarly, the larger $T$, the better. However, the larger $L$ and $T$ mean more expensive computational resources. There is a trade-off between the performance and computational cost. We have determined that SpatialScope is robust to a wide range of parameter settings (Supplementary Figs. 57 and 58). Therefore, we suggested the default setting of SpatialScope as $L = 232, T = 5$ according to the dimension of single-cell gene expression profiles, and we use the default setting $\sigma_{yl} = \sqrt{\sigma_l}$ for all real data analysis.

## Real data analysis

In this study, we evaluated our SpatialScope on five publicly available spatial transcriptomics datasets.

**Visium human heart dataset.** The human heart sample was from BioIVT Asterand and profiled by 10x Visium that measured the whole transcriptome within 55 μm diameter spots. After removing spots that did not map to the tissue region or with total UMI counts less than 100, 3813 spots were left for subsequent analysis. We then focused on an ROI of 331 spots that shows a spatial pattern characterized by vascular cells. Through the matched H&E image, we annotated the main vascular structure in the center of ROI that covers 18 tissue spots. For cell type identification, we used a paired human heart snRNA-seq atlas that consists of 10 major cell types, ranging from widespread cardiomyocytes to less common adipocytes and neuronal cells[40]. Following the standard pre-processing procedure, we normalized total counts per cell with median transcript count, then performed $log(1 + x)$ transformation and selected the top 1000 most highly variable genes and 50 top marker genes for each cell type as training genes. For gene expression decomposition, we first included an additional 876 ligands/receptors provided by Giotto[10] into the training genes for the detection of cellular communication in the downstream analysis. Then we applied the deep generative model to learn the expression distribution of the training genes in the snRNA-seq reference data. Finally, by leveraging the learned single-cell gene expression distribution, we performed the gene expression decomposition for the low-resolution Visium data and generated the single-cell resolved spatial transcriptomics for human heart.

**Visium mouse brain cortex dataset.** The two adjacent sagittal slices of mouse brain anterior tissue were from BioIVT Asterand and profiled by 10x Visium. After removing spots that did not map to the tissue region, 2695 and 2825 spatial spots from the two slices were left for subsequent analysis. We first filtered out spatial locations that have less than 100 total read counts. Then, using the matched H&E-stained histological images, we segmented the cerebral cortex regions, resulting in 812 and 794 cortex spots left in slice 1 and slice 2, respectively. Finally, we used the recently developed tool, PASTE[34], to compute a pairwise slice alignment between these two segmented cortex slices, which allowed us to construct an aligned 3D mouse brain cortex ST data. We used mouse brain cortical scRNA-seq data as a reference[33]. This dataset was collected from the mouse Primary visual (VISp) area using the Smart-seq2 technology and contains 14,249 cells across 23 cell types. Similarly, we first performed total counts normalization and $log(1 + x)$ transformation, and then selected the top 1000 most highly variable genes and 50 top marker genes from each cell type as training genes. In cell type identification, we incorporated the spatial information in 3D space and thus can produce more reliable spatial priors. Next, in the gene expression decomposition task, we included ligands/receptors and decomposed the gene expressions from the spot-level into the single-cell level using the learned gene expression distribution.

**Slide-seq v2 mouse cerebellum dataset.** The mouse cerebellum dataset was profiled by Slide-seq V2 and measured with the whole transcriptome within 10 μm diameter spots[14]. This dataset consists of gene expression measurements for 23,096 genes and 11,626 spatial spots. We filtered out genes that have zero counts across all spots and filtered out spots with total UMI counts less than 100, leading to 20,141 genes and 8952 spots for subsequent analysis. As the paired histological images are not available for Slide-seq data, we replaced the first step, Nucleus segmentation, with Singlet/Doublet classification inspired by RCTD, which assumes that there are at most two cells per spot as the spot size (10 μm) almost matches the single cell size. Overall, we detected 10975 cells, including 6929 spots containing one cell and 2023 spots containing two cells. Following RCTD, we used a paired mouse cerebellum snRNA-seq data as the reference[85]. This dataset contains 24,387 genes and 15,609 cells from 19 cell types. Similarly, we first performed total counts normalization and $log(1 + x)$ transformation, and then selected the top 1000 most highly variable genes and 50 top marker genes from each cell type as training genes in the cell type identification task. Finally, we generated the corrected high-throughput single-cell resolution Slide-seq data by leveraging the gene expression distribution learned from the snRNA-seq reference.

**MERFISH MOp dataset.** The mouse brain MOp dataset was profiled by the image-based ST approach, MERFISH, with single-cell resolution. This dataset comprised of 254 genes and about 300,000 single cells located in 64 mouse brain MOp slices from 12 different samples[58]. As a concrete example demonstrated in MERFISH paper, we used the slice, mouse1_slice180, from mouse1_sample4 to evaluate the imputation performance of SpatialScope and the compared methods. We used a paired droplet-based snRNA-seq profiles collected from mouse MOp as the reference, which measures the expression of 26,431 cells across 20 cell types[18]. Following the standard pre-processing procedure, we normalized total counts per cell with median transcript count and performed and $log(1 + x)$ transformation. Using the 252 genes that overlapped with snRNA-seq reference data as training genes, we first identified the cell type label for each cell in the MERFISH dataset. Then we applied the deep generative model to learn the distribution of gene expressions in the snRNA-seq reference data. Finally, using the learned high-throughput gene expressions distribution, we imputed the gene expressions of unmeasured genes in MERFISH dataset by conditioning on the observed expressions of 252 overlapped genes.

## Downstream analysis

**Cell-cell interactions.** Although ST is believed to be the best suited technology to elucidate cellular/molecular interactions[52], current ST dataset is still limited by either low resolution or low capture rate. Fortunately, the efficient in silico generation of single-cell resolution and high throughput spatially resolved transcriptomics by SpatialScope perfectly solved this issue. As a recently developed tool for detecting cellular communications mediated by ligand-receptor interactions, Giotto was applied on SpatialScope outputs following the protocol (https://rubd.github.io/Giotto_site/articles/tut14_giotto_signaling.html). Specifically, Giotto first ran pre-processing to remove low-quality genes/cells and create a spatial network connecting single cells using Delaunay triangulation, then ran 'spatCellCellcom' function to analyze the ligand-receptor signaling with spatial_network_name parameters being Delaunay_network. Finally, we selected top and reliable ligand-receptor signaling with the following threshold: p.adj < 0.25, abs(log2fc) > 0.1, lig_nr > 10, rec_nr > 10, lig_expr > 0.5 & rec_expr > 0.5. For raw Slide-seq data, we detected cell-cell interactions by simply assuming each spot is a single cell whose cell type is determined by the majority proportion. Then the following ligand-receptor signaling analysis by Giotto is identical.

**Cell-type specific spatially DE genes.** We ran C-SIDE to detect cell-type specific spatially DE genes on the MERFISH dataset using function: run.CSIDE.nonparametric. We followed the guidelines (https://raw.githack.com/dmcable/spacexr/master/vignettes/merfish_nonparametric.html) with parameters: gene_threshold = .001, cell_type_threshold = 10, and fdr = 0.2.

**Spatially DE genes.** Compared to C-SIDE, SPARK-X was developed to consider genes that exhibit spatially DE patterns in a global perspective instead of restricting to specific cell types. We applied SPARK-X to MERFISH dataset using the default parameters following the instruction (https://xzhoulab.github.io/SPARK/03_experiments/). As suggested by SPARK-X, we treated cell type labels as covariates to exclude the spatially DE genes explained by spatial distribution of cell types.

## Compared methods

For cell type identification task, we compared SpatialScope with three single-cell alignment (Tangram, CytoSPACE and StarDist+RCTD) and ten deconvolution (SpatialDWLS, RCTD, Cell2location, CARD, SpaOTsc, novoSpaRc, DestVI, STRIDE, SPOTlight, and DSTG) methods.

**Tangram.** We followed the instructions of Tangram: https://tangram-sc.readthedocs.io/en/latest. In order to constrain the number of

mapped single cell profiles, we set the mode parameters as constrained, target_count=the total number of segmented cells, and density_prior=fraction of cells per spot.

**CytoSPACE.** We followed the guidelines on GitHub repository: https://github.com/digitalcytometry/cytospace. We first used Seurat to obtain an overall cell type composition across all spatial spots, then the estimated fractional composition of each cell type was used as input for alignment.

**StarDist+RCTD.** We proposed a discrete version of RCTD model, StarDist+RCTD, as a baseline method in the comparison. StarDist+RCTD first uses StarDist to detect the cell number in each spot, the same as SpatialScope. Then using the information from StarDist, it directly discretizes the cell type proportion produced by RCTD to get the distribution of single-cell cell type label (see Supplementary Note section 2.9.3).

**SpatialDWLS.** We followed the instructions on the SpatialDWLS website: https://rubd.github.io/Giotto_site/articles/tut7_giotto_enrichment.html. We set the parameter as n_cell = 20.

**RCTD.** We followed the guidelines on the RCTD GitHub repository: https://github.com/dmcable/spacexr. We set the doublet_mode parameter being full.

**Cell2location.** We followed the guidelines on the Cell2location Github repository: https://github.com/BayraktarLab/cell2location. The single-cell regression model was trained with parameters max_epochs = 250, batch_size=2500, and lr = 0.002. The cell2location model was trained with parameters max_epochs = 30,000.

**CARD.** We followed the guidelines and used the recommended default parameter setting on the CARD GitHub repository: https://github.com/YingMa0107/CARD.

**SpaOTsc.** We followed the guidelines on the SpaOTsc GitHub repository: https://github.com/zcang/SpaOTsc. We set alpha=0, rho=1.0, epsilon=1.0.

**novoSpaRc.** We followed the guidelines and used the recommended default parameter setting on the novoSpaRc GitHub repository: https://github.com/rajewsky-lab/novosparc.

**DestVI.** We followed the guidelines on the DestVI GitHub repository: https://github.com/scverse/scvi-tools. We set max_epochs=250, lr=0.0001.

**STRIDE.** We followed the guidelines on the STRIDE GitHub repository: https://github.com/DongqingSun96/STRIDE. We set –gene-use=All, –st-scale-factor=300, –sc-scale-factor=300.

**SPOTlight.** We followed the guidelines and used the recommended default parameter setting on the SPOTlight GitHub repository: https://github.com/MarcElosua/SPOTlight.

**DSTG.** We followed the guidelines and used the recommended default parameter setting on the DSTG GitHub repository: https://github.com/Su-informatics-lab/DSTG.

For gene expression decomposition task, we only compared SpatialScope with the three single-cell alignment methods.

**Tangram.** According to the instructions, Tangram only provides a prediction of spot-level gene expression using the mapped single cell profiles through the function project_genes. To make it comparable

with our SpatialScope in the task of gene expression decomposition, we provided a script that takes the Tangram output as input and generates the single-cell resolution spatial transcriptomics data. Specifically, with the single cells to spatial spots mapping matrix output by Tangram, we first obtained the most probable spot that each single cell belongs to. Then we removed the noise cells with mapping probability less than 0.5, and grouped the remaining cells by spot ids. Finally, for each spot, we regarded the grouped cells are mapped single cells from scRNA-seq reference and used their gene expressions as the decomposed single-cell level gene expression profiles.

**CytoSPACE**. CytoSPACE provides the mapped single cell ids from scRNA-seq reference for each spatial spot, we therefore directly used the mapped single cell's gene expressions as the decomposed gene expression profiles.

**StarDist+RCTD**. RCTD uses the mean expression level of each cell type for cell type deconvolution. Therefore, we used the mean gene expression corresponding to the identified cell type as the decomposed single-cell expression for each individual cell. For example, if a cell is identified as cell type A by StarDist+RCTD, then we use the mean expression level of cell type A in the scRNA-seq reference as the decomposed single-cell expression for this cell.

For gene expression imputation task, we compared SpatialScope with Tangram, gimVI, SpaGE, SpaOTsc, novoSpaRc, stPlus and Seurat.

**Tangram**. We followed the instructions of Tangram: https://tangram-sc.readthedocs.io/en/latest. We used the function project_genes to generate the new spatial data with the whole transcriptome using the mapped single cell.

**gimVI**. We followed the guidelines on the gimVI website: https://docs.scvi-tools.org/en/0.8.0/user_guide/notebooks/gimvi_tutorial.html. We used the model.get_imputed_values function with parameter normalized = False to impute the unmeasured gene expressions.

**SpaGE**. We followed the instructions on the GitHub repository of SpaGE: https://github.com/tabdelaal/SpaGE. We set the parameter n_pv = $N_{gene}$/2 if the number of genes used for integration was greater than 50.

**SpaOTsc**. We followed the guidelines on the SpaOTsc GitHub repository: https://github.com/zcang/SpaOTsc. We used the gamma_mapping mapping matrix to multiply the scRNA-seq gene expression matrix to obtain the imputed unmeasured gene expressions.

**novoSpaRc**. We followed the guidelines and used the recommended default parameter setting on the novoSpaRc GitHub repository: https://github.com/rajewsky-lab/novosparc.

**stPlus**. We followed the guidelines and used the recommended default parameter setting on the stPlus GitHub repository: https://github.com/xy-chen16/stPlus.

**Seurat**. We followed the instructions of Seurat: https://satijalab.org/seurat/articles/get_started.html. We used the function, TransferData, to generate the new spatial data with the whole transcriptome using thescRNA-seq as reference.

## Statistics and reproducibility
R (version 4.1.1) and Python 3.9 were used for all statistical analyses. No statistical method was used to predetermine sample size, SpatialScope was evaluated across four publicly available spatially resolved transcriptomics datasets in real data applications using as many samples as possible in these datasets, including human heart (spot sample size = 3813) and mouse brain cortex data (spot sample size = 1606) from 10x Visium datasets, mouse cerebellum data from Slide-seq dataset (spot sample size = 8952), mouse MOp data from MERFISH dataset (spot sample size = 5551). Following standard quality control practice, we retained genes with non-zero expression level on at least 10 spots and retained spots with non-zero expression for at least 50 genes for analysis, in order to avoid false positives. All data are publicly available and we do not perform any randomized controlled trial, so randomization and blinding are not relevant to this study.

## Reporting summary
Further information on research design is available in the Nature Portfolio Reporting Summary linked to this article.

## Data availability
For the benchmarking datasets, the MERFISH MOp data were downloaded from the brain image library (https://doi.org/10.35077/g.8), the MERFISH Mouse brain data were downloaded from the project page (https://cellxgene.cziscience.com/collections/31937775-0602-4e52-a799-b6acdd2bac2e), the STARmap PLUS Hippocampus data were downloaded from the single cell portal project (https://singlecell.broadinstitute.org/single_cell/study/SCP1375). For real data analysis, the 10x human heart and mouse brain cortex datasets were downloaded from the 10x official website (https://www.10xgenomics.com/resources/datasets), and the paired human heart and mouse brain cortex scRNA-seq reference are available from the project page (https://www.heartcellatlas.org/v1.html) and (https://celltypes.brain-map.org/rnaseq/mouse/v1-alm), respectively. Both Mouse cerebellum Slide-seq V2 dataset and the paired scRNA-seq reference were downloaded from the single cell portal project (https://singlecell.broadinstitute.org/single_cell/study/SCP948). Source data are provided with this paper.

## Code availability
The SpatialScope software package and source code are available in Github (https://github.com/YangLabHKUST/SpatialScope)[86]. We also uploaded all scripts and materials to reproduce all the analyses at the same website.

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

## Acknowledgements

We acknowledge the following grants: Hong Kong Research Grant Council grants nos. 16301419, 16308120, 16307221 and 16307322, Hong Kong University of Science and Technology Startup Grants R9405 and Z0428 from the Big Data Institute, Guangdong-Hong Kong-Macao Joint Laboratory grant no. 2020B1212030001 and the RGC Collaborative Research Fund grant no. C6021-19EF to C.Y.; Shenzhen Science and Technology Program JCYJ20220818103001002), and the Guangdong Provincial Key Laboratory of Big Data Computing, The Chinese University of Hong Kong, Shenzhen to Xiang W.; Shenzhen Research Institute of Big Data Internal Project J00220230008 to J.X.; Chinese University of Hong Kong startup grant (4930181), the Chinese University of Hong Kong Science Faculty's Collaborative Research Impact Matching Scheme (CRIMS 4620033), and Hong Kong Research Grant Council (24301419, 14301120) to Z.L.; Hong Kong Research Grant Council grant no. 16209820, the Innovation and Technology Commission (ITCPD/17-9), Lo Ka Chung Foundation through the Hong Kong Epigenomics Project, Chau Hoi Shuen Foundation, the SpatioTemporal Omics Consortium (STOC) and the STOmics Grant Program to A.R.W.

## Author contributions

Xiaomeng W. and J.X. conceived and designed the study. Xiaomeng W., J.X., Z.L. and C.Y. developed the algorithm of SpatialScope. S.S.T.T interpreted the results with support from A.R.W.; Xiaomeng W., J.X., Z.L., A.R.W. and C.Y wrote the manuscript. M.C., R.S., Y.W., Xiang W. provided critical feedback during the study and helped revise the manuscript.

## Competing interests

The authors declare no competing interests.
