## [Peer Review File · Nature Communications]

Integrating spatial and single-cell transcriptomics data using deep generative models with SpatialScopeReviewer #1 (Remarks to the Author):

The authors developed a probabilistic approach to integrating single-cell data and spatial data, named as SpatialScope. The authors claim that the model can accurately decompose spot-level data into single-cell expression data. The model also has the imputation capability to predict the expression of genes not in the panel. The algorithm is innovative, and the comparison with existing methods shows superior performance in simulated and actual biological data. The manuscript is well written, and the methodology is described clearly with sufficient details.

Major comments:

- As the cell-type decomposition methods implemented in SpatialScope and RCTD are similar (except for continuous vs discrete models), more comparison between these two methods is recommended. From Figure S3, it looks like they perform similarly well, and RCTD looks like performing slightly better.
- The model performance is expected to change depending on gene groups. Less abundant and more variable genes would likely be more challenging to predict. Could the authors add some assessment of the performance for low, medium and high abundant/variable genes? For imputation, how would the existing panels affect the imputation of genes not in the panels? For example, how well can the model predict genes in non-brain tissue if a panel is designed for brain markers?
- Similarly, the performance would change for rare cell types and tissue regions with dense nuclei. How would these two factors affect the model performance?

Minor suggestions

- Abstract: revise "remain unable to provide spatial characterization at transcriptome-wide single-cell resolution" as there are methods able to do so like STomics and Slide-seq; similarly, in the Introduction section, the author should update that seqFISH and MERFISH can both measure more than 10,000 genes
- Figure S3 legend: 3 not 4 datasets; GT = Ground Truth

Reviewer #2 (Remarks to the Author):

The authors presented a deep-learning based method, named SpatialScope, to integrate scRNA-seq reference data and spatial transcriptomics (ST) data, which provides a variety of different functions and is suitable for a variety of spatial transcriptome data analysis tasks. However, some concerns have to be addressed before publication: 1) it lacks a careful comparison with other methods in the field, and 2) the model lacks novelty.

Below are the detailed review comments:

Comments on results

- 1) Lack of more detailed performance comparison. In the review by Qu et al.[1], a total of eight integration methods were evaluated, including Tangram, gimVI, SpaGE, Seurat, SpaOTsc, novoSpaRc, LIGER and stPlus, for predicting the spatial distribution of undetected RNA transcripts in spatial transcriptomics datasets. In addition to predict the spatial distribution of RNA transcripts, Tangram, Seurat, SpaOTsc, and novoSpaRc can also assign cells from scRNA-seq data to spatial locations in tissue sections. Additionally, Cell2location, SpatialDWLS, RCTD, Stereoscope, DestVI, STRIDE, SPOTlight, and DSTG can be used to predict the cell type composition of spots in tissue sections by combining spatial transcriptomics data and scRNA-seq data. As mentioned above, it would be great if the authors could add the missing comparison of state-of-art methods, including SpaOTsc, novoSpaRc, DestVI, STRIDE, SPOTlight and DSTG. There is also an additional review available for reference[2].
- 2) The authors performed benchmark tests on multiple datasets, and the error rate in the main Fig. 2C is around 20%, these results are positive. But in b-d of Fig. S3, the error rate is as high as more than 50%, leading to this result may be because the ground truth may be inaccurate. I suggest that the authors can replace the dataset with better quality for benchmarking.
- 3) The authors describe that SpatialScope has the ability to integrate multiple slices. The authors

used PASTE to obtain a 3D aligned ST data for cell type identification in SpatialScope. Lines 19-21 on page 10 "Alignment-based and deconvolution...". I think Tangram and CytoSPACE are also applicable ST data for this 3D alignment. At the same time, the comparison of single and double slices lacks fairness.

4) The model lacks a systematic hyperparameter comparative analysis. The authors provide a comparison of hyperparameter epochs (Fig. S7). But there are still many hyperparameter settings that will affect the performance of the model, such as LT and g_ss in the Train_scRef.py script. It would be great if the authors could add these hyperparameter comparisons. Of course, the hyperparameter evaluation of the system can also test the SpatialScope's stability and robustness.

**Comments on methodological innovation

1) The first step of the model is nuclei segmentation on hematoxylin and eosin (H&E)-stained histological images to count the number of cells at each spot. As shown in lines 21 to 24 on page 3, this step is also an important step different from the method of analyzing the composition of cells in the point (such as Cell2location, SpatialDWLS, RCTD, etc.), but this step only uses the built-in function of squidpy (in demo Nuclei_Segmentation.py), and lacks innovations. This is more like building block stacking for SpatialScope. Although the authors used StarDist and Cellpose to compare the cell nucleus segmentation, and finally decided to use StarDist (Fig. S20). At present, many algorithms for cell nucleus segmentation have been developed, such as Baysor[3] and DeepCell[4], where Cellpose is upgraded to version 2.0[5]. Does the author need to compare with these methods? Since it is an important step, is this overall embedding into SpatialScope considered an algorithmic innovation?

2) Lines 15-18 on page 32, "Because of the high...". Consistent with the first comment, using RCTD for nuclei segmentation when there is no histological image does not seem to be an algorithmic innovation either. It would be great if the author could further explain the innovation of the SpatialScope's model.

3) Also in the step of cell type identification, SpatialScope seems to be similar to the method of RCTD. On page 26, the author describes "Our model differs from RCTD in... spot i.", does SpatialScope only use the average expression of cell types for replacement, and the rest are the same as RCTD? In other words, will StarDist+RCTD get similar results to the article?

**Comments on writing

1) On page 21, lines 18-22, "As shown in Fig. 6a... (Fig. 6a)". "Fig. 6a" is quoted repeatedly.

2) Fig. S9c, Fig. S19b, the paper is not cited.

3) Line 21 on page 7 and line 18 on page 9, colons should be changed to full stops.

4) "As some cells..." on page 12 is missing references to figures. Although detailed in this section "supplementary note section 2.7.3", it is better to add supplementary figures.

5) Lines 2-13 on page 18 describe the RTN4-LINGO intercellular interaction in detail. It is recommended that you display this part of the results in Fig.4.

**Comments on implementation and example codes

I downloaded, installed and ran the sample code. But it didn't quite work and need to be improved.

1) Python package should use standard python packaging facilities. It would be preferable if setuptools or poetry or similar would be used.

2) The list of required dependencies in the readme is incomplete. In addition to what is described in the README, the following package must be installed: scikit-misc.

3) There is a slight omission in the description of the sample code. The link to the original dataset in the demo Human-Heart.ipynb does not exist.

4) An error is reported when running the sample code. As a result, errors were reported in the "Learning the gene expression distribution of scRNA-seq reference using score-based model" step and "Step3: Gene expression decomposition", and some results of the article could not be reproduced. The error code is as follows:

```
``
```

```
from utils import configure_logging, ConfigWrapper
ImportError: cannot import name 'ConfigWrapper' from 'utils'
```

```
``
```

**References

1. Li, B., et al., Benchmarking spatial and single-cell transcriptomics integration methods for transcript distribution prediction and cell type deconvolution. *Nature methods*, 2022. 19(6): p. 662-670.
2. Li, H., et al., A comprehensive benchmarking with practical guidelines for cellular deconvolution of spatial transcriptomics. *Nature Communications*, 2023. 14(1): p. 1548.
3. Petukhov, V., et al., Cell segmentation in imaging-based spatial transcriptomics. *Nature biotechnology*, 2022. 40(3): p. 345-354.
4. Bannon, D., et al., DeepCell Kiosk: scaling deep learning-enabled cellular image analysis with Kubernetes. *Nature methods*, 2021. 18(1): p. 43-45.
5. Pachitariu, M. and C. Stringer, Cellpose 2.0: how to train your own model. *Nature Methods*, 2022: p. 1-8.

Reviewer #3 (Remarks to the Author):

The paper with title of "SpatialScope: A unified approach for integrating spatial and single-cell transcriptomics data using deep generative models" presents a unified approach to integrating scRNA-seq reference data and ST data using generative models. In the model, the authors try to achieve single cell resolution on ST datasets, which can facilitate downstream analysis including cell communication, ligand-receptor interactions, spatially DE gene analysis, etc. However, despite the multiple applications in the paper, I believe this work needs to address the following questions.

Major:

- 1, Recently, there are more and more super high resolution ST technologies, for example, Stereo-seq [PMID: 35512705] can provide spot with diameter of 220 nm, which is much smaller than the size of single cell. Is it still necessary to focus on deconvolution for ST?
- 2, SpatialScope assumes that neighboring cells are more likely to belong to the same cell type, which is not always true [for example, <https://doi.org/10.1038/s43588-022-00266-5>] and should be discussed carefully.
- 3, In Fig. S3, SpatialScope can not beat RCTD method. In addition, the core cell type identification part, spatialScope also follows modelling way very similar to RCTD. The improvement of modeling, algorithm and performance over RCTD should be discussed comprehensively.
- 4, Nucleus segmentation is the very beginning of SpatialScope and is of vital importance because it provides cell location and number within one spot. More analysis is required for the accuracy and performance of StarDist.
- 5, Fig. 3a assumes that the spot only contains two cells (seems not quite reasonable), and Fig. S27 shows result of spot with up to 5 cells. I am wondering if there is any analysis on how many cells are contained within one spot, which is also relevant to my question #4.
- 6, How to determine the exact position of each deconvoluted cells within one spot? Subspot resolution methods, such as BayesSpace (PMID: 34083791) should be compared with.
- 7, What if ST data lacks histological images? The assumption that at most two cell types co-exist within a spot is too strong now. More data or analysis should be provided to support it.

minor:

- 1, It would be better to move the complex equations (1) (2) to Methods part.

From: Can Yang

Department of Mathematics
Hong Kong University of Science and Technology
Hong Kong

September 6, 2023

To: Reviewers,

RE: The revision of “SpatialScope: A unified approach for integrating spatial and single-cell transcriptomics data using deep generative models” (NCOMMS-23-12222)

Dear Reviewers,

We would like to thank you for the very detailed and constructive comments, which have greatly helped us to improve our manuscript. We have revised the manuscript to address all the comments. The major updates of our paper are summarized as follows:

- 1) As several reviewers asked the relationship between SpatialScope and RCTD, we first provide a section titled “**Comparison between SpatialScope and RCTD**” to explain the major difference between them. In this section, we highlight the advancements of SpatialScope over RCTD in terms of method utility, model, algorithm and downstream applications, as summarized in Table 1. We have also provided details in the point-to-point responses.
- 2) To provide a better comparison between SpatialScope and other related methods, we re-designed our benchmarking study using four single-slice spatial transcriptomics (ST) datasets and two multiple-slice ST datasets (see section “**Benchmarking datasets**” and Figure R1). We systematically compared SpatialScope with those methods for different tasks, including cell type identification at the single-cell level, cell type proportion estimation at the spot level, gene expression decomposition at single-cell resolution, and imputation of gene expression.
- 3) We believe that we have addressed the reviewers’ questions with supported evidence. We have taken significant steps to enhance the clarity and coherence of our method, ensuring that any confusion presented in the previous version has been eliminated.

Regarding reproducibility, we have provided relevant codes and results for the experiments in the SpatialScope website: <https://github.com/YangLabHKUST/SpatialScope>. We have corrected bugs in the software, and provided a more detailed installation instruction there. The updated parts of the main text and the Supplementary Information are marked in blue. We look forward to

receiving further feedback on our revisions.

Sincerely yours,

Can Yang and co-authors

Response to questions shared by all reviewers

We have observed that all reviewers express interest in a comprehensive discussion regarding the advancements of SpatialScope over RCTD. In light of this, prior to addressing the reviewers’ comments individually, we have added a dedicated section that provides an overview of the comparison between SpatialScope and RCTD. Additionally, we have included a section titled “Benchmarking datasets” to introduce the simulation datasets and experimental settings utilized in our study.

Comparison between SpatialScope and RCTD

Before we response to reviewers in a point-to-point manner, we would like to highlight advancements of SpatialScope over RCTD in terms of method utility, model, algorithm and downstream applications.

From the method utility standpoint, RCTD is a powerful method for cell-type deconvolution in spatial transcriptomics (ST) data analysis. It inputs low-resolution ST data (e.g., 10X Visium) and single-cell reference data, and outputs cell-type proportions at each spatial spot. It only offers the average gene expression pattern of a given cell type. In contrast, SpatialScope offers several advantages beyond low-resolution ST data analysis. It can infer (i) the number of cells, (ii) their corresponding cell types, and (iii) the gene expression of individual cells at each spot, specifically for low-resolution ST data. Beyond low-resolution ST data analysis, SpatialScope can also impute gene expression for image-based high-resolution ST data (e.g., MERFISH data). In summary, **SpatialScope can provide transcriptome-wide expression levels at single-cell resolution, whereas RCTD only offers cell type proportions at the spot level.** The generation of single-cell resolution ST data by SpatialScope not only facilitates the visualization of fine-grained cell gradients but also enables the detection of spatially resolved cellular communication. This enables the discovery of meaningful biological processes and intercellular dynamics in space.

From a model design perspective, **the main novelty of SpatialScope lies in its gene expression decomposition step**, which represents the third step of the method. In this step, SpatialScope leverages single-cell reference data to learn the expression patterns of different cell types, which serve as the prior distribution. By combining this prior information with the likelihood term of the observed ST data, SpatialScope formulates a posterior sampling process using Langevin dynamics to perform gene expression decomposition. This unique approach in model design allows SpatialScope to infer transcriptome-wide expression levels at single-cell resolution. In contrast, RCTD does not incorporate a specific model design for gene expression decomposition, distinguishing it from SpatialScope’s innovative approach.

To achieve the desired performance of gene expression decomposition, a major challenge comes from learning the prior distribution of different cell types from single-cell reference data. While it has been very successful in learning a score function with natural images, it is highly non-trivial to learn the conditional score function with single-cell datasets. Let $\mathbf{s}_\theta(\mathbf{x}^{(l)}, \sigma_l, k)$ be the conditional score function, where $\mathbf{x}^{(l)} \in \mathbb{R}^G$ is the log-scale expression levels of G genes at the l -th noise level σ_l , and k is the cell type label. In our experiment, we find that the learning process often tends to largely ignore the cell type information because the neural network naturally focuses on the vector $x^{(l)} \in \mathbb{R}^G$ rather than the scalar k . To successfully incorporate cell type information, we embed cell

type information in a vector whose dimension is comparable to $\mathbf{x}^{(l)}$. Therefore, we propose to learn the score function $\mathbf{s}_\theta(\mathbf{x}^{(l)}, \sigma_l, \boldsymbol{\mu}_k)$ which takes the mean expression level of cell type k as input. The benefits are twofold. First, $\boldsymbol{\mu}_k$ provides precise information about cell type k . Second, $\boldsymbol{\mu}_k \in \mathbb{R}^G$ has the same dimension as $\mathbf{x}^{(l)}$ so it will not be ignored. With this key idea, we have designed a **novel network architecture to learn the conditional score function $\mathbf{s}_\theta(\mathbf{x}^{(l)}, \sigma_l, \boldsymbol{\mu}_k)$** . This is the contribution of SpatialScope from the perspective of architecture and algorithm design. We leave more details in the point-to-point response.

As SpatialScope provides transcriptome-wide expression levels at single-cell resolution, the downstream analysis enables **detection of cellular communication by identifying ligand-receptor interactions from seq-based ST data**, which is NOT supported by the RCTD results. For image-based ST data (e.g., MERFISH data), SpatialScope accurately **imputes expression levels of unmeasured genes and further allows identification of more spatially differentially expressed genes**, gaining biological insights from the downstream analysis.

Overall, SpatialScope offers advancements over RCTD as highlighted above. We also use Table 1 to summarize these key points and add more details in the point-to-point discussion. We have incorporated this part to section 2.8 of the revised supplementary text.

	▼ RCTD	▼ SpatialScope
Method Utility	 ▼ Seq-based data only  ▶ Cell type proportions at each spot ▶ Average gene expression levels of cell types 	 ▼ Seq-based data (e.g., 10 X Visium)  ▶ The number of Cells at each spot ▶ Cell type labels of individual cells ▶ Expression levels of individual cell ▼ Image-based ST data (e.g., MERFISH)  ▶ Imputation of gene expression In summary, SpatialScope provides transcriptome-wide expression levels at single-cell resolution.
Model design	Poisson model with single-cell reference data	 • Deep Generative model (prior distribution learning with reference data) • Langevin dynamics (posterior sampling) with observed spatial transcriptomics data
Architecture and algorithm	 • Methods of moments • Likelihood-based method for parameter estimation 	A novel deep neural network structure to learn the conditional score function from single-cell reference data
Downstream analysis	Spatially differentially expressed (DE) genes for seq-based ST data	 ▶ Detection of cellular communication by identifying ligand-receptor interactions from seq-based ST data ▶ Spatial DE for seq-based ST data ▶ Identification of more spatial DE genes for image-based ST data

Table 1: Comparison between SpatialScope and RCTD.

Benchmarking datasets

To better compare SpatialScope and other related methods, we redesigned our benchmarking study. Following the idea of the benchmarking paper [1], we utilized real MERFISH and STARmap datasets to generate simulation datasets, leveraging their high single-cell resolution capabilities. As the cell type labels and gene expression levels are known at the single-cell level, it is straightforward for us to utilize this information to establish the ground truth for evaluation. Our simulated datasets (Figure R1) in the benchmarking study consist of four single-slice datasets (Dataset 1- Dataset 4) and two multiple-slice datasets (Datasets 5 and 6), as detailed below.

Dataset 1: MERFISH MOp

The MERFISH MOp dataset consists of 254 genes and approximately 300,000 single cells obtained from 64 mouse brain MOp slices belonging to 12 different samples [2]. For our study, we focused on the “mouse1_slice180” from the “mouse1_sample4” and used it to construct a simulation dataset (Figure R1a). The selected slice, “mouse1_slice180”, contains 5,551 cells and exhibits a horizontally structured multi-layer pattern. To obtain the single-cell reference data, we vertically partitioned this dataset into two parts. The right part, comprising approximately 4,000 cells, served as the paired single-cell reference data. The left part, containing around 1,000 cells, was used to generate low-resolution ST data by aggregating the cells on uniform grids, creating simulated spots. We generated simulated spots with a grid size of $34 \times 30 \mu\text{m}$, resulting in 1-5 cells within each simulated spot. To study the robustness of the compared methods to data quality, we downsampled unique molecular identifier counts (UMIs), specifically using values of 130, 260, and 520, which corresponded to 0.5, 1, and 2 cell UMIs in the raw MERFISH data of this slice.

Dataset 2: MERFISH Mouse brain section 1

The MERFISH Mouse brain section 1 dataset was obtained from the mouse frontal cortex and striatum regions provided by the Allen dataset [3]. This dataset consisted of approximately 0.3 million single cells derived from multiple slices of juvenile and old mice. For our study, we selected tissue slice 0 from donor ID 12 as the MERFISH Mouse brain section 1 dataset. This particular dataset comprised expression values of 374 genes across 17,462 single cells after preprocessing. To simulate ST data and single-cell reference data, we divided these cells into two groups. The first subset consisted of a cropped region (containing 3,489 cells) which was utilized to generate pseudo-spots by aggregating cells within each grid. The remaining cells served as the paired scRNA-seq reference. Using the same simulation pipeline, we generated simulated spots with a grid size of $32 \times 32 \mu\text{m}$, resulting in 1-6 cells within each simulated spot (Fig. R1b). In addition, we introduced variability by subsampling UMIs, specifically using values of 53, 107, and 214, which corresponded to 0.5, 1, and 2 cell UMIs in the raw MERFISH frontal cortex and striatum data of this slice.

Dataset 3: MERFISH Mouse brain section 2

The MERFISH Mouse brain section 2 dataset was obtained from the mouse frontal cortex and striatum regions provided by the Allen dataset[3]. This dataset consisted of approximately 0.3 million single cells obtained from multiple slices of juvenile and old mice. For our study, we utilized tissue slice 1 from donor ID 8 as the MERFISH Mouse brain section 2 dataset, which contained expression values of 374 genes across 12,133 single cells after preprocessing. The right bottom region (cell number=1,768) was cropped to make pseudo-spots by aggregating the cells within each grid.

The cells that serve as scRNA-seq reference in Dataset 2 will be regarded as unpaired scRNA-seq reference. Similarly, we applied the same simulation pipeline to generate simulated spots with grid size equal $31 \times 34 \mu\text{m}$, leading to 1-5 cells within the simulated spots (Fig. R1c). We varied the subsampled UMIs as 63, 127 and 255, corresponding to 0.5, 1, and 2 cell UMIs in the raw MERFISH frontal cortex and striatum data of this slice.

Dataset 4: MERFISH Mouse brain section 3

The MERFISH Mouse brain section 3 dataset was obtained from the mouse frontal cortex and striatum regions from the Allen dataset [3]. This dataset imaged about 0.3 million single cells from multiple slices of juvenile and old mice. We used the tissue slice 1 from donor ID 12 as the MERFISH Mouse brain section 3 dataset, which contains expression values of 374 genes on 15,675 single cells after preprocessing. The top middle region (cell number=2,829) was cropped to make pseudo-spots by aggregating the cells within each grid. The cells that serve as scRNA-seq references in Dataset 2 will be regarded as unpaired scRNA-seq reference. Similarly, we applied the same simulation pipeline to generate simulated spots with grid size equal $34 \times 34 \mu\text{m}$, leading to 1-7 cells within the simulated spots (Fig. R1d). We varied the subsampled UMIs as 53, 106 and 212, corresponding to 0.5, 1, and 2 cell UMIs in the raw MERFISH frontal cortex and striatum data of this slice.

Dataset 5: STARmap PLUS Hippocampus 3D

The STARmap PLUS Hippocampus 3D dataset was obtained from the mouse cortical and hippocampal regions as provided by Zeng [4]. This dataset encompassed approximately 2,766 genes observed in 72,165 single cells across eight slices from both TauPS2APP and control mice at 8 and 13 months of age. We selected two slices from the control group that exhibited similar cell type distribution patterns and used the recently developed tool, PASTE [5], to compute a pairwise slice alignment between these two slices, which allowed us to construct an aligned 3D ST data (Fig. R1e). The alignment process resulted in a 3D-aligned ST dataset with 9,428 cells in slice 1 and 9,803 cells in slice 2. To generate paired scRNA-seq reference data, we considered cells from an additional slice in the control group. We employed the same simulation pipeline to simulate spots for evaluation and generated simulated spots with a grid size of $33 \times 33 \mu\text{m}$. This resulted in simulated 3D-aligned spots containing 1-16 cells within each spot.

Dataset 6: MERFISH MOp 3D

The MERFISH MOp dataset was obtained using an image-based spatial transcriptomics (ST) approach with single-cell resolution [2]. This dataset consists of 254 genes and approximately 300,000 single cells located in 64 mouse brain MOp slices derived from 12 different samples. For our analysis, we selected three adjacent slices, namely “mouse1_slice162”, “mouse1_slice170”, and “mouse1_slice180” from “mouse1_sample4”, to construct a 3D aligned ST dataset (Fig. R1f). After preprocessing and cropping, the three slices contained 972, 950, and 1024 cells, respectively. For the single-cell reference data, here we used the same reference dataset as in Dataset 1. We applied the same simulation pipeline to generate simulated spots with a grid size of $36 \times 36 \mu\text{m}$. This resulted in simulated 3D aligned spots containing 1-7 cells within each spot (Fig. R1f).

Figure R1: Benchmarking Datasets. Four single-slice and two multiple-slice ST datasets with annotated cell types for every single cell are used to generate simulated low-resolution spatial data. Figure shows the process of simulating spatial ST data. A spatial scatter plot displays the ground truth cell types at single-cell resolution (left). Red dashed lines indicate the grids for aggregating cells to spots. Different color represents different cell types. After the aggregation, a scatter plot shows the simulated spots and the cell number of spots at low resolution to show the tissue spatial anatomy (right).

Responses to Reviewer 1’s comments:

[Major comments]

1. *As the cell-type decomposition methods implemented in SpatialScope and RCTD are similar (except for continuous vs discrete models), more comparison between these two methods is recommended. From Figure S3, it looks like they perform similarly well, and RCTD looks like performing slightly better.*

Response:

We thank the reviewer for this constructive comment. As we highlight in Table 1, SpatialScope and RCTD are quite different methods to analyze ST data. Our SpatialScope framework consists of three steps: “Step 1: Nucleus segmentation”, “Step 2: cell type identification”, and “Step 3: gene expression decomposition”. Perhaps, here the reviewer mainly asks about the comparison between “Step 2: cell type identification” of SpatialScope and RCTD. Before we elaborate on the comparison of cell type identification between RCTD and SpatialScope, we would like to emphasize that the main innovation of SpatialScope lies in its “Step 3: gene expression decomposition” by integrating ST data and single-cell reference data with deep generative models. “Step 1: Nucleus segmentation” and “Step 2: cell type identification” are the auxiliary steps for “Step 3: gene expression decomposition”. This is because the number of cells and their corresponding cell types are required in Step 3 to infer gene expression levels of individual cells. Compared to RCTD which only provides the proportion of cell types for each spot, the “Step 3: gene expression decomposition” of SpatialScope refines the spatial transcriptomic landscape at single-cell resolution for low-resolution seq-based ST data, enabling fine-grained cell gradients to be clearly visualized. Single-cell resolution ST data produced by SpatialScope further enables the detection of spatially resolved cellular communication, revealing meaningful biological processes and inter-cellular dynamics in space.

We agree with the reviewer that our model for Step 2 is an extension of the discrete RCTD model. **The major advantage of our extended discrete model comes from the spatial smoothness constraint imposed by the Potts model.** Specifically, let i, i' be the index of spots, and m, m' be the index of cells, and $k_{i,m}$ denote the cell type of m -th cell at spot i . The Potts model is given as

$$p(k_{i,m} | k_{-\{i,m\}}) = \frac{1}{Z} \exp\{-U(\mathbf{K})\}, \quad (\text{R1})$$

where $U(\mathbf{K}) = \sum_{\{i,m\},\{i',m'\} \in \mathbb{N}_{i,m}} \nu [1 - \mathbb{I}(k_{i,m} = k_{i',m'})]$, $\mathbb{I}(\cdot)$ is the indicator function which equals to 1 when $k_{i,m} = k_{i',m'}$ and 0 otherwise, Z is a normalization constant and $\mathbb{N}_{i,m}$ is the set of neighbors of the m -th cell in spot i . Parameter ν controls the smoothness of the cell type labels over space. The larger the ν , the smoother the cell type labels. When we infer cell type identification in Step 2, we perform a maximum posterior probability (MAP) estimate of cell types based on the discrete RCTD model (the likelihood term) and the Potts model (the prior term). In other words, the discrete RCTD model allows us to incorporate spatial information for cell type identification, while it is not quite easy for the original RCTD model.

Next, we perform a simulation study to compare SpatialScope (after steps 1 and 2) and RCTD. We would like to take this opportunity to highlight their different utilities. For SpatialScope (after steps 1 and 2), it outputs the cell types of detected cells at each spot (single-cell resolution). For RCTD, it

outputs the cell type proportions at each spot (spot-level resolution). As their outputs have different resolutions, we first visualize their results of Dataset 1, Dataset 2, Dataset 3, and Dataset 4 in Figure R2, where the simulation details are described in the section titled “Benchmarking Datasets”. As shown in Figure R2, the overall patterns of SpatialScope and RCTD are quite consistent with each other. **Their major difference lies in the resolution of their output, where SpatialScope can output the cell type labels for individual cells within the spots and RCTD only outputs the cell type proportions at each spot.**

Dataset 1

Dataset 2

Dataset 3

Dataset 4

Figure R2: Visualization of the results of “Cell type identification” given by SpatialScope and RCTD for Datasets 1, 2, 3, and 4. (a left, b,c,d upper) Spatial scatter plots depict the ground truth and the cell type identification results obtained by SpatialScope at the single-cell resolution. (a right, b,c,d lower) Spatial scatter pie plots display the ground truth and the inferred cell-type compositions by RCTD at the spot level.

To better demonstrate the effectiveness of spatial smoothness constraint imposed by the Potts model, Reviewer 2 suggests that we should include a discrete version of RCTD model, “StarDist+RCTD”, as a baseline method in the comparison. Specifically, we used StarDist to detect the cell number in each spot, the same as Step 1 of SpatialScope. Based on the number of cells detected at each spot, we directly discretized the cell type proportion estimated by RCTD to obtain the distribution of cell type labels for detected cells. For example, suppose that a spot had 4 cells. If the cell type proportion estimated by RCTD were $[0.23, 0.46, 0.21]$ for cell types A, B and C, respectively. After discretization ($4 \times [0.23, 0.46, 0.21] \approx [1, 2, 1]$), the “StarDist+RCTD” method output 1 cell, 2 cells, and 1 cell of cell types A, B, and C, in the given spot, respectively. Finally, we randomly assigned the cell type labels to the detected cells at the spot, and used it as the final output of the “StarDist+RCTD” method. Clearly, **the main difference between our method and the “StarDist+RCTD” method should be attributed to the spatial smoothness constraint.**

To verify the role of the smoothness constraint, we initially assessed SpatialScope (after steps 1 and 2) and the “StarDist+RCTD” method based on Datasets 1, 2, 3, and 4 (Figure R1). We varied the smoothness hyperparameter ν from 0 to 1,000 for the four single-slice datasets. We computed the error rate for the identified single cell types at each cell location for both SpatialScope and “StarDist+RCTD”. The error rates are depicted in Figure R3. The error rate trends with different ν values remained consistent across the four datasets. When $\nu = 0$, indicating no smoothness constraint, we anticipated that the error rate of SpatialScope would be comparable to that of the “StarDist+RCTD” method. Conversely, for very large ν values (e.g., $\nu = 1,000$), the error rate increased significantly due to excessive smoothing. By incorporating spatial information within the range of $\nu = 10 \sim 50$, SpatialScope demonstrated substantial improvement in accurately identifying single cell types at each cell location. Therefore, we set $\nu = 10$ as the default setting in our software to avoid fine-tuning this smoothness hyperparameter in real data analysis.

Figure R3: The effect of incorporating spatial information under different ν values in cell type identification task compared with the “StarDist+RCTD” method for Datasets 1, 2, 3, and 4. Bar plots of the error rates under different ν values for the cell type identification step of SpatialScope and the “StarDist+RCTD” method. The red line indicates the error rate of the “StarDist+RCTD” method as a baseline. Data are presented as mean values $\pm 95\%$ confidence intervals; $n = 10$ simulation replicates.

Regarding Figure S3, we redesigned a comprehensive evaluation of cell type identification between methods in two cases: (Case a) For methods that can infer cell type labels at the single-cell level, namely SpatialScope, Tangram, and CytoSPACE, we compared their performance in cell type identification at the single-cell resolution using error rate; (Case b) For methods that provide cell type proportions at spot level, such as RCTD, SpatialDWLS, and Cell2location, we compared their performance estimating cell type proportions at the spot level using Pearson correlation (PCC) and root-mean-square error (RMSE). Therefore, in order to present a quantitative evaluation of SpatialScope and RCTD, we need to **aggregate the results at the single-cell level to the spot level** since SpatialScope provides cell type labels for each individual cell. Here is how we aggregate the results of SpatialScope to the spot level. For instance, suppose there are four cells at a spot, and the inferred type labels for these cells are A cell, B cell, C cell, and B cell, respectively. Then, the aggregated cell type proportions from SpatialScope’s output at the spot level would be [0.25, 0.5, 0.25] for cell types A, B, and C, respectively. We aggregate the results of StarDist+RCTD in the same way for comparison with RCTD. This allows for a fair comparison between SpatialScope, RCTD, and StarDist+RCTD at the spot level.

We calculated the PCC and RMSE of the cell type proportions estimated by SpatialScope and RCTD at the spot level, based on the four simulated datasets. We varied ν to observe the impact of spatial smoothness. The results are shown in Figures R4 and R5. When ν is small, SpatialScope demonstrates higher PCC and lower RMSE for most of the simulated data compared to RCTD and the “StarDist+RCTD” method. This indicates the effectiveness of incorporating spatial information in SpatialScope.

We have included the aforementioned comparison between SpatialScope and RCTD as a dedicated section within the Methods section of the revised manuscript. Detailed information regarding this comparison can also be found in the revised Supplementary Note Section 2.8. In the revised manuscript, we have highlighted the effectiveness of incorporating the spatial smoothness constraint, as extensively described in Supplementary Note Section 2.9.4. Regarding Figure S3, we have excluded it from the revised manuscript as we have incorporated the discussion of a new benchmarking study using six newly acquired datasets of enhanced quality in a dedicated section titled “A benchmarking study on cell type identification and gene expression decomposition.”

PCC

Figure R4: The effect of incorporating spatial information under different ν in deconvolution task compared with baseline method RCTD and StarDist+RCTD for Datasets 1, 2, 3, and 4. Metric: PCC, the higher the better. Bar plots of PCC under different ν values for the cell type identification step of SpatialScope are shown. The blue and purple lines represent the PCC values of RCTD and StarDist+RCTD, respectively, serving as baselines. Data are presented as mean values $\pm 95\%$ confidence intervals; $n = 10$ simulation replicates.

RMSE

Figure R5: The effect of incorporating spatial information under different ν in deconvolution task compared with baseline method RCTD and StarDist+RCTD for Datasets 1, 2, 3, and 4. Metric: RMSE, the lower the better. Bar plots of the error rate under different ν values for the cell type identification step of SpatialScope are displayed. The blue and purple lines represent the RMSE values of RCTD and StarDist+RCTD, respectively, serving as baselines. Data are presented as mean values $\pm 95\%$ confidence intervals; $n = 10$ simulation replicates.

2. The model performance is expected to change depending on gene groups. Less abundant and more variable genes would likely be more challenging to predict. Could the authors add some assessment of the performance for low, medium and high abundant/variable genes? For imputation, how would the existing panels affect the imputation of genes not in the panels? For example, how well can the model predict genes in non-brain tissue if a panel is designed for brain markers?

Response:

We appreciate the reviewer for the valuable suggestions. In order to assess the performance of predicting low, medium, and high abundant/variable genes, we used a MERFISH dataset, where the expression profiles for 254 genes were measured in 5,551 single cells in a mouse brain slice from the primary motor cortex (MOp) [2] and droplet-based snRNA-seq profiles from mouse MOp as the reference dataset [6], in the benchmarking study. We compared SpatialScope with other imputation methods such as Tangram, gimVI, and SpaGE.

We categorized genes into low, medium, and high abundant groups based on their expression levels. Additionally, we categorized genes into low, medium, and high variable groups using the p -values obtained from spatial differential expression analysis. Specifically, genes with expression levels falling within the 0 ~ 10, 45 ~ 55, and 90 ~ 100 quantiles of all gene expression levels were assigned to the low, medium, and high abundant groups, respectively. Moreover, genes with p -values falling within the 0 ~ 10, 45 ~ 55, and 90 ~ 100 quantiles of all gene p -values generated by SPARK-X [7] were selected as high, medium, and low variable genes, respectively.

To assess the imputation performance, we randomly selected 25 genes from each group for testing purposes. These genes were excluded from the training set, while the remaining genes were used as training genes. The expression profiles of the 25 testing genes measured in the dataset were considered as the ground truth for evaluation. We utilized a metric called relative MAE to quantify the imputation performance, which is defined as follows:

$$\text{Relative MAE} = \frac{\sum_g \|x_g^{\text{Impute}} - x_g^{\text{GT}}\|}{\sum_g x_g^{\text{GT}}}, \quad (\text{R2})$$

where x_g^{Impute} represents the imputed value of gene g , and x_g^{GT} denotes the ground truth value of gene g . Both x_g^{Impute} and x_g^{GT} are normalized from the count scale to the range (0,1). The relative MAE reflects that less abundant genes are more challenging to predict, and the performance of all methods deteriorates as the gene’s abundance level decreases (Fig. R6a).

Slightly different from expectation, the performance of all methods becomes better when the gene expression level is highly variable (Fig. R6b). This is because if the gene is more variable, it tends to show a specific spatial pattern, or it’s more likely a marker gene of a cell type (Fig. R7). With a more obvious spatial pattern for the gene, capturing the pattern from the data for all methods is easier. For example, for SpatialScope, it’s easy to learn the pattern of cell type marker genes from single-cell reference data. On the contrary, the genes with less abundant or less variable are hard to see spatial patterns (Fig. R7) or even opposite to the pattern in single-cell reference data (e.g., gene *Adam2* in “low abundant genes” in Figure R7). The low measurement quality and randomness degrades the performance of all methods. For example, SpatialScope predicts the expressions of

gene *Adam2* mainly in the upper layer, which aligns with the gene expression signatures observed in snRNA-seq reference data (Fig. R7b). However, in the MERFISH dataset, the expression pattern of *Adam2* appears more random, as it is observed in both the upper and lower layers, contrary to the expression pattern observed in the snRNA-seq reference. This observation helps explain the SpatialScope’s results pertaining to “Low Abundant Genes” (Fig. R6, red arrow).

Overall, SpatialScope outperformed other methods in almost all gene groups (Fig. R6). For low abundant or low variable genes, whose expression levels measured in the MERFISH data are very low and have little spatial pattern, SpatialScope can still predict expressions that are consistent with the gene expression signatures in snRNA-seq reference. For example, the measured gene *B4galnt3* that belongs to low variable genes shows little spatial pattern (Fig. R7e, the first column). However, SpatialScope predicts its relatively high expression in L6 CT and L6b layer (Fig. R7e, the second column), which is in accordance with the signatures in snRNA-seq reference (Fig. R7e, the last column). Other methods overestimate its expression in the upper layer. For medium and high abundant genes or medium and high variable genes, the spatial expression pattern of genes becomes more obvious. For these genes, SpatialScope can accurately predict genes’ spatial patterns, which are consistent with the measured spatial patterns and the signatures in the snRNA-seq reference. Tangram sometimes overestimates the expression or gives the wrong spatial pattern, like gene *Syndig1* (Fig. R7d) or gene *Brinp3* (Fig. R7g). Methods gimVI and SpaGE missing some spatial patterns in the prediction. For example, gimVI misses the high expression of gene *Lsp1* (Fig. R7f) in L6b and SpaGE misses overestimates the expression of gene *Slc30a3* (Fig. R7g) in L6 CT and L6b layer. Overall, the metric Relative MAE for evaluating performance illustrates the high prediction accuracy of SpatialScope.

Figure R6: Performance of different methods imputing different gene groups with metric Relative MAE. **a**, Bar plot of Relative MAE from methods Tangram, gimVI, SpaGE, SpatialScope imputing low, medium and high abundant genes. **a**, Bar plot of Relative MAE from methods Tangram, gimVI, SpaGE, SpatialScope imputing low, medium and high variable genes. Data are presented as mean values $\pm 95\%$ confidence intervals; $n = 25$ selected genes in each group.

Figure R7 (previous page): The results of different methods predicting low, medium, and high abundant genes and low, medium, and high variable genes. **a**, Cell type identification results of MERFISH MOp data by SpatialScope showing a clear layer structure of mouse brain. Cell type identification results in each of the three major categories are shown on the right. **a,b,c,d,e,f,g** shows the prediction results of low abundant genes, medium abundant genes, high abundant genes, low variable genes, medium variable genes, and high variable genes, respectively. In each group, the figure shows measured and imputed expressions of selected three genes in that group. Each row corresponds to a single gene. The first column from the left shows the measured spatial gene expression in the MERFISH dataset, while the second to fifth columns show the corresponding imputed expression pattern by SpatialScope, Tangram, gimVI, SpaGE. The imputation accuracy was evaluated by Relative MAE and displayed with bar plots (sixth column). The marker gene expression signatures in snRNA-seq reference were displayed with a heatmap plot (seventh column).

We also appreciate the reviewer for bringing up the question regarding the imputation performance of genes not present in the MERFISH data. We agree that further evaluation of the imputation accuracy for non-brain tissue markers would be valuable to support the robustness of our method. To address this concern, we utilized a publicly available marker gene database called CellMarker 2.0 [8]. This database offers a manually curated collection of experimentally supported markers for various cell types in different human and mouse tissues. It encompasses a total of 35,197 tissue-cell type-marker triplets and 9,616 unique markers. To focus specifically on non-brain tissue markers, we excluded all markers related to brain tissue or those present in the MERFISH data, resulting in a selection of 4,806 unique markers for further analysis. Subsequently, we identified three non-brain tissues with the highest number of unique markers and evaluated the imputation accuracy of their respective gene panels. Since ground truth data for these non-MERFISH genes is not available, we utilized the Allen ISH dataset [9] for validation purposes. The Allen ISH dataset provides a valuable resource for assessing the accuracy of gene expression patterns, allowing us to validate the imputation performance of the non-brain tissue gene panels. By leveraging the Allen ISH dataset as a validation set, we can assess the imputation accuracy of the non-MERFISH genes and compare the imputed expression patterns with the actual gene expression patterns observed in the Allen ISH dataset. This approach enables us to evaluate the reliability and effectiveness of the imputation process for non-brain tissue markers.

Figure R8 displays the imputation performance of the compared methods for the gene panel in kidney tissue. As an example, we examined the marker gene *Pde1a*, which is associated with duct intercalated cells in the kidney. The Allen ISH dataset reveals that *Pde1a* tends to exhibit higher expression in the bottom cortical layers of the MOp region, consistent with the imputation results obtained by SpatialScope. Conversely, other imputation methods tend to overestimate the spatial expression of *Pde1a* in the upper cortical layers (Fig. R8, first row). Similarly, we considered the marker gene *Ntrk3*, associated with podocyte cells in the kidney. SpatialScope is the only method capable of successfully recovering the observed spatial expression pattern in the Allen ISH dataset, where higher expression is observed in the L2/3 IT layer and lower expression in other layers (Fig. R8, third row). To demonstrate the robustness of our method, we further evaluated the imputation

performance of gene panels from bone marrow (Fig. R9) and lung (Fig. R10) tissues. For instance, both *Col1a* and *Lum* show expression solely in the top cortical layer of the MOp region in the Allen ISH dataset (Fig. R9), consistent with the imputation results generated by SpatialScope. In summary, these findings provide evidence supporting the ability of SpatialScope to reasonably impute non-brain tissue gene panels, even when using a limited number of markers designed for brain tissue.

We have included a discussion on the new experimental results of SpatialScope regarding the prediction of low, medium, and high abundant/variable genes and gene panels from diverse tissues in the revised manuscript on page 23. Additionally, we have revised Supplementary Note Section 2.9.12 to cover these findings in detail.

Figure R8: Measured and imputed expressions of non-MERFISH kidney panel genes. Each row corresponds to a single gene. The first column from the left displays the ISH images from the Allen Brain Atlas, while the second to fifth columns show the corresponding imputed expression patterns by SpatialScope, Tangram, gimVI, and SpaGE. The gene expression signatures in the snRNA-seq reference are depicted using a heatmap plot in the sixth column.

Bone marrow panel genes

Figure R9: Measured and imputed expressions of non-MERFISH bone marrow panel genes. Each row corresponds to a single gene. The first column from the left shows the ISH images from the Allen Brain Atlas, while the second to fifth columns show the corresponding imputed expression patterns by SpatialScope, Tangram, gimVI, and SpaGE. The gene expression signatures in the snRNA-seq reference are displayed using a heatmap plot in the sixth column.

Figure R10: Measured and imputed expressions of non-MERFISH lung panel genes. Each row corresponds to a single gene. The first column from the left shows the ISH images from the Allen Brain Atlas, while the second to fifth columns show the corresponding imputed expression patterns by SpatialScope, Tangram, gimVI, and SpaGE. The gene expression signatures in the snRNA-seq reference are displayed with a heatmap plot in the sixth column.

3. *Similarly, the performance would change for rare cell types and tissue regions with dense nuclei. How would these two factors affect the model performance?*

Response:

We appreciate the reviewer’s constructive comment. First, we investigate the influence of cell type abundance. Next, we examine tissue regions with dense nuclei. We use Dataset 1 in the benchmarking study.

In the simulated single-cell reference data, L4/5 IT and Vip have the highest abundance (12.5%) and lowest abundance (0.5%), respectively. We first trained a neural network to approximate the conditional score function, such that we learned the distribution of expression patterns of different cell types based on the single-cell reference data. Then we ran the Langevin dynamics to perform posterior sampling for gene expression decomposition at each spot. We used the posterior means as the inferred expression levels of individual cells. After that, we calculated the cosine similarity between the inferred expression levels and the observed expression to quantify the accuracy of gene expression decomposition.

As shown in Figure R11a, we can see that the accuracy of gene expression decomposition is not largely affected by the cell type abundance. For example, the decomposition accuracies of two cell types L4/5 IT (highest abundance, 12.5%) and Vip (lowest abundance 0.5%) are more or less the same. This indicates that our conditional score function can be well-trained even with a relatively small number of cells. However, we observe that the decomposition accuracy is more related to the heterogeneity of a cell type. Here the heterogeneity of a cell type is measured by the average cosine similarity of expression levels between two randomly chosen cells from the given cell types, using the single-cell reference data. The higher cosine similarity, the lower heterogeneity of the given cell type. Figure R11b shows that the decomposition result is more accurate when the expression levels are similar within a cell type. The linear regression results further support this notion (Fig. R11c). When regressing the decomposition accuracy against cell type proportion, the R-squared value is merely 0.11. However, when regressing the decomposition accuracy against cell type heterogeneity, the R-squared value increases significantly to 0.71. This suggests that the decomposition accuracy is more associated with the heterogeneity within a cell type rather than the proportion of the cell type in the single-cell reference data.

Figure R11 (previous page): Influence of unbalanced cell types in training the score-based generative model. **a**, Gene expression decomposition performance of SpatialScope for each cell type in the MERFISH simulation dataset, the cell types are sorted by their proportion (shown in the x-ticks) in the single-cell reference data. **b**, The mean cosine similarities between two randomly selected cells for each cell type in the paired single-cell reference of the MERFISH simulation dataset. Error bars represent the 95% confidence interval of cosine similarity evaluated on cells from different cell types. **c**, Left, Linear regression of decomposition accuracy within a cell type on cell type proportion. Right, Linear regression of decomposition accuracy within a cell type on cell type homogeneity. Different colors represent the cell type of the data point.

Next, we quantitatively measured the performance of gene expression decomposition **by separate evaluation of spots with different cell numbers**. Recall that in the gene expression decomposition step, we infer the transcriptome-wide expression levels for each single cell. We compare SpatialScope with Tangram and CytoSPACE, which can also achieve gene expression decomposition. We still use the simulated dataset generated from the benchmarking Dataset 1 (Fig. R12) to illustrate the effect of cell numbers on inferring gene expression levels of each single cell. We examined two simulation settings, where either the paired single-cell reference data (produced from the same sample as the ST data, in this case, the right part of Figure R12) or the unpaired single-cell reference data (independently generated scRNA-seq reference of the same tissue type, in this case, an external mouse Primary visual (VISp) scRNA-seq data [10]) are used for gene expression decomposition.

We first explored the setting where we used paired single-cell reference data for gene expression decomposition. As shown in Figure R13, the performance of all compared methods (measured by cosine similarity) degrades as the cell number increases. This is because more cells in a spot means more components introduce more uncertainty in the decomposition. Nevertheless, it is important to note that our method SpatialScope achieved the best performance in different scenarios of cell numbers and UMI subsample rates in the construction of simulated spots.

When an independently generated scRNA-seq reference of the same tissue type was used (Fig. R14), we observed consistent patterns regarding different cell numbers within spots, and SpatialScope outperformed other methods in all settings. As a comparison, the performances of alignment-based methods, Tangram and CytoSPACE, decrease dramatically when the reference data is unpaired (batch effects exist across platforms).

We have incorporated a discussion on the robustness of SpatialScope to variations in cell type abundance in the single-cell reference data and unbalanced cell numbers within spots in the revised manuscript on page 11. Furthermore, we have provided detailed discussions in the Methods section and Supplementary Note Sections 2.9.9 and 2.9.10.

Figure R12: The process of generating a simulated dataset using MERFISH dataset. MERFISH cells are divided into two subsets. The left part is used to make pseudo-spots by aggregating the cells within each grid, and the right part is regarded as paired scRNA-seq reference. Uniform gridding is performed and cells in squares are aggregated to generate simulated spots.

Figure R13: Comparison of gene expression decomposition for spots with varying cell numbers when reference is paired. The cosine similarities between the ground truth and predicted gene expressions for cells with correctly identified cell type label (top) or all cells (bottom) under different combination scenarios of UMI subsample rate. Error bars represent the 95% confidence interval of cosine similarity evaluated on spots with varying cell numbers.

Figure R14: Comparison of gene expression decomposition for spots with varying cell numbers when reference is unpaired. The cosine similarities between the ground truth and predicted gene expressions for cells with correctly identified cell type label (top) or all cells (bottom) under different combination scenarios of UMI subsample rate. Error bars represent the 95% confidence interval of cosine similarity evaluated on spots with varying cell numbers.

[Minor suggestions]

1. *Abstract: revise "remain unable to provide spatial characterization at transcriptome-wide single-cell resolution" as there are methods able to do so like STomics and Slide-seq; similarly, in the Introduction session, the author should update that seqFISH and MERFISH can both measure more than 10,000 genes*

Response:

Thanks for this comment. We have revised the abstract as “Although current ST methods, whether based on next-generation sequencing (seq-based approaches) or fluorescence in situ hybridization (image-based approaches), offer valuable insights, they face limitations either in cellular resolution or transcriptome-wide profiling. This restricts their capacity to fully elucidate intricate tissue structures and identify intercellular communications.”

We have updated the introduction section to have a better overview of ST technologies.

“Current ST approaches are predominantly based on either next-generation sequencing (seq-based) or fluorescence in situ hybridization (image-based). Seq-based approaches, such as 10x Visium [11], Slide-seq [12] and Stereo-seq [13], can detect transcriptome-wide gene expression within spatial spots. Among them, the Visium technology has gained considerable maturity over the years, becoming a well-established commercially available method in the field of ST. According to the database collected by the museum of spatial transcriptomic project [14], more than half of studies in the

past year still utilized the Visium technology to quantify gene expression in space, accumulating a substantial amount of data [14]. However, considering the larger spot size of 55 μm , a Visium spot often contains multiple cells, which limits its usage in resolving detailed tissue structure and in characterizing cellular communications (e.g., identifying ligand-receptor interactions [15]). Image-based approaches such as seqFISH [16] and MERFISH [17] are designed to measure thousands of genes with single-cell resolution, but they often lack whole-transcriptome coverage, resulting in only a few hundred genes in real applications.”

2. *Figure S3 legend: 3 not 4 datasets; GT = Ground Truth*

Response:

Thank you for the thoughtful reminder. Following the suggestion of Reviewer 2, we have conducted a comprehensive benchmarking study using six new datasets of improved quality. As a result, we have removed Figure S3 from our revised manuscript. We have incorporated the discussion of the new benchmarking study in a revised manuscript section titled "A benchmarking study on cell type identification and gene expression decomposition." The details of the simulations are described in the revised Supplementary Note Section 2.9.1.

Responses to Reviewer 2’s comments:

[Comments on results]

1. *Lack of more detailed performance comparison. In the review by Qu et al.[1], a total of eight integration methods were evaluated, including Tangram, gimVI, SpaGE, Seurat, SpaOTsc, novoSpaRc, LIGER and stPlus, for predicting the spatial distribution of undetected RNA transcripts in spatial transcriptomics datasets. In addition to predict the spatial distribution of RNA transcripts, Tangram, Seurat, SpaOTsc, and novoSpaRc can also assign cells from scRNA-seq data to spatial locations in tissue sections. Additionally, Cell2location, SpatialDWLS, RCTD, Stereoscope, DestVI, STRIDE, SPOTlight, and DSTG can be used to predict the cell type composition of spots in tissue sections by combining spatial transcriptomics data and scRNA-seq data. As mentioned above, it would be great if the authors could add the missing comparison of state-of-art methods, including SpaOTsc, novoSpaRc, DestVI, STRIDE, SPOTlight and DSTG. There is also an additional review available for reference [18].*

Response:

We thank the reviewer for this constructive comment. In the previous manuscript, we selectively included three top-performing methods, namely RCTD, cell2location, and SpatialDWLD, based on the review paper by Qu [1]. Additionally, we incorporated two new methods, CARD and CytoSPACE, which were not covered in Qu’s review paper but were included in the new review paper by Li [18]. However, in response to the reviewer’s concern, we recognize the importance of conducting a comprehensive and systematic comparison among many other methods mentioned by the reviewer. Therefore, we would like to take this opportunity to perform a comprehensive analysis by comparing SpatialScope with all the methods mentioned. In total, we compared SpatialScope with twelve existing methods from published and preprint papers, including Tangram [6], CytoSPACE [19], RCTD [20], SpatialDWLS [21], Cell2location [22], CARD [23], SpaOTsc [24], novoSpaRc [25], DestVI [26], STRIDE [27], SPOTlight [28], and DSTG [29].

Furthermore, in response to the suggestion made by Reviewer 2 in Question 7, we have included StarDist+RCTD as a baseline method for comparison. Thus, our analysis encompasses a total of twelve methods, including SpatialScope, Tangram, CytoSPACE, StarDist+RCTD, RCTD, cell2location, SpatialDWLS, CARD, SpaOTsc, novoSpaRc, DestVI, STRIDE, SPOTlight, and DSTG. It is worth noting that among these methods, only SpatialScope, Tangram, CytoSPACE, and StarDist+RCTD have the capability to infer cell type labels at single-cell resolution. In contrast, the remaining methods can only provide cell type proportions at the spot level.

To ensure a comprehensive comparison, we have conducted a redesigned benchmarking study using six datasets, as described in the section titled "Benchmarking datasets" at the beginning of this response letter and illustrated in Figure R1. The comparison study is organized into three distinct categories: (i) Cell type identification. (ii) Gene expression decomposition. (iii) Imputation of gene expression.

Before we delve into the details of the simulation study, it is worth mentioning how we designed and implemented StarDist+RCTD in detail to avoid any confusion. (i) In the cell type identification task, we used StarDist to detect the cell number in each spot, which was the same as Step 1 of SpatialScope.

Based on the number of cells detected at each spot, we directly discretized the cell type proportion estimated by RCTD to obtain the distribution of cell type labels for the detected cells. For example, suppose a spot had 4 cells. If the cell type proportions estimated by RCTD were $[0.23, 0.46, 0.21]$ for cell types A, B, and C, respectively, after discretization ($4 \times [0.23, 0.46, 0.21] \approx [1, 2, 1]$), the “StarDist+RCTD” method output 1 cell, 2 cells, and 1 cell of cell types A, B, and C, respectively, in the given spot. Finally, we randomly assigned the cell type labels to the detected cells at the spot and used it as the final output of the “StarDist+RCTD” method. (ii) In the gene expression decomposition task, we directly assigned the average expression of cell types learned from single-cell RNA-seq data as the inferred gene expression for each single cell, as suggested by Reviewer 2 in Question 7. Using the same example, suppose that StarDist+RCTD had detected 1 cell, 2 cells, and 1 cell of cell types A, B, and C in the given spot. The inferred gene expression for the cell type A cell was the average expression of cell type A learned from the single-cell reference data. The average expressions of cell types B and C were assigned to the detected locations of cells B and C, respectively. It should be noted that the two cells of type B had the same inferred gene expression level, both being the average expression of cell type B, by StarDist+RCTD.

(i) Cell type identification

We conducted a comprehensive evaluation of cell type identification accuracy in two cases.

Case (a): For methods that can infer cell type labels at the single-cell level, namely SpatialScope, Tangram, CytoSPACE, and StarDist+RCTD, we compared their performance in cell type identification at the single-cell resolution. We assessed the accuracy of cell type identification using the misclassification error rate, which quantifies the rate of misclassified cells among the inferred cell type labels.

Case (b): Many existing methods, such as RCTD, SpatialDWLS, and Cell2location, provide cell type proportions at the spot level. To compare SpatialScope with these methods, we aggregated its cell type identification results from the single-cell level to the spot level. For instance, if SpatialScope identified 1 cell, 2 cells, and 1 cell as cell types A, B, and C, respectively, at a given spot, we aggregated the single-cell level results to obtain estimated cell type proportions of 25%, 50%, and 25% for cell types A, B, and C, respectively, at that spot. We applied the same procedure to aggregate the single-cell level results of Tangram, CytoSPACE, and StarDist+RCTD to the spot level, ensuring that all methods could be evaluated on a comparable spot level basis. To assess the accuracy of estimated cell type proportions at the spot level, we employed two metrics: Pearson correlation (PCC) and root-mean-square error (RMSE). These metrics quantified the correlation and the deviation between the estimated cell type proportions obtained by each method and the ground truth values.

We used two kinds of datasets to evaluate the accuracy of each method in identifying cell types at the single-cell level (Case (a)) or providing cell type proportions at the spot level (Case (b)). One type is single-slice datasets, which include Dataset 1, Dataset 2, Dataset 3, and Dataset 4 (Fig. R1). Each dataset in this type consists of a single slice. The other type is multiple-slice datasets, which include Dataset 5 and Dataset 6. Each dataset in this type contains multiple slices, with 2 slices for Dataset 5 and 3 slices for Dataset 6. We included multiple-slice (3D) datasets in the benchmarking to test the ability of each method to integrate information from multiple slices. It is

worth noting that SpatialScope is the only method that can leverage information from adjacent slices when applied to multiple-slice (3D) datasets.

In Case (a), we first applied SpatialScope, Tangram, CytoSPACE, and StarDist+RCTD to 4 single-slice datasets (Fig. R15a). SpatialScope outperformed all other cell type identification methods, with a 50.3%, 20.6%, and 6.3% improvement in terms of error rate compared to Tangram, CytoSPACE, and StarDist+RCTD, respectively, across all four single-slice datasets. We observed the same pattern when applying the four methods to the two multiple-slice datasets (Fig. R15b). SpatialScope remained the most accurate method for inferring cell type labels at the single-cell level, with a 22.9-50.0% improvement in error rate for Dataset 5 and a 4.6-48.3% improvement in error rate for Dataset 6. To examine the ability to integrate multiple slices using different methods, we prepared two settings for applying the four methods to multiple-slice datasets. In setting (i), we applied the methods to each single slice one by one in the dataset (Fig. R15b, “Run with single slice”). In setting (ii), we applied the methods to all slices at once in the dataset (Fig. R15b, “Run with multiple slices”), where all slices were aligned in the xy-axis and evenly spaced in the z-axis. If a method could utilize spatial information and borrow information from adjacent slices, the cell type identification accuracy should improve in setting (ii) compared to setting (i). This pattern was observed for SpatialScope, where the error rate decreased under setting (ii) compared to setting (i) in all multiple-slice datasets. However, the same pattern was not observed for the error rate of other methods. For example, the error rate of Tangram decreased under setting (ii) compared to setting (i) in Dataset 5 but increased in Dataset 6. SpatialScope can benefit from incorporating spatial information in the model design when applied to multiple-slice data.

In Case (b), SpatialScope was compared to existing deconvolution methods that can only provide cell type proportions at the spot level, measured by PCC and RMSE (Fig. R15c,d). SpatialScope outperformed or was comparable to the other methods in all datasets in terms of PCC and achieved the highest deconvolution accuracy in terms of RMSE across all datasets. SpatialScope showed improvements in PCC ranging from 9.4% to 157.5% compared to Tangram, CytoSPACE, SpaOTsc, novoSpaRc, STRIDE, and SPOTlight across the six datasets. It also achieved a maximum improvement of 51.4% compared to RCTD, SpatialDWLS, Cell2location, CARD, DestVI, and DSTG for the same datasets. In terms of RMSE, SpatialScope achieved an accuracy improvement of 25.3% to 89.7% over all other methods across the six datasets. However, when applying SpatialScope to multiple-slice datasets, PCC and RMSE did not show significant improvement when integrating multiple slices compared to using only a single slice (Fig. R15d). This pattern differs from the improvement observed when measuring the error rate (Fig. R15b). The reason behind this is that PCC and RMSE are measured at the spot level, which is a coarse resolution that cannot capture the improvement gained from borrowing information across slices. The improvement can only be observed at a higher resolution by measuring the error rate at the single-cell level.

(ii) Gene expression decomposition

The novelty of SpatialScope mainly comes from the gene expression decomposition step, which is the third step of SpatialScope. After inferring the cell type label at the single-cell level, we further infer the transcriptome-wide expression levels for each single cell in the gene expression decomposition step by leveraging deep generative models (Table 1). Note that inferring the transcriptome-wide expression

levels for each single cell can not be achieved by most deconvolution methods. Among the twelve methods in our comparison, only SpatialScope, Tangram, CytoSPACE, and StarDist+RCTD are capable of inferring the expression levels at single-cell level. Therefore, we conducted a benchmarking study to compare the performance of the four methods in inferring the expression levels at the single-cell level using four single-slice datasets from the section “Benchmarking datasets” (Dataset 1-4).

To assess the robustness of the compared methods to data quality, we examined three simulation settings. Each setting corresponds to a different sparsity level of the expression matrix, achieved by varying the unique molecular identifier (UMI) counts through downsampling. Different sparsity levels simulate different capture rates of spatial transcriptomics data, which are correlated with data quality. A high sparsity level indicates a lower capture rate and lower data quality, while a low sparsity level indicates a higher capture rate and higher data quality. The three settings correspond to three levels of simulated capture rates: low, medium, and high (Fig. R16a). To quantify the accuracy of gene expression decomposition, we computed the cosine similarity between the estimated gene expression and the underlying truth for each single cell.

For all four datasets, the inferred transcripts at single-cell level by SpatialScope had significantly higher accuracy than other methods, with 64.6%, 32.1% and 11.4% improvement in terms of cosine similarity compared to Tangram, CytoSPACE and StarDist+RCTD in overall settings and datasets (Fig. R16a). SpatialScope’s superiority in gene expression decomposition stems from its fundamental differences compared to other methods. Unlike alignment-based methods such as Tangram and CytoSPACE, which assign existing cells from scRNA-seq data to spatial spots, or methods like StarDist+RCTD that assign the average gene expression of cell types, SpatialScope has the ability to generate the gene expressions of pseudo-cells using the learned deep generative model. This generation process allows SpatialScope to match the observed spot-level gene expression in space, leading to more accurate results.

Next, we examined the gene expression accuracy of different methods at distinct simulated capture rate levels of the datasets. SpatialScope exhibits a consistent pattern where the accuracy increases with higher capture rates (Fig. R16a), indicating its ability to fully leverage data quality. This pattern is not observed or not evident in the results of other methods, suggesting that they are unable to fully leverage the information contained in the data. These results have been updated in the revised main text and section 2.9.2 of the revised supplementary text.

(iii) Imputation of gene expression

SpatialScope is a unified approach capable of handling both sequence-based data and image-based ST data. For image-based ST data, SpatialScope can perform the imputation of expression levels for unmeasured genes. In our previous manuscript, we compared SpatialScope with three top-performing methods for the gene imputation task, namely Tangram, gimVI, and SpaGE, as identified in the review paper by Qu [1]. In response to the reviewer’s concern, we have included four additional methods for comparison: Seurat, SpaOTsc, novoSpaRc, and stPlus. Overall, we compared the performance of gene expression imputation using SpatialScope against seven existing methods from published papers and preprints, including Tangram [6], gimVI [30], SpaGE [31], SpaOTsc [24], novoSpaRc [25], stPlus [32], and Seurat [33].

Using a MERFISH dataset, where the expression profiles for 254 genes were measured in 5,551 single-cells in a mouse brain slice from the primary motor cortex (MOp) [2] and a droplet-based snRNA-seq profiles from mouse MOp as the reference dataset [6], we applied SpatialScope and other seven methods to predict unmeasured spatial gene expression patterns. To evaluate the imputation performance, we selected the cortical layer-specific markers (*Cux2*, *Otof*, *Rorb*, *Rspo1*, *Sulfb2*, *Fezf2* and *Osr1*) as testing genes for better visualization of the predicted spatial gene expression patterns and then removed them from the dataset. The remaining genes are used as training genes and serve as input for eight methods to predict the spatial expression pattern of leaved-out marker genes. We evaluated the imputation performance by computing MAE between the real measurement and the predicted gene expression of the testing genes. Results show a significant improvement in performance for SpatialScope compared to Tangram, gimVI, SpaOTsc, novoSpaRC, with 33.6%, 34.3%, 43.4% and 53.6% and SpatialScope is comparable to state-of-art methods SpaGE, stPlus and Seurat in terms of predicting spatial gene expression of seven cortical layer-specific markers (Fig. R17). Next, we used all overlapped genes between MERFISH data and used single-cell reference data as training genes, and evaluated the imputation performance of non-MERFISH genes. Since ground truth data for these non-MERFISH genes is unavailable, we utilized the Allen ISH dataset [9] for validation purposes. The imputation performance is consistent with the previous setting (Fig. R18). The state-of-the-art performance of SpatialScope demonstrates that it can also be a powerful tool in gene expression imputation as a unified framework.

We have incorporated the discussion of the new benchmarking study in a revised manuscript section titled “A benchmarking study on cell type identification and gene expression decomposition.” The details of the simulations are described in the revised Supplementary Note Section 2.9.1.

Figure R15: Benchmarking of cell type identification based on the single-cell level (Case (a)) and the spot level (Case (b)). **a**, The bar plots of error rate of each method in inferring cell type label at single-cell level for 4 single-slice benchmarking datasets (Dataset 1-4). **b**, The bar plots of error rate of each method in inferring cell type label at the quesi-cell level for 2 multiple-slice benchmarking datasets (Dataset 5-6). Two settings are considered. One is to apply methods to the single slice in the dataset one by one (“Run with single slice”), and another is to apply methods to all slices at once (“Run with multiple slices”). **c**, The bar plots of PCC and RMSE of each method in inferring cell type proportion at spot level for 4 single-slice benchmarking datasets (Dataset 1-4). Data are presented as mean values $\pm 95\%$ confidence intervals; n is the number of spots. **d**, The bar plots of PCC and RMSE of each method in inferring cell type proportion at spot level for 2 multiple-slice benchmarking datasets (Dataset 5-6). Two settings are considered. One is to apply methods to the single slice in the dataset one by one (“Run with single slice”), and another is to apply methods to all slices at once (“Run with multiple slices”). Data are presented as mean values $\pm 95\%$ confidence intervals; n is the number of spots.

Figure R16: Benchmarking of gene expression decomposition and imputation tasks. **a**, The bar plots of cosine similarity of each method in inferring transcriptome-wide expression levels for each single cell for 4 single-slice benchmarking datasets (Dataset 1-4) at different simulated capture rates. **b**, The bar plots of the overall MAE (first column) and MAE of seven cortical layer-specific marker genes (second to eighth column) of each method in predicting unmeasured spatial gene expression patterns.

Figure R17: Comparison of gene expression imputation of new included methods for MERFISH genes. Measured and imputed expressions of known spatially patterned genes in the MERFISH dataset. Each row corresponds to a single gene. The first column from the left shows the measured spatial gene expression in the MERFISH dataset, while the second to sixth columns show the corresponding imputed expression pattern by SpatialScope, SpaOTsc, novoSpaRc, stPlus, and Seurat. The imputation accuracy was evaluated by MAE and displayed with bar plots (seventh column). The marker gene expression signatures in snRNA-seq reference were displayed with heatmap plots (eighth column).

Figure R18: Comparison of gene expression imputation of new included methods for Non-MERFISH genes. Each row corresponds to a single gene. The first column from the left shows the ISH images from the Allen Brain Atlas, while the second to sixth columns show the corresponding imputed expression pattern by SpatialScope, SpaOTsc, novoSpaRc, stPlus, and Seurat. The imputation accuracy was evaluated by MAE and displayed with bar plots (seventh column). The marker gene expression signatures in snRNA-seq reference were displayed with heatmap plots (eighth column).

2. *The authors performed benchmark tests on multiple datasets, and the error rate in the main Fig. 2C is around 20%, these results are positive. But in b-d of Fig. S3, the error rate is as high as more than 50%, leading to this result may be because the ground truth may be inaccurate. I suggest that the authors can replace the dataset with better quality for benchmarking.*

Response:

We thank the reviewer for this constructive comment. We redesigned the comprehensive benchmarking study using six new datasets of improved quality (Fig. R15). We have summarized these datasets in Figure R1, and the simulation details are described in the section titled “Benchmarking Datasets”. Due to the execution of a new benchmarking study using six newly acquired datasets of enhanced quality, we have excluded Figure S3 from our revised manuscript. Within the revised manuscript, we have included the redesigned benchmarking as a dedicated section titled “A benchmarking study on cell type identification and gene expression decomposition” and provided details about simulation in the revised Supplementary Note Section 2.9.1.

3. *The authors describe that SpatialScope has the ability to integrate multiple slices. The authors used PASTE to obtain a 3D aligned ST data for cell type identification in SpatialScope. Lines 19-21 on page 10 “Alignment-based and deconvolution...”. I think Tangram and CytoSPACE are also applicable ST data for this 3D alignment. At the same time, the comparison of single and double slices lacks fairness.*

Response:

We would like to express our gratitude to the reviewer for raising this question. We acknowledge that the comparison between the methods (Tangram and CytoSPACE) and SpatialScope may appear unfair at first glance. However, it is important to note that the compared methods were designed to handle one slice at a time and cannot effectively utilize 3D spatial information. As a result, our evaluation was conducted using a single slice as input for these methods.

We understand the reviewer’s concern and recognize that it is technically feasible to force Tangram and CytoSPACE to utilize two slices by directly inputting the aligned 3D ST data. However, based on our understanding and empirical observations, the results obtained in this setting would be similar to those obtained in the single-slice setting. This is supported by the benchmarking results presented in Fig. R15b, where Tangram, CytoSPACE, and StarDist+RCTD achieved nearly identical cell type identification error rates regardless of whether single or multiple slices were used. In contrast, SpatialScope has the advantage of integrating multiple slices, which allows it to leverage 3D spatial information and further enhance cell type identification performance. This ability to leverage 3D spatial context is a unique feature of SpatialScope that sets it apart from the compared methods.

We appreciate the reviewer’s concern and have taken it into consideration. We hope this clarification provides a better understanding of the experimental design and the rationale behind our comparisons. Furthermore, we have included a comprehensive discussion of these results in the revised manuscript’s section titled “SpatialScope enables the integration of multiple slices and interpretation of cell-cell interactions by leveraging single-cell resolution gene expression profiles.”

4. *The model lacks a systematic hyperparameter comparative analysis. The authors provide a comparison of hyperparameter epochs (Fig. S7). But there are still many hyperparameter settings that will affect the performance of the model, such as LT and g_{ss} in the `Train_scRef.py` script. It would be great if the authors could add these hyperparameter comparisons. Of course, the hyperparameter evaluation of the system can also test the *SpatialScope*'s stability and robustness.*

Response:

We thank the reviewer for these constructive comments. One of the key innovations and strengths of *SpatialScope* lies in its utilization of a score-based generative model to accurately approximate the distribution of gene expressions from the scRNA-seq reference data. This unique approach sets *SpatialScope* apart from other methods and contributes to its superiority. We agree with the reviewer that the evaluation of the hyperparameters is very important for the score-based generative model. By systematically assessing these hyperparameters, we can gain insights into *SpatialScope*'s stability and robustness across various datasets and experimental conditions.

In the following, we have listed several key hyperparameters that we tested in our model, including the hyperparameters L , T , and g_{ss} that the reviewer specifically mentioned. These hyperparameters cover a wide range and are crucial for understanding the behavior and performance of our model. To facilitate understanding of the experimental results and our model, we provide intuitive explanations for the role each hyperparameter plays. This will help readers grasp the significance of the hyperparameters and interpret the experimental outcomes effectively. It is important to note that while we focused on evaluating these specific hyperparameters, there are other hyperparameters in the original score-based generative model that have been extensively discussed in the original paper [34]. Therefore, we have omitted those hyperparameters in our evaluation as they have already been addressed in the literature. Throughout our experiments, we utilize Dataset 1 in the benchmarking study. When testing each hyperparameter, we maintain the other hyperparameters at their default values in *SpatialScope*. This ensures a consistent and fair comparison across different hyperparameter settings.

Hyperparameters L and T

Intuitively, the score-based generative model learns data distribution by perturbing data with various levels of noise. In the training process, the perturbed data distributions with noise level σ_l is given as $p_{\sigma_l}(\mathbf{x}^{(l)}) = \int p(\mathbf{x})\mathcal{N}(\mathbf{x}^{(l)}|\mathbf{x}, \sigma_l^2\mathbf{I})d\mathbf{x}$, where $\mathbf{x}^{(l)}$ represents a sample perturbed by the noise level σ_l^2 , $\sigma_L > \sigma_{L-1} > \dots > \sigma_1 \approx 0$. In this process, we use the network $\mathbf{s}_\theta(\mathbf{x}^{(l,t)}, \sigma_l)$ to train the score of perturbed data distribution $\nabla_{\mathbf{x}^{(l,t)}} \log p_{\sigma_l}(\mathbf{x}^{(l,t)}|\mathbf{x})$. In the sampling process, Langevin dynamics is used to take samples from the learned data distribution. From $l = L$ to $l = 1$ we run:

$$\mathbf{x}^{(l,t+1)} = \mathbf{x}^{(l,t)} + \eta \mathbf{s}_\theta(\mathbf{x}^{(l,t)}, \sigma_l) + \sqrt{2\eta} \boldsymbol{\varepsilon}^{(l,t)}, \tag{R3}$$

where $\boldsymbol{\varepsilon}^{(l,t)} \sim \mathcal{N}(\mathbf{0}, \mathbf{I})$. For each noise level σ_l , we obtain samples $\mathbf{x}^{(l,t)}$ approximately follow the perturbed data distribution $p_{\sigma_l}(\mathbf{x}^{(l)})$. This process can also be called the denoising process because the noise level is progressively reduced, and perturbed data distribution gradually approximates target data distribution.

We can see from above that L represent how many noise level we set, and T is the number of steps

when computing (Fig. R3) at a specific noise level. Intuitively, the more extensive the grid of noise levels $\{\sigma_l\}_{l=1}^L$, the better for learning (i.e., the larger L , the better). For the sampling step T , similarly, the larger T , the better. However, the larger L and T mean more expensive computational resources. There is a trade-off between the performance and computational cost.

We used Dataset 1 in “Benchmarking datasets” to test the effect of L and T on model performance. We test $L = 10, 40, 80, 150, 232, 500$ and $T = 1, 3, 5, 10$ and evaluate SpatialScope’s performance on gene expression decomposition. The cosine similarities between the ground truth and decomposed single-cell level gene expression profiles under different L and T are calculated and compared (Fig. R19). We can see that SpatialScope works very well in a wide range of parameter settings. Therefore, we suggested the default setting of SpatialScope as $L = 232, T = 5$ according to the dimension of single-cell gene expression profiles.

Hyperparameter g_{ss}

When training the score-based generative model to approximate the distribution of gene expressions from the scRNA-seq reference data, we conduct evaluations at regular intervals of 500 epochs. During these evaluations, we generate samples from the model at the current checkpoint. The hyperparameter g_{ss} represents the sample size used during evaluation, indicating the number of samples we take from the model. In Figure R20, we present the sampling results obtained under different values of g_{ss} , using Dataset 1 in “Benchmarking datasets”. The UMAP plot illustrates the distribution of samples. The blue dots represent samples from the single-cell reference data, while the red dots represent generated samples from the trained model. The number of red dots corresponds to the value of g_{ss} . As depicted in the figure, increasing the value of g_{ss} results in a higher number of red dots. Regardless of the specific value of g_{ss} , the red dots consistently overlap with the blue dots, indicating that the model accurately approximates the distribution of gene expressions from the scRNA-seq reference data.

Figure R19: The evaluation of hyperparameters L and T . Bar plot of the cosine similarities between the ground truth and decomposed single-cell level gene expression profiles under different values of L and T . We use the same simulation data as in the main text. Different color represents different UMI subsample rate. The lower UMI subsample rate, the more difficult for gene expression decomposition task.

Figure R20: The evaluation of hyperparameters L and T . Bar plot of the cosine similarities between the ground truth and decomposed single-cell level gene expression profiles under different values of L and T . We use the same simulation data as in the main text. Different color represents different UMI subsample rate. The lower UMI subsample rate, the more difficult for gene expression decomposition task.

Hyperparameters σ_{yl}

The hyperparameter σ_{yl} shows in Algorithm 1 in the main text, and it is related to the the distribution we assign to the count-scale spot level gene expression profile $\mathbf{y}|\mathbf{x}_1, \mathbf{x}_2, \dots, \mathbf{x}_M$, where $\mathbf{x}_m, m = 1, 2, \dots, M$ represents the true count-scale gene expression levels of cells in the spot, and M is the number of cells in that spot. We assign Gaussian distribution to $\mathbf{y}|\mathbf{x}_1, \mathbf{x}_2, \dots, \mathbf{x}_M \sim \mathcal{N}\left(\sum_{m=1}^M \mathbf{x}_m, \sigma_{yl}^2\right)$. The hyperparameter σ_{yl}^2 is the variance of \mathbf{y} corresponding to the perturbed data distribution at the noise level σ_l . In the sampling process, σ_{yl} decreases as σ_l decreases. Formally, we set $\sigma_{yl} = \frac{power}{\sigma_l^2}$. We evaluate the gene expression decomposition accuracy under $power = 0.5, 0.8, 1.0, 1.3, 1.5, 1.8, 2.0, 2.3$, and 2.5 (Fig. R21). The performance is quite stable when

$power < 2$ and gets the best performance around 1.0. In the default setting of SpatialScope, we set $power = 1$ for all real data analysis.

We have included the sensitivity analysis of hyperparameters in the revised manuscript’s “Methods” section. Additionally, we have provided a detailed discussion of this analysis in the revised Supplementary Note Section 2.9.7.

Figure R21: The evaluation of hyperparameters σ_{yl} . Bar plot of the cosine similarities between the ground truth and decomposed single-cell level gene expression profiles under different values of $power$, where $\sigma_{yl} = \sigma_l^{\frac{power}{2}}$ and σ_{yl} shows in Algorithm 1 in the main text. Different color represents different UMI subsample rate. The lower the UMI subsample rate, the more difficult for gene expression decomposition task.

[Comments on methodological innovation]

5. *The first step of the model is nucleus segmentation on hematoxylin and eosin (H&E)-stained histological images to count the number of cells at each spot. As shown in lines 21 to 24 on page 3, this step is also an important step different from the method of analyzing the composition of cells in the point (such as Cell2location, SpatialDWLS, RCTD, etc.), but this step only uses the built-in function of squidpy (in demo Nuclei_Segmentation.py), and lacks innovations. This is more like building block stacking for SpatialScope. Although the authors used StarDist and Cellpose to compare the cell nucleus segmentation, and finally decided to use StarDist (Fig. S20). At present, many algorithms for cell nucleus segmentation have been developed, such as Baysor[35] and DeepCell[36], where Cellpose is upgraded to version 2.0 [37]. Does the author need to compare with these methods? Since it is an important step, is this overall embedding into SpatialScope considered an algorithmic innovation?*

Response:

We thank the reviewer for these questions. We would like to take this opportunity for clarification. As we summarized in Table 1, **the major methodological innovation of SpatialScope lies in the integration of ST data and single-cell reference data using the deep generative**

model, which is Step 3 (gene expression decomposition). We will elaborate more technical details in our response to Questions 6 and 7. With Step 3, SpatialScope has its unique advantage by providing transcriptome-wide expression levels at single-cell resolution. Step 1 (nucleus segmentation) and Step 2 (cell type identification) are auxiliary steps for SpatialScope to achieve the goal at Step 3.

We agree that nucleus segmentation is an important step for SpatialScope to obtain the number of cells at each spot. Nucleus segmentation is a long-standing yet important research topic in computer vision for the medical imaging field, and several publicly available tools exist [36; 38; 39; 40]. In our previous manuscript, we considered two most commonly used methods, StarDist (version 0.8.3) [38] and Cellpose (version 2.2) [39], and selected the one with better performance as a building block in SpatialScope.

We appreciate the reviewer for bringing up two recently developed methods, Baysor [35] and DeepCell [36]. Upon careful examination of the literature on Baysor and DeepCell, we found that they were primarily designed for cell segmentation on image-based single-molecule resolution ST data. More specifically, Baysor is a Bayesian mixture model-based method specifically tailored for distinguishing cell boundaries based on detected transcripts from image-based single-molecule resolution ST data (e.g., MERFISH, seqFISH, etc.) [35]. DeepCell is a deep learning-based method developed for cell detection, segmentation, and classification tasks [36]. DeepCell has been exclusively trained on single-channel nuclear image data, such as DAPI, making it challenging to adapt it to other types of image data [36].

These segmentation methods, namely StarDist, Cellpose, Baysor and DeepCell, are designed for different image data and different purposes. StarDist and Cellpose are designed for H&E images for nucleus segmentation, while Baysor and DeepCell are designed for DAPI images for cell segmentation. When applied to applicable data, all of them can be used to count the number of cells at each spot and thus can serve as the Step 1 building block of SpatialScope. Considering 10x Visium data with H&E images is widely used and was more discussed for the SpatialScope model, we conduct comprehensive evaluations of the compared methods for segmentation on H&E-stained images.

We utilized two benchmarking datasets that provide manually annotated ground truth nuclei: CoNSeP [40] and Kumar [41], which are based on H&E-stained images. The CoNSeP (Counting Nuclei in Synthetic Images) dataset is a publicly available dataset specifically designed for training and evaluating algorithms for nucleus detection and counting in microscopy images. It comprises 41 microscopy images obtained from UHCW, with a total of 24,319 annotated nuclei [40]. This dataset serves as a valuable resource for assessing the accuracy and effectiveness of nucleus segmentation methods. In addition to CoNSeP, we also employed the Kumar nucleus segmentation dataset, also known as the Kumar Dataset 2017. This dataset is widely recognized and utilized in the field of computer vision and image analysis. It consists of 30 microscopy images sourced from TCGA and provides a comprehensive collection of 21,623 annotated nuclei [41]. The Kumar dataset offers a diverse set of challenging images for evaluating the performance of nucleus segmentation methods.

We used two widely used metrics, Dice’s coefficient (DICE) and aggregated Jaccard index (AJI), to quantitatively evaluate the segmentation accuracy of the compared methods. These metrics provide objective measures of overlap and agreement between the segmented regions and the ground truth nuclei masks, with higher values indicating better performance. Figure R22 illustrates the results

obtained. StarDist exhibited the highest performance, achieving average DICE and AJI scores of 0.85 and 0.68, respectively. Cellpose also demonstrated comparable performance, with average DICE and AJI scores of 0.83 and 0.65, respectively. This aligns with our previous findings in the two 10X Visium H&E stained images, where Cellpose tended to miss more nuclei compared to StarDist (Fig. R27). In contrast, DeepCell’s performance was not good, as it was trained solely on DAPI images rather than H&E stained images. Furthermore, the conventional segmentation method, Watershed, exhibited inferior performance in nucleus segmentation for H&E stained images. To validate the robustness of our findings, we conducted similar evaluations using the Kumar benchmarking dataset (Fig. R23). The results were consistent, further affirming the reliability and robustness of StarDist as the best off-the-shelf nucleus segmentation tool for H&E-stained histological images.

In summary, based on our comprehensive analysis, StarDist emerges as the top-performing nucleus segmentation tool for H&E-stained histological images. Its superior performance, as indicated by high DICE and AJI scores, establishes its robustness and reliability in accurately segmenting nuclei in such images. When considering nucleus segmentation on H&E images, we choose StarDist as the building block of SpatialScope for cell counting. We have incorporated the discussion on different cell segmentation methods in the revised manuscript’s “Discussion” and “Methods” sections. Additionally, we have included relevant details in the revised Supplementary Note Section 2.1.

Figure R22: Comparison of nucleus segmentation performance among compared methods in CoNsep benchmarking dataset. a, H&E-stained histological images from CoNsep dataset. b, Manually annotated ground truth nuclei. c, Nucleus segmentation results by StarDist. d, Nucleus segmentation results by Cellpose. e, Nucleus segmentation results by DeepCell. f, Nucleus segmentation results by Watershed. g, DICE and AJI metrics.

Figure R23: Comparison of nucleus segmentation performance among compared methods in Kumar benchmarking dataset. a, H&E-stained histological images from Kumar dataset. **b,** Manually annotated ground truth nuclei. **c,** Nucleus segmentation results by StarDist. **d,** Nucleus segmentation results by Cellpose. **e,** Nucleus segmentation results by DeepCell. **f,** Nucleus segmentation results by Watershed. **g,** DICE and AJI metrics.

6. Lines 15-18 on page 32, "Because of the high...". Consistent with the first comment, using RCTD for nucleus segmentation when there is no histological image does not seem to be an algorithmic innovation either. It would be great if the author could further explain the innovation of the SpatialScope's model.

7. Also in the step of cell type identification, SpatialScope seems to be similar to the method of RCTD. On page 26, the author describes "Our model differs from RCTD in... spot i.", does SpatialScope only use the average expression of cell types for replacement, and the rest are the same as RCTD? In other words, will StarDist+RCTD get similar results to the article?

Response:

We appreciate reviewer's questions 6 and 7 regarding RCTD and our model's innovation, and we would like to take this opportunity to provide a detailed comparison between SpatialScope and RCTD. At the beginning of this response letter, we included a section titled "Comparison between RCTD and SpatialScope", which highlighted the advancements of SpatialScope over RCTD in terms of method utility, model, algorithm, and downstream applications (Table 1). Next, we give a detailed comparison between SpatialScope and StarDist+RCTD, and also our model's innovation with technical details.

In terms of cell type identification, SpatialScope incorporates spatial smoothness constraints into a discretized version of RCTD, whereas StarDist+RCTD does not utilize such constraints. Through comprehensive experimental results, we have demonstrated the advantages of employing spatial smoothness constraints in our response to Question 1 raised by Reviewer 1. Regarding gene expression decomposition, SpatialScope first learns the expression patterns associated with cell types from single-cell reference data. By using the learned expression distribution as the prior, it decomposes the total gene expression observed in each spot into contributions from individual cells. In contrast, StarDist+RCTD simply assigns the average expression of cell types to individual cells, which loses the cell-to-cell variation of the cells from the same cell type. In our benchmarking study, we have compared the accuracy of gene expression decomposition, as detailed in the response to Question 1 of Reviewer 2.

To further highlight the advantages of SpatialScope over StarDist+RCTD, here we add a real data example using a 3D mouse brain cortex dataset [42]. This dataset consists of two adjacent slices from the mouse brain cortex. We initially employed PASTE [5] to align the two adjacent tissue slices. Subsequently, we applied the nucleus segmentation step of SpatialScope to detect cell numbers and locations, successfully constructing a 3D-aligned ST dataset for the mouse brain cortex tissue (Fig. R24a). In this dataset, we identified a total of 3,777 and 3,034 cells within 812 and 794 spots from slice 1 and slice 2 of the brain cortex, respectively. Subsequently, we applied SpatialScope and StarDist+RCTD to the 3D-aligned ST data for cell type identification. To evaluate the accuracy of the inferred cell type labels, we relied on the known spatial organization of cell types within the brain cortex. Specifically, the mouse brain cortex comprises four main layers of glutamatergic neurons (L2/3, L4, L5, and L6) (Fig. R24a). Comparing the results, SpatialScope demonstrates a clearer layer structure compared to StarDist+RCTD (Fig. R24). Notably, StarDist+RCTD misidentifies L4 and L6 IT cells in other layers for both slices, while SpatialScope accurately identifies them within their corresponding layers. The high accuracy achieved by SpatialScope can be attributed to its

unique feature of incorporating spatial information through the incorporation of spatial smoothness constraints during the cell type identification step. SpatialScope effectively leverages the 3D spatial structure and borrows information from adjacent slices when applied to multi-slice data. In contrast, StarDist+RCTD simply discretizes the results of RCTD and randomly assigns cell type labels to cells within each spot.

Figure R24: Comparison of performance between SpatialScope and StarDist+RCTD on mouse brain cortex data. **a**, The SpatialScope identified cell type labels for the stacked 3D ST data constructed by two slices. **b,c**, Comparison cell type identification results (**b**: slice1 of mouse brain cortex; **c**: slice2 of mouse brain cortex) of SpatialScope using multiple-slice neighboring information (top) and StarDist+RCTD (bottom). The figure shows the scatter plot of spatial cell locations that are identified as L4 (the first column), L6 CT (the second column), and L6 IT (the third column). The circled area indicates regions of different layers: L4, L6 CT, and L6 IT.

Now, we will delve into further details from the perspective of methodological innovation. **The novelty of SpatialScope mainly comes from the gene expression decomposition step.** In this step, SpatialScope first learns the expression patterns of different cell types from single-cell reference data as the prior distribution. By combining the prior information with the likelihood term of the observed ST data, SpatialScope then formulates a posterior sampling to perform gene expression decomposition via the Langevin dynamics. As briefly mentioned in the “Comparison between RCTD and SpatialScope” section, the major challenge comes from learning the conditional score function $\mathbf{s}_\theta(\mathbf{x}^{(l)}, \sigma_l, \mu_k)$ and how to use $\mathbf{s}_\theta(\mathbf{x}^{(l)}, \sigma_l, \mu_k)$ to conduct posterior sampling via the Langevin dynamics.

- Learning the conditional score function $\mathbf{s}_\theta(\mathbf{x}^{(l)}, \sigma_l, \mu_k)$.

The major challenge in designing the conditional score function $\mathbf{s}_\theta(\mathbf{x}^{(l)}, \sigma_l, \boldsymbol{\mu}_k)$ is how to incorporate cell type information by utilizing the mean expression level of cell type k as input. We have innovatively devised a conditional UNet architecture [43] specifically for this purpose. To integrate cell type information, we have added another UNet that takes $\boldsymbol{\mu}_k$ as input (Fig. R25, “Conditional part”). Consequently, there are two UNets in total: one takes \mathbf{x} as input, while the other takes $\boldsymbol{\mu}_k$ as input. Both UNets comprise smaller network blocks, which are further composed of convolutional networks (Fig. R26). Within each UNet, three main blocks (MBlock) or conditional blocks (CBlock) are utilized to gradually downsample the dimensionality of the gene expression vector by factors of 3, 4, 5, with the number of channels being 128, 256, and 512, respectively. Subsequently, another set of three MBlocks or CBlocks, symmetric to the downsampling process, is employed to gradually upsample the dimensionality of the gene expression vector. The downsampling and upsampling processes are connected by additional MBlocks or CBlocks that do not involve downsampling or upsampling, resulting in a U-shaped architecture.

To incorporate the information learned by the conditional UNet, we utilize the feature-wise linear modulation (FiLM) module [44] (Fig. R25, R26c). This module generates feature-wise affine parameters, namely the scale vector \mathbf{W} and the bias vector \mathbf{b} , by taking the middle output of the conditional UNet as input (Fig. R25). Specifically, it takes the middle layer output from the CBlock, denoted as $\boldsymbol{\mu}_{k,mid}$, and produces $\mathbf{W}(\boldsymbol{\mu}_{k,mid})$ and $\mathbf{b}(\boldsymbol{\mu}_{k,mid})$ as functions of $\boldsymbol{\mu}_{k,mid}$. These scale and bias vectors are then applied to the intermediate results of the UNet that takes \mathbf{x} as input. Formally,

$$\mathbf{x}_{mid} = \mathbf{W}(\boldsymbol{\mu}_{k,mid}) \odot \mathbf{x}_{mid} + \mathbf{b}(\boldsymbol{\mu}_{k,mid}), \quad (\text{R4})$$

where \mathbf{x}_{mid} is the corresponding middle layer output from MBlock.

Figure R25: Architecture of the conditional score network $\mathbf{s}_\theta(\mathbf{x}^{(l)}, \sigma_l, \boldsymbol{\mu}_k)$. The red line represents the UNet that takes \mathbf{x} as input, while the green line represents the conditional UNet that takes $\boldsymbol{\mu}_k$ as input, where the subscript k denotes the cell type to which \mathbf{x} belongs. We incorporate cell type information into the score function using the FiLM module, which generates scale and bias vectors for feature-wise affine transformation, as depicted in Eq. (R4).

Figure R26: The diagrams of MBlock, CBlock and FiLM. **a**, The diagram of MBlock. Two residual blocks are used. The output of the FiLM block will be the input to the Feature-wise Affine block, as shown in Eq. (R4). **b**, The diagram of CBlock. One residual block is used, and this block’s output will be the FiLM block’s input (Fig. R26c). **c**, The diagram of FiLM. The block will take the output of CBlock as input and output scale \mathbf{W} and shift \mathbf{b} .

- Posterior sampling via the Langevin dynamics

Suppose that a spot has M cells and denote \mathbf{x}_i as the expression vector of the i -th cell at a given spot, $i = 1, \dots, M$. Based on Langevin dynamics [45; 46], we can obtain the decomposition by sampling $\mathbf{X} = [\mathbf{x}_1, \mathbf{x}_2, \dots, \mathbf{x}_M]$ from the posterior distribution $p(\mathbf{X}|\mathbf{y}, k_1, k_2, \dots, k_M)$,

$$\mathbf{X}^{(t+1)} = \mathbf{X}^{(t)} + \eta \nabla_{\mathbf{X}} \log p(\mathbf{X}^{(t)} | \mathbf{y}, k_1, k_2, \dots, k_M) + \sqrt{2\eta} \boldsymbol{\varepsilon}^{(t)}, \quad (\text{R5})$$

Where k_i denotes the cell type of the i th cell, $\boldsymbol{\varepsilon}^{(t)} \sim \mathcal{N}(0, I)$ and $\eta > 0$ is the step size, $t = 1, \dots, \infty$. By Bayes rule, we have

$$\begin{aligned} \nabla_{\mathbf{X}} \log p(\mathbf{X}^{(t)} | \mathbf{y}, k_1, k_2, \dots, k_M) &= \nabla_{\mathbf{X}} \log p(\mathbf{y} | \mathbf{X}^{(t)}, k_1, k_2, \dots, k_M) \\ &+ \sum_{i=1}^M \nabla_{\mathbf{x}_i} \log p(\mathbf{x}_i^{(t)} | k_i) - \nabla_{\mathbf{X}} \log p(\mathbf{y} | k_1, k_2, \dots, k_M). \end{aligned} \quad (\text{R6})$$

Noting that $\nabla_{\mathbf{X}} \log p(\mathbf{y} | k_1, k_2) = 0$ and we have learned the prior term $\nabla_{\mathbf{x}_i} \log p(\mathbf{x}_i^{(t)} | k_i)$ from scRNA-seq data, using the conditional score function $\mathbf{s}_{\theta}(\mathbf{x}, \sigma_l, \boldsymbol{\mu}_k)$. Therefore, we can conduct posterior sampling via the Langevin dynamics through

$$\mathbf{X}^{(l,t+1)} = \mathbf{X}^{(l,t)} + \eta \left[\nabla_{\mathbf{x}_i} \log p(\mathbf{y}_i | \mathbf{X}_i^{(l,t)}) + \begin{bmatrix} \mathbf{s}_\theta(\mathbf{x}_{i,1}^{(l,t)}, \sigma_l, \boldsymbol{\mu}_{k_{i,1}}) \\ \vdots \\ \mathbf{s}_\theta(\mathbf{x}_{i,M_i}^{(l,t)}, \sigma_l, \boldsymbol{\mu}_{k_{i,M_i}}) \end{bmatrix} \right] + \sqrt{2\eta} \boldsymbol{\varepsilon}^{(l,t)}, \quad (\text{R7})$$

The samples from the posterior distribution $p(\mathbf{X}^{(t)} | \mathbf{y}, k_1, k_2)$ recover gene expression levels of the two cells, achieving single-cell resolution.

The key distinction between SpatialScope and StarDist+RCTD for gene expression decomposition lies in how they infer the expression levels of individual cells. SpatialScope utilizes a posterior distribution that incorporates prior information learned from single-cell reference data. This approach ensures that the inferred cellular gene expression aligns with the observed spot-level gene expression in spatial context. In contrast, StarDist+RCTD assigns the average gene expression of a particular cell type to all cells inferred to belong to that cell type, losing the information of cell-to-cell variation within a cell type. In addition, RCTD does not consider the spatial smoothness of cell type labels.

We have included a detailed comparison between SpatialScope and RCTD in a dedicated section within the Methods section of the revised manuscript, as well as in Supplementary Note Section 2.8. Furthermore, we have incorporated StarDist+RCTD as a baseline method in both the benchmarking study and the analysis of multiple slice real data in the revised manuscript. The design and implementation details of the baseline method, StarDist+RCTD, are described comprehensively in the revised Supplementary Note Section 2.9.3.

[Comments on writing]

1. *On page 21, lines 18-22, “As shown in Fig. 6a... (Fig. 6a)”. “Fig. 6a” is quoted repeatedly.*

Response:

Thank you for the comments. We have revised the sentence as follows:

“SpatialScope successfully reconstructed the known spatial organization of cell types in the MOp of the brain cortex (Fig. 6a). Specifically, glutamatergic neuronal cells showed distinct cortical layer patterns, while GABAergic neurons and most non-neuronal cells were granularly distributed.”

2. *Fig. S9c, Fig. S19b, the paper is not cited.*

Response:

Thank you for the comments. We have cited Fig. S9c (Fig. S20c in the revised manuscript) in the revised manuscript page 14 in the following sentence:

“Furthermore, we demonstrate that the spatially resolved transcriptomic data at single-cell resolution, generated by SpatialScope with the aid of 3D alignment, allowed us to infer reliable spatially proximal cell-cell communications (Fig. 3e, Fig. S20c).”

We have cited Fig. S19b (Fig. S33b in the revised manuscript) in the revised manuscript page 25 in the following sentence:

“As expected, compared to 63 cell-type specific spatially DE genes detected in MERFISH genes under an FDR of 1% (Fig. 6e), the number of significant genes with FDR < 1% increases to 293 by incorporating the imputed Non-MERFISH genes (Fig. 6f, Fig. S33b).”

3. *Line 21 on page 7 and line 18 on page 9, colons should be changed to full stops.*

Response:

Thanks for the comments. Initially, we introduced the construction of a simulation dataset in the sentence located at line 21 on page 7. The sentence originally at line 21 on page 7 is as follows:

“Since ground truth cell type of individual cells are unknown in real low-resolution ST data, we used a single-cell resolution MERFISH dataset [25] to generate a simulated dataset: The MERFISH data is partitioned into two regions, with one region serving as the single-cell reference data, and the second region serving to mimic low-resolution ST data by aggregating the cells on uniform grids to simulate spots (Fig. S1).”

As we have reorganized the manuscript and incorporated the redesigned benchmarking in the revised version, the sentence has been removed. Instead, detailed information regarding the simulation design can be found in the revised Supplementary Note Section 2.9.1.

The sentence on line 18 on page 9 is the caption of Figure 2f:

“Circled area indicates regions of different layers: L4, L6 CT and L6 IT.”

Since we have reorganized Figure 2 in the revised manuscript, the sentence has been removed.

4. *“As some cells...” on page 12 is missing references to figures. Although detailed in this section “supplementary note section 2.7.3”, it is better to add supplementary figures.*

Response:

Thank you for the comments. The original sentence is as follows:

“As some cells with weak signals may be missed in the nucleus segmentation step, we also evaluated the performance of SpatialScope and the compared method when the ground truth cell number is inconsistent with the estimated cell number in the spots. In brief, we observed that SpatialScope is robust over inconsistent cell numbers and capable of identifying the remaining ground truth cells with highly matched transcriptional profiles (supplementary note section 2.7.3).”

We have revised the sentence and cited the corresponding figures on page 11, line 11 as follows:

“Furthermore, considering that the nucleus segmentation step may miss some cells with weak signals, we also evaluated the performance of SpatialScope and the compared methods when the estimated cell number in the spots did not match the ground truth cell number (see supplementary note section 2.9.6). We observed that SpatialScope demonstrated robustness in handling inconsistent cell numbers and was able to accurately identify the remaining ground truth cells with highly matched transcriptional profiles (Fig. S50-S52).”

5. *Lines 2-13 on page 18 describe the RTN4-LINGO intercellular interaction in detail. It is recommended that you display this part of the results in Fig.4.*

Response:

Thanks for the comments. We have incorporated the visualization of molecular interactions between the ligand-receptor pair *RTN4-LINGO* in Figure 4 of the revised manuscript.

[Comments on implementation and example codes]

1. *Python package should use standard python packaging facilities. It would be preferable if setuptools or poetry or similar would be used.*

Response:

Thank the reviewer for this comment. In our original instructions for installation of SpatialScope, we provided a guide for setting up a conda environment with python packages required by SpatialScope. Previously, we successfully tested the installation on a Linux system without SpatialScope installed before. In response to the reviewer's suggestion, we further added "setup.py" to the software website at Github (<https://github.com/YangLabHKUST/SpatialScope>). User can run "python setup.py develop" or "python setup.py install" to install the SpatialScope module with the Distutils now.

We have successfully set up the environmental and installed the SpatialScope on a Linux server with the following new installation instructions:

```
$ git clone https://github.com/YangLabHKUST/SpatialScope.git
$ cd SpatialScope
$ conda env create -f environment.yml
$ conda activate SpatialScope
$ python setup.py develop
```

After installation, we can check the installation status by running the following modules in the command line:

```
$ Cell_Type_Identification.py -h
$ Decomposition.py -h
```

2. *The list of required dependencies in the readme is incomplete. In addition to what is described in the README, the following package must be installed: scikit-misc.*

Response:

Thank the reviewer for this comment. We have added scikit-misc in "environment.yml". The modified "environment.yml" contains all python packages required by SpatialScope:

```
name: SpatialScope
channels:
  - conda-forge
  - bioconda
  - pytorch
  - anaconda
dependencies:
  - python=3.9
  - scanpy=1.9.1
```

```
- squidpy=1.2.2
- stardist=0.8.3
- tensorflow=2.10.0
- qpsolvers=2.4.0
- pytorch=1.12.1
- pandas=1.4.2
- matplotlib=3.5.2
- anndata
- ipykernel
- scipy
- tqdm
- scikit-learn
- scikit-misc
- pip
- pip:
  - ray==2.0.0
```

3. *There is a slight omission in the description of the sample code. The link to the original dataset in the demo `Human-Heart.ipynb` does not exist.*

Response:

Thank the reviewer for this comment. After double checking the dataset link in “Human-Heart.ipynb”, we found that we forgot to add “https:/" before the dataset link “www.heartcellatlas.org”. We have fixed this bug in the latest version of software in Github.

4. *An error is reported when running the sample code. As a result, errors were reported in the “Learning the gene expression distribution of scRNA-seq reference using score-based model” step and “Step3: Gene expression decomposition”, and some results of the article could not be reproduced. The error code is as follows:*

```
from utils import configure_logging, ConfigWrapper
ImportError: cannot import name ConfigWrapper from utils
```

Response:

We appreciate the reviewer for bringing this issue to our attention. Upon careful examination of the git logs, we have identified that certain updates made to the `utils.py` file resulted in the removal of the “ConfigWrapper” class. We apologize for this oversight.

However, we want to assure the reviewer that we have rectified this bug in the most recent version of the software available on GitHub. The “ConfigWrapper” class has been reinstated, and users can now access it without any issues.

Responses to Reviewer 3's comments:

[Major comments]

1. *Recently, there are more and more super high resolution ST technologies, for example, Stereo-seq [PMID: 35512705] can provide spot with diameter of 220 nm, which is much smaller than the size of single-cell. Is it still necessary to focus on deconvolution for ST?*

Response:

We thank the reviewer for raising these important questions. We fully agree that there is a growing number of super high-resolution ST technologies emerging, such as Stereo-seq, which has demonstrated remarkable advancements in the field. Stereo-seq utilizes DNA nanoball (DNB) sequencing technology [47] and achieves an impressive resolution of up to 500 nm. This high resolution enables the capture of transcripts at the subcellular level, providing detailed spatial information. By combining aggregated DNB of different ranges, Stereo-seq can generate ST data at various levels of resolution, offering flexibility in capturing spatial transcriptomics information [13]. Stereo-seq has been successfully applied in numerous research studies related to biological growth and development [13; 48], as well as in investigations of tumors [49] and cancer [50]. These applications have contributed significantly to the advancement of life science, providing valuable insights into cellular behaviors, tissue organization, and disease mechanisms.

As a novel technology, Stereo-seq offers several advantages; however, it also presents new challenges that require innovative solutions. Some of these challenges can be addressed by leveraging the idea of SpatialScope. First, Stereo-seq encounters the issue of RNA diffusion, where RNA transcripts diffuse laterally to neighboring areas [48]. The high resolution of Stereo-seq exacerbates this problem, necessitating careful consideration due to the potential for inaccurate downstream analysis. Second, Stereo-seq faces limitations in capture rate due to its high resolution. The capture of unique molecular identifier (UMI) counts is affected by the resolution, with an average range of 69 per 2 mm (diameter) bin (bin 3, 3×3 DNB) and 1,450 per 10 mm (diameter) bin (bin 14, 14×14 DNB, equivalent to 1 medium-sized cell) [48]. The low capture rate at high resolution poses challenges for data analysis, even for tasks like clustering analysis. As a result, aggregation of DNB from larger regions becomes necessary to achieve acceptable data quality [13; 51]. Although Stereo-seq can aggregate transcripts within the range of approximately a single cell, the capture rate still falls short of being ideal [48]. From this perspective, the need for aggregation restricts the resolution of Stereo-seq to achieving single-cell level resolution, which aligns with the capabilities of SpatialScope. Third, how to aggregate sub-cellular ST data into the cellular level remains to be challenging. Stereo-seq defines each nucleus and its surrounding area as a cell region, aggregating transcripts within this region to construct the gene expression profile of a single cell, thereby achieving single-cell resolution. However, cell morphology varies significantly across different tissues. For example, nerve cells and muscle cells exhibit more elongated shapes compared to cells from normal tissue. Failing to consider cell morphology during transcript aggregation may introduce substantial errors in downstream analysis. Addressing this challenge necessitates the development of methodologies that account for variations in cell morphology during the aggregation process.

The field of ST comprises a diverse range of technologies, each with its own set of advantages

and challenges. Visium technology has gained considerable maturity over the years, becoming a well-established commercially available method in the field of ST. According to the database collected by the museum of spatial transcriptomic project [14], more than half of studies in the past year still utilized Visium technique to quantify gene expression in space, accumulating a substantial amount of data. Even presently, new Visium data continues to emerge, further enriching the existing knowledge base. In particular, when comparing Visium technology with high-resolution Stereo-seq technology, it becomes evident that Visium is a more established and widely adopted approach, offering distinct benefits such as a higher capture rate and easier access.

While SpatialScope may not be universally applicable to all ST technologies, it can provide valuable assistance across a wide range of platforms. This includes sequence-based ST technologies like Visium and Slide-seq, as well as image-based technologies such as MERFISH. Moreover, even for Stereo-seq data, SpatialScope can still offer benefits. As previously mentioned, Stereo-seq has certain limitations, one of which is its restricted capture rate, although it has shown improvements compared to some other technologies [13]. To address this challenge, SpatialScope can be utilized by integrating single-cell reference data with deep generative models. The effectiveness of this approach has been demonstrated through the analysis of Slide-seq data using SpatialScope.

2. SpatialScope assumes that neighboring cells are more likely to belong to the same cell type, which is not always true [for example, <https://doi.org/10.1038/s43588-022-00266-5>] and should be discussed carefully.

Response:

We thank the reviewer for this insightful comment. We acknowledge that the assumption of neighboring cells belonging to the same cell type may not always hold true. In our SpatialScope model, we have addressed this limitation by incorporating spatial information using the Potts model. However, we agree that a more effective approach would be to allow the model to adaptively learn from the data and determine the similarity of cell types among neighboring cells. This can be achieved by using spatial information as a key input for Deep Graph Infomax (DGI) framework, like the model in the paper of the reviewer mentioned (<https://doi.org/10.1038/s43588-022-00266-5>), or by adopting GCN with attention mechanism to adaptively learn the similarity of neighboring spots/cells [52].

Although the Potts model may not be the optimal approach for incorporating spatial information, it still can improve accuracy. In the SpatialScope model, we introduce a hyperparameter, denoted as ν , which controls the smoothness of cell type labels. A larger value of ν leads to smoother cell type labels. By varying the value of ν , we conduct experiments to assess the impact of incorporating spatial information on four single-slice datasets (Fig. R1, details of benchmarking datasets are described in the section titled “Benchmarking Datasets” at the beginning of this response letter). We compared the results with the RCTD model, as shown in Figures R3, R4, and R5. Further details of these experiments can be found in the response to Question 1 of Reviewer 1. The experimental results demonstrate that the error rate of cell type identification at single-cell resolution initially decreases and then increases with an increase in the value of ν . This finding suggests that incorporating a small amount of spatial information (small ν) can enhance the performance of cell type identification. Therefore, we set $\nu = 10$ as the default setting in our software to avoid fine-tuning this smoothness

hyperparameter in real data analysis.

We have included a discussion on the validity of the smoothness constraint assumption and potential directions for its improvement in the “Discussion” section of the revised manuscript.

3. *In Fig. S3, SpatialScope can not beat RCTD method. In addition, the core cell type identification part, spatialScope also follows modelling way very similar to RCTD. The improvement of modeling, algorithm and performance over RCTD should be discussed comprehensively.*

Response:

We thank the reviewer to raise this comment. Reviewers 1 and 2 also asked related questions. We apologize that we didn’t clearly explain the relationship between SpatialScope and RCTD in the previous version. We have prepared a section titled “comparison between SpatialScope and RCTD” and Table 1 for clarification. In a nutshell, compared to RCTD which only provides the proportion of cell types for each spot, SpatialScope refines the spatial transcriptomic landscape at single-cell resolution for both seq-based and image-based ST data, enabling fine-grained cell gradients to be clearly visualized and the detection of spatially resolved cellular interactions/communications.

Besides, as suggested by Reviewer 2, we redesigned the comprehensive benchmarking study using six new datasets with better quality. We have summarized these datasets in Figure R1, and the simulation details are described in the section titled “Benchmarking Datasets”. We have compared the performance between SpatialScope and RCTD on six datasets in the benchmarking study (Fig. R15), as detailed in the response to Question 1 of Reviewer 2.

We have included a comparison between SpatialScope and RCTD as a dedicated section within the Methods section of the revised manuscript, as well as in Supplementary Note Section 2.8. Additionally, in the revised manuscript, we have included a dedicated section titled “A benchmarking study on cell type identification and gene expression decomposition” that describes the execution of a new benchmarking study using six newly acquired datasets of enhanced quality. As a result of this new benchmarking study, we have excluded Figure S3 from our revised manuscript.

4. *Nucleus segmentation is the very beginning of SpatialScope and is of vital importance because it provides cell location and number within one spot. More analysis is required for the accuracy and performance of StarDist.*

Response:

We thank the reviewer for this constructive comment. As a fundamental step of SpatialScope, nucleus segmentation plays a critical role in our framework. We agree that it is necessary to evaluate the performance of StarDist as well as other methods. We considered two criteria to evaluate the accuracy and performance of StarDist:

Evaluation with benchmarking datasets To comprehensively compare the nucleus segmentation performance of StarDist and all other included methods, we utilized two H&E-stained image-based nucleus segmentation benchmarking datasets with manually annotated ground truth nuclei: CoNSeP [40] and Kumar [41]. As shown in Figure R22 and Figure R23, StarDist achieved the best performance in both benchmarking datasets, suggesting the robustness and reliability of StarDist

for H&E images. The detailed benchmarking settings and results are described and summarized in the response to Question 5 of Reviewer 2. Please refer to the response for more details.

Evaluation with 10X Visium real datasets We further evaluated the compared methods in two 10x Visium datasets with H&E-stained histological images (Fig. R27). Since the 10X Visium data lack manually annotated ground truth nuclei, we assessed the accuracy of segmentation results by comparing segmented region and visible nuclei in images through naked eye. In the first 10x human heart data, StarDist located 1,797 single cells while Cellpose only found 1,301 cells. Clearly, Cellpose performed worse as a result of substantial missing cells, especially in the zoom-in region. For the second 10x mouse brain cortex dataset, we observed similar results that StarDist ($n=1,563$) segments more cells than Cellpose ($n=1,250$). In summary, our evaluation demonstrates that StarDist is currently the best off-the-shelf nucleus segmentation tool for H&E-stained histological images.

We have incorporated the discussion about the accuracy and performance of StarDist in the revised manuscript's "Methods" sections. Additionally, we have included relevant details in the revised Supplementary Note Section 2.1.

Figure R27: Comparison of nucleus segmentation performance between StarDist and Cellpose in two 10x Visium datasets. We applied Squidpy, which provides the interface of StarDist and Cellpose, to segment nuclei in the pair HE images. We used the default parameters following the instruction (<https://squidpy.readthedocs.io/en/stable/index.html>). In the first 10x human heart data, the H&E-stained histological image (first column) was used as input. The segmentation results of StarDist and Cellpose were shown in the second and last column, respectively, where StarDist located 1,797 single cells while Cellpose only found 1,301 cells. Clearly, Cellpose performed worse as a result of substantial missing cells, especially in the zoom-in region. For the second 10x mouse brain cortex dataset, we observed similar results that StarDist ($n=1,563$) segments more cells than Cellpose ($n=1,250$).

5. *Fig. 3a* assumes that the spot only contains two cells (seems not quite reasonable), and *Fig. S27*

shows result of spot with up to 5 cells. I am wondering if there is any analysis on how many cells are contained within one spot, which is also relevant to my question #4.

Response:

We appreciate the reviewer's comment and would like to provide clarification on the matter. In our previous manuscript, Figure 3a was included as an illustrative example to visually demonstrate a spot containing two cells. The purpose was to aid in visualization and not to assume that spots only contain two cells in simulation or Visium data. It is important to note that this assumption of spots containing two cells was specifically made for slide-seq data. This assumption was based on the matching spot size ($10\ \mu\text{m}$) with the size of a single cell and the unavailability of paired histological images for slide-seq. However, for the simulation dataset (Dataset 1: MERFISH MOp) depicted in Figure 3a, the number of cells within each spot actually ranges from 1 to 5, considering a grid size of $34 \times 30\ \mu\text{m}$. This explains why Figure S27 in the previous manuscript was able to display results for spots with up to 5 cells.

In response to the reviewer's suggestion, we have included Figure R28, which illustrates the distribution of cell numbers within the spots. This figure provides a visual representation of the variability in the number of cells found within each spot. For the first dataset, the 10x Visium human heart data, we observed that the number of cells within the spots ranged from 1 to 12. Notably, the spots located in the vascular region at the center tended to exhibit a higher density of nuclei compared to other regions. Regarding the second dataset, the 10x Visium mouse brain cortex dataset, we found that the number of cells within the spots reached as high as 16 and 14 in slice 1 and slice 2, respectively.

In summary, we would like to emphasize that there are no constraints on the number of cells in simulation or Visium data, especially when dealing with larger spot sizes. The assumption that spots contain at most two cells was specifically made for Slide-seq data, which provides nearly single-cell resolution.

Figure R28: The distribution of cell numbers within the spots in two 10X Visium datasets. a, 10X Visium human heart data. **b,** 10X Visium mouse brain cortex slice 1 data. **c,** 10X Visium mouse brain cortex slice 2 data.

6. How to determine the exact position of each deconvoluted cells within one spot? Subspot resolution methods, such as BayesSpace (PMID: 34083791) should be compared with.

Response:

We thank the reviewer for raising this important question and suggestion. Instead of inferring cell type proportion in spots and then discretizing to reach single-cell resolution, we first apply an existing nucleus segmentation method (StarDist [38]) on hematoxylin and eosin (H&E)-stained histological images to count the number of nuclei in spots and also the position of each cell (Step1: “Nucleus segmentation”), and then use a statistical method to infer the cell type label at each cell locations (Step2: “Cell type identification”). In other words, we get cell locations from Step 1, which is the direct output of the nucleus segmentation method StarDist. It is worth noting that we use “cell nuclei” counts and locations to approximate cell counts in the spots and cell locations, respectively, which is reasonable since we do not need to use the information of cell morphology information in our model.

To address the reviewer’s concern, we qualitatively add the comparison between SpatialScope and BayesSpace using Dataset 1 to Dataset 4 in benchmarking datasets (Fig. R1, Fig. R29). Although SpatialScope and BayesSpace can accurately identify spatial domains, they have different utilities and methodologies. SpatialScope creates a spatial map of cell type at single-cell resolution. It integrates spatial transcriptomics data and single-cell RNA-seq to infer cell type labels at each cell location, resulting in specific cell type labels for each cell (e.g., astrocytes, endothelial cells) (Fig. R29)). These features are in direct contrast to BayesSpace, which performs clustering analysis using spatial transcriptomics data only and enhances the data to a resolution of six or nine times higher than the original (Fig. R29)). Noted that we couldn’t match the color in the BayesSpace’s results to the cell type colors because BayesSpace performs clustering analysis and cannot specifically identify the clusters’ cell type.

Dataset 1

Dataset 2

Dataset 3

Dataset 4

Figure R29: Comparison of the results of SpatialScope “Cell type identification” and BaysSpace on Datasets 1, 2, 3, and 4. (a, left, b,c,d, upper) Spatial scatter plots display identified single-cell types on each cell location from ground truth and SpatialScope’s “Cell type identification” step. (a, right, b,c,d, lower), Clustering results of BaysSpace. Left and right are BaysSpace’s spot-level clustering results and enhanced clustering, respectively. Different color represents different cell types and clustering, respectively.

7. *What if ST data lacks histological images? The assumption that at most two cell types co-exist within a spot is too strong now. More data or analysis should be provided to support it.*

Response:

We thank the reviewer for this comment. Firstly, it is important to note that several widely used ST technologies, such as Visium, Stereoseq, and Slide-lock, currently provide paired histological images. Furthermore, the availability of paired histological images is expected to increase in the future as these techniques become more accessible and cost-effective. However, we would like to clarify that in the specific case of the Slide-seq technique, where no paired histological images are currently available. As a result, we made the assumption that at most two cells coexist within a

spot in Slide-seq data. This assumption is based on the fact that the spot size in Slide-seq ($10\ \mu\text{m}$) already matches the size of a single cell. In comparison, Visium data typically detect 1-10 cells within each spot, given the larger spot size of $55\ \mu\text{m}$. Considering that the area of a Slide-seq spot is only 1/30 of a Visium spot, it is reasonable to assume a maximum of two cells within a Slide-seq spot. We hope this clarification provides a better understanding of the rationale behind our assumption and the context in which it was made.

Secondly, even in situations where ST data lacks paired histological images, there exist alternative approaches to estimate the number of cells within each spot. In the case of Slide-seq data, the initial step of nucleus segmentation can be replaced with singlet/doublet classification. Based on the assumption that at most two cells coexist within a spot, we can perform singlet/doublet classification using the estimated cell type compositions. If a Slide-seq spot exhibits two dominant cell type compositions, confidently classifying it as a doublet becomes feasible. Similarly, for Visium data where histological images are absent due to experimental failure or other reasons, it is possible to devise alternative methods to estimate cell numbers by considering both cell type compositions and the total number of UMIs within the spots. Spots with higher UMIs and more diverse cell type compositions are likely to contain a greater number of cells. These alternative strategies allow for the estimation of cell numbers within spots, even in situations where paired histological images are unavailable. By leveraging information about cell type compositions and UMIs, valuable insights can be gained regarding the cell composition within ST datasets.

We have included a discussion in the “Discussion” section of the revised manuscript regarding the impact of the lack of paired histology images on the model.

[Minor suggestions]

1. *It would be better to move the complex equations (1) (2) to Methods part.*

Response:

We thank the reviewer for the comment. We have tried our best to reorganize the manuscript to make it more clear and structured.

References

- [1] Bin Li, Wen Zhang, Chuang Guo, Hao Xu, Longfei Li, Minghao Fang, Yinlei Hu, Xinye Zhang, Xinfeng Yao, Meifang Tang, et al. Benchmarking spatial and single-cell transcriptomics integration methods for transcript distribution prediction and cell type deconvolution. *Nature Methods*, pages 1–9, 2022.
- [2] Meng Zhang, Stephen W Eichhorn, Brian Zingg, Zizhen Yao, Kaelan Cotter, Hongkui Zeng, Hongwei Dong, and Xiaowei Zhuang. Spatially resolved cell atlas of the mouse primary motor cortex by merfish. *Nature*, 598(7879):137–143, 2021.
- [3] William E Allen, Timothy R Blosser, Zuri A Sullivan, Catherine Dulac, and Xiaowei Zhuang. Molecular and spatial signatures of mouse brain aging at single-cell resolution. *Cell*, 186(1):194–208, 2023.
- [4] Hu Zeng, Jiahao Huang, Haowen Zhou, William J Meilandt, Borislav Dejanovic, Yiming Zhou, Christopher J Bohlen, Seung-Hye Lee, Jingyi Ren, Albert Liu, et al. Integrative in situ mapping of single-cell transcriptional states and tissue histopathology in a mouse model of alzheimer’s disease. *Nature Neuroscience*, 26(3):430–446, 2023.
- [5] Ron Zeira, Max Land, Alexander Strzalkowski, and Benjamin J Raphael. Alignment and integration of spatial transcriptomics data. *Nature Methods*, 19(5):567–575, 2022.
- [6] Tommaso Biancalani, Gabriele Scalia, Lorenzo Buffoni, Raghav Avasthi, Ziqing Lu, Aman Sanger, Neriman Tokcan, Charles R Vanderburg, Åsa Segerstolpe, Meng Zhang, et al. Deep learning and alignment of spatially resolved single-cell transcriptomes with tangram. *Nature methods*, 18(11):1352–1362, 2021.
- [7] Jiaqiang Zhu, Shiquan Sun, and Xiang Zhou. Spark-x: non-parametric modeling enables scalable and robust detection of spatial expression patterns for large spatial transcriptomic studies. *Genome biology*, 22(1):1–25, 2021.
- [8] Congxue Hu, Tengyue Li, Yingqi Xu, Xinxin Zhang, Feng Li, Jing Bai, Jing Chen, Wenqi Jiang, Kaiyue Yang, Qi Ou, et al. Cellmarker 2.0: an updated database of manually curated cell markers in human/mouse and web tools based on scrna-seq data. *Nucleic Acids Research*, 51(D1):D870–D876, 2023.
- [9] Ed S Lein, Michael J Hawrylycz, Nancy Ao, Mikael Ayres, Amy Bensinger, Amy Bernard, Andrew F Boe, Mark S Boguski, Kevin S Brockway, Emi J Byrnes, et al. Genome-wide atlas of gene expression in the adult mouse brain. *Nature*, 445(7124):168–176, 2007.
- [10] Bosiljka Tasic, Zizhen Yao, Lucas T Graybuck, Kimberly A Smith, Thuc Nghi Nguyen, Darren Bertagnolli, Jeff Goldy, Emma Garren, Michael N Economo, Sarada Viswanathan, et al. Shared and distinct transcriptomic cell types across neocortical areas. *Nature*, 563(7729):72–78, 2018.
- [11] Ludvig Larsson, Jonas Frisén, and Joakim Lundeberg. Spatially resolved transcriptomics adds a new dimension to genomics. *Nature methods*, 18(1):15–18, 2021.
- [12] Robert R Stickels, Evan Murray, Pawan Kumar, Jilong Li, Jamie L Marshall, Daniela J Di Bella, Paola Arlotta, Evan Z Macosko, and Fei Chen. Highly sensitive spatial transcriptomics at near-cellular resolution with slide-seqv2. *Nature biotechnology*, 39(3):313–319, 2021.
- [13] Ao Chen, Sha Liao, Mengnan Cheng, Kailong Ma, Liang Wu, Yiwei Lai, Xiaojie Qiu, Jin Yang, Jiangshan Xu, Shijie Hao, et al. Spatiotemporal transcriptomic atlas of mouse organogenesis using dna nanoball-patterned arrays. *Cell*, 185(10):1777–1792, 2022.
- [14] Lambda Moses and Lior Pachter. Museum of spatial transcriptomics. *Nature Methods*, 19(5):534–546, 2022.
- [15] Ruben Dries, Qian Zhu, Rui Dong, Chee-Huat Linus Eng, Huipeng Li, Kan Liu, Yuntian Fu, Tianxiao

- Zhao, Arpan Sarkar, Feng Bao, et al. Giotto: a toolbox for integrative analysis and visualization of spatial expression data. *Genome biology*, 22(1):1–31, 2021.
- [16] Sheel Shah, Yodai Takei, Wen Zhou, Eric Lubeck, Jina Yun, Chee-Huat Linus Eng, Noushin Koulana, Christopher Cronin, Christoph Karp, Eric J Liaw, et al. Dynamics and spatial genomics of the nascent transcriptome by intron seqfish. *Cell*, 174(2):363–376, 2018.
- [17] Jeffrey R Moffitt, Dhananjay Bambah-Mukku, Stephen W Eichhorn, Eric Vaughn, Karthik Shekhar, Julio D Perez, Nimrod D Rubinstein, Junjie Hao, Aviv Regev, Catherine Dulac, et al. Molecular, spatial, and functional single-cell profiling of the hypothalamic preoptic region. *Science*, 362(6416):eaau5324, 2018.
- [18] Haoyang Li, Juexiao Zhou, Zhongxiao Li, Siyuan Chen, Xingyu Liao, Bin Zhang, Ruochi Zhang, Yu Wang, Shiwei Sun, and Xin Gao. A comprehensive benchmarking with practical guidelines for cellular deconvolution of spatial transcriptomics. *Nature Communications*, 14(1):1548, 2023.
- [19] Milad R Vahid, Erin L Brown, Chloé B Steen, Wubing Zhang, Hyun Soo Jeon, Minji Kang, Andrew J Gentles, and Aaron M Newman. High-resolution alignment of single-cell and spatial transcriptomes with cytospace. *Nature Biotechnology*, pages 1–6, 2023.
- [20] Dylan M Cable, Evan Murray, Luli S Zou, Aleksandrina Goeva, Evan Z Macosko, Fei Chen, and Rafael A Irizarry. Robust decomposition of cell type mixtures in spatial transcriptomics. *Nature Biotechnology*, 40(4):517–526, 2022.
- [21] Rui Dong and Guo-Cheng Yuan. Spatialdws: accurate deconvolution of spatial transcriptomic data. *Genome biology*, 22(1):1–10, 2021.
- [22] Vitalii Kleshchevnikov, Artem Shmatko, Emma Dann, Alexander Aivazidis, Hamish W King, Tong Li, Rasa Elmentaite, Artem Lomakin, Veronika Kedlian, Adam Gayoso, et al. Cell2location maps fine-grained cell types in spatial transcriptomics. *Nature biotechnology*, 40(5):661–671, 2022.
- [23] Ying Ma and Xiang Zhou. Spatially informed cell-type deconvolution for spatial transcriptomics. *Nature Biotechnology*, pages 1–11, 2022.
- [24] Zixuan Cang and Qing Nie. Inferring spatial and signaling relationships between cells from single cell transcriptomic data. *Nature communications*, 11(1):2084, 2020.
- [25] Noa Moriel, Enes Senel, Nir Friedman, Nikolaus Rajewsky, Nikos Karaïskos, and Mor Nitzan. Novosparc: flexible spatial reconstruction of single-cell gene expression with optimal transport. *Nature protocols*, 16(9):4177–4200, 2021.
- [26] Romain Lopez, Baoguo Li, Hadas Keren-Shaul, Pierre Boyeau, Merav Kedmi, David Pilzer, Adam Jelinski, Ido Yofe, Eyal David, Allon Wagner, et al. Destvi identifies continuums of cell types in spatial transcriptomics data. *Nature biotechnology*, 40(9):1360–1369, 2022.
- [27] Dongqing Sun, Zhaoyang Liu, Taiwen Li, Qiu Wu, and Chenfei Wang. Stride: accurately decomposing and integrating spatial transcriptomics using single-cell rna sequencing. *Nucleic Acids Research*, 50(7):e42–e42, 2022.
- [28] Marc Elosua-Bayes, Paula Nieto, Elisabetta Mereu, Ivo Gut, and Holger Heyn. Spotlight: seeded nmf regression to deconvolute spatial transcriptomics spots with single-cell transcriptomes. *Nucleic acids research*, 49(9):e50–e50, 2021.
- [29] Qianqian Song and Jing Su. Dstg: deconvoluting spatial transcriptomics data through graph-based artificial intelligence. *Briefings in Bioinformatics*, 22(5):bbaa414, 2021.
- [30] Romain Lopez, Achille Nazaret, Maxime Langevin, Jules Samaran, Jeffrey Regier, Michael I Jordan, and Nir Yosef. A joint model of unpaired data from scrna-seq and spatial transcriptomics for imputing missing gene expression measurements. *arXiv preprint arXiv:1905.02269*, 2019.

- [31] Tamim Abdelaal, Soufiane Mourragui, Ahmed Mahfouz, and Marcel JT Reinders. Spage: spatial gene enhancement using scrna-seq. *Nucleic acids research*, 48(18):e107–e107, 2020.
- [32] Chen Shengquan, Zhang Boheng, Chen Xiaoyang, Zhang Xuegong, and Jiang Rui. stplus: a reference-based method for the accurate enhancement of spatial transcriptomics. *Bioinformatics*, 37(Supplement_1):i299–i307, 2021.
- [33] Tim Stuart, Andrew Butler, Paul Hoffman, Christoph Hafemeister, Efthymia Papalexi, William M Mauck, Yuhan Hao, Marlon Stoeckius, Peter Smibert, and Rahul Satija. Comprehensive integration of single-cell data. *Cell*, 177(7):1888–1902, 2019.
- [34] Yang Song and Stefano Ermon. Improved techniques for training score-based generative models. *Advances in neural information processing systems*, 33:12438–12448, 2020.
- [35] Viktor Petukhov, Rosalind J Xu, Ruslan A Soldatov, Paolo Cadinu, Konstantin Khodosevich, Jeffrey R Moffitt, and Peter V Kharchenko. Cell segmentation in imaging-based spatial transcriptomics. *Nature biotechnology*, 40(3):345–354, 2022.
- [36] Dylan Bannon, Erick Moen, Morgan Schwartz, Enrico Borba, Takamasa Kudo, Noah Greenwald, Vibha Vijayakumar, Brian Chang, Edward Pao, Erik Osterman, et al. Deepcell kiosk: scaling deep learning-enabled cellular image analysis with kubernetes. *Nature methods*, 18(1):43–45, 2021.
- [37] Marius Pachitariu and Carsen Stringer. Cellpose 2.0: how to train your own model. *Nature Methods*, pages 1–8, 2022.
- [38] Uwe Schmidt, Martin Weigert, Coleman Broaddus, and Gene Myers. Cell detection with star-convex polygons. In *International Conference on Medical Image Computing and Computer-Assisted Intervention*, pages 265–273. Springer, 2018.
- [39] Carsen Stringer, Tim Wang, Michalis Michaelos, and Marius Pachitariu. Cellpose: a generalist algorithm for cellular segmentation. *Nature methods*, 18(1):100–106, 2021.
- [40] Simon Graham, Quoc Dang Vu, Shan E Ahmed Raza, Ayesha Azam, Yee Wah Tsang, Jin Tae Kwak, and Nasir Rajpoot. Hover-net: Simultaneous segmentation and classification of nuclei in multi-tissue histology images. *Medical Image Analysis*, 58:101563, 2019.
- [41] Neeraj Kumar, Ruchika Verma, Sanuj Sharma, Surabhi Bhargava, Abhishek Vahadane, and Amit Sethi. A dataset and a technique for generalized nuclear segmentation for computational pathology. *IEEE transactions on medical imaging*, 36(7):1550–1560, 2017.
- [42] Genomics, 10x. 10x Genomics Visium. Mouse Brain Serial Section 1 (Sagittal-Anterior), Mouse Brain Serial Section 2 (Sagittal-Anterior). <https://www.10xgenomics.com/resources/datasets/mouse-brain-serial-section-1-sagittal-anterior-1-standard-1-0-0>, <https://www.10xgenomics.com/resources/datasets/mouse-brain-serial-section-1-sagittal-anterior-1-standard-1-0-0>. Accessed: 2022-02-25.
- [43] Olaf Ronneberger, Philipp Fischer, and Thomas Brox. U-net: Convolutional networks for biomedical image segmentation. In *International Conference on Medical image computing and computer-assisted intervention*, pages 234–241. Springer, 2015.
- [44] Nanxin Chen, Yu Zhang, Heiga Zen, Ron J Weiss, Mohammad Norouzi, and William Chan. Wavegrad: Estimating gradients for waveform generation. *arXiv preprint arXiv:2009.00713*, 2020.
- [45] Yang Song and Stefano Ermon. Generative modeling by estimating gradients of the data distribution. *Advances in Neural Information Processing Systems*, 32, 2019.
- [46] Max Welling and Yee W Teh. Bayesian learning via stochastic gradient langevin dynamics. In *Proceedings of the 28th international conference on machine learning (ICML-11)*, pages 681–688. Citeseer, 2011.
- [47] Radoje Drmanac, Andrew B Sparks, Matthew J Callow, Aaron L Halpern, Norman L Burns, Bahram G

- Kermani, Paolo Carnevali, Igor Nazarenko, Geoffrey B Nilsen, George Yeung, et al. Human genome sequencing using unchained base reads on self-assembling dna nanoarrays. *Science*, 327(5961):78–81, 2010.
- [48] Xiaoyu Wei, Sulei Fu, Hanbo Li, Yang Liu, Shuai Wang, Weimin Feng, Yunzhi Yang, Xiawei Liu, Yan-Yun Zeng, Mengnan Cheng, et al. Single-cell stereo-seq reveals induced progenitor cells involved in axolotl brain regeneration. *Science*, 377(6610):eabp9444, 2022.
- [49] Panagiotis Karras, Ignacio Bordeu, Joanna Pozniak, Ada Nowosad, Cecilia Pazzi, Nina Van Raemdonck, Ewout Landeloos, Yannick Van Herck, Dennis Pedri, Greet Bervoets, et al. A cellular hierarchy in melanoma uncouples growth and metastasis. *Nature*, 610(7930):190–198, 2022.
- [50] Zhihua Ou, Shitong Lin, Jiaying Qiu, Wencheng Ding, Peidi Ren, Dongsheng Chen, Jiaxuan Wang, Yihan Tong, Di Wu, Ao Chen, et al. Single-nucleus rna sequencing and spatial transcriptomics reveal the immunological microenvironment of cervical squamous cell carcinoma. *Advanced Science*, 9(29):2203040, 2022.
- [51] Xiang Zhou, Kangning Dong, and Shihua Zhang. Integrating spatial transcriptomics data across different conditions, technologies, and developmental stages. *bioRxiv*, pages 2022–12, 2022.
- [52] Kangning Dong and Shihua Zhang. Deciphering spatial domains from spatially resolved transcriptomics with an adaptive graph attention auto-encoder. *Nature communications*, 13(1):1739, 2022.

Reviewer #1 (Remarks to the Author):

The authors have addressed my comments and suggestions. The additional analyses and results are interesting and add value to the manuscript.

Reviewer #2 (Remarks to the Author):

I want to thank you for your kind response. 1) Added hyperparameter comparison and benchmark testing. Tangram and CytoSPACE were two of the twelve cell deconvolution techniques the author investigated. 2) Improved the code's usability and repeatability. Although I believe this work generally satisfies the requirements for publishing as a paper, the following problems still need to be fixed.

Minor recommendations

1. There appears to be a missing image in Figure 1 that may be an image of H&E staining.
2. The publication contains several figures, many of which are not acknowledged in the text, such as Fig. S1, etc. It is requested that the author double-check.

Reviewer #3 (Remarks to the Author):

I really appreciate for the great efforts the authors made to revise the paper. However, there are still several points that are not convincing and need further investigation.

1, For my Question 3,"3. In Fig. S3, SpatialScope can not beat RCTD method". With the comprehensive comparison in the revised Fig. 2, it looks like that in Figs. 2c and 2d, SpatialScope is just comparable to RCTD in term of error rate and pcc, despite the improvement on gene expression decomposition. Given the complex math model over RCTD, significant improvement is expected in both tasks of cell type identification and gene expression decomposition. In addition, the latter task should be able to improve the former as well.

2, For Question 7, "The assumption that at most two cell types co-exist within a spot is too strong now.". The answer is confusing. If the spot size matches the size of single cell, why not assume that there is only one cell in spot, or that the spot contains parts of several cells?

From: Can Yang

Department of Mathematics
Hong Kong University of Science and Technology
Hong Kong

October 15, 2023

To: Reviewers,

RE: The revision of “SpatialScope: A unified approach for integrating spatial and single-cell transcriptomics data using deep generative models” (NCOMMS-23-12222)

Dear Reviewers,

We would like to thank you for the very detailed and constructive comments, which have greatly helped us to improve our manuscript.

We have revised the paper to address all the editorial comments. Changes in the revised manuscript were highlighted by using the **blue** text. Supporting materials were updated accordingly. For reproducibility, we have provided relevant codes and results for the experiments in the SpatialScope website: <https://github.com/YangLabHKUST/SpatialScope>. We look forward to receiving further feedback on our revisions.

Sincerely yours,

Can Yang and co-authors

Responses to Reviewer 1’s comments:

1. *The authors have addressed my comments and suggestions. The additional analyses and results are interesting and add value to the manuscript.*

Response:

We really appreciate the reviewer’s encouragement and compliment.

Responses to Reviewer 2’s comments:

1. *There appears to be a missing image in Figure 1 that may be an image of H&E staining.*

Response:

Thank the reviewer for this comment. We apologize for this oversight of the missing subfigure in Figure 1. In the revised manuscript, we have added the H&E staining image in Figure 1.

2. *The publication contains several figures, many of which are not acknowledged in the text, such as Fig. S1, etc. It is requested that the author double-check.*

Response:

We appreciate the reviewer for the careful inspection. We apologize for not acknowledging some supplementary figures in the main text. This oversight occurred because these figures are more closely associated with the supplementary methods section. In Supplementary Note 2.9.10, line 7 of this section, we referenced Fig. S1 to provide a clearer illustration of the differences between paired and unpaired single-cell reference data. Some other supplementary figures, such as Figures S37 and S38, which illustrate the schematics of the UNet model, are referenced in Supplementary Note 2.5 to elucidate the network architectures employed for modeling the gene expression distribution of single-cell reference data. However, we understand the reviewer’s recommendation and have made efforts to acknowledge all supplementary figures to the fullest extent feasible. For instance, in the revised manuscript, we have acknowledged Figure S1 in the subsequent sentence: “We generated simulation datasets by ‘gridding’ and aggregating cells on uniform grids to create simulated spots (Fig. 2a, Fig. S1).” We have also double-checked and ensured that all relevant supplementary materials are correctly acknowledged in the main text or supplementary note.

Responses to Reviewer 3’s comments:

1. *For my Question 3, “3. In Fig. S3, SpatialScope can not beat RCTD method”. With the comprehensive comparison in the revised Fig. 2, it looks like that in Figs. 2c and 2d, SpatialScope is just comparable to RCTD in term of error rate and pcc, despite the improvement on gene expression decomposition. Given the complex math model over RCTD, significant improvement is expected in both tasks of cell type identification and gene expression decomposition. In addition, the latter task should be able to improve the former as well.*

Response:

We appreciate the reviewer for raising these important questions, and we would like to provide further clarification regarding the sequential relationship of the three steps in our SpatialScope

framework. In SpatialScope, Step 1 involves nucleus segmentation, and Step 2 focuses on cell type identification. These two steps serve as auxiliary processes that contribute to the overall goal of Step 3, which is gene expression decomposition. It is important to note that these steps are performed sequentially, and based on the current SpatialScope model, the results from Step 3 cannot directly improve the results obtained from Step 2. However, we acknowledge the reviewer’s suggestion that combining steps 2 and 3 to share information could potentially enhance the performance of cell type identification. We agree that this extension could be a valuable improvement, but it falls outside the scope of the current SpatialScope framework.

Single-slice dataset

Multiple-slice dataset

Figures R1 (previous page) Performance comparison of SpatialScope and RCTD on simulated spatial transcriptomic data with different spot size. **a**, Schematic diagram for generating simulated ST data with large grid size (left) and small grid size (right) for single-slice data. **b**, Bar plot of error rate (left) across various grid scales from methods SpatialScope and RCTD on cell type identification task for single-slice data. Data are presented as mean values $\pm 95\%$ confidence intervals; $n = 10$ is the number of experiment replicates. Bar plot of PCC (middle) across various grid scales from methods SpatialScope and RCTD on cell type identification task for single-slice data. Data are presented as mean values $\pm 95\%$ confidence intervals; n is the number of spots. Bar plot of cosine similarity across various grid scales from methods SpatialScope and RCTD on gene expression decomposition task for single-slice data (right). Data are presented as mean values $\pm 95\%$ confidence intervals; n is the number of cells. **c**, Schematic diagram for generating simulated ST data with large grid size (upper) and small grid size (lower) for multiple-slice data. **d**, Bar plot of error rate (left) across various grid scales from methods SpatialScope and RCTD on cell type identification task for single-slice data. Data are presented as mean values $\pm 95\%$ confidence intervals; $n = 10$ is the number of experiment replicates. Bar plot of PCC (middle) across various grid scales from methods SpatialScope and RCTD on cell type identification task for multiple-slice data. Data are presented as mean values $\pm 95\%$ confidence intervals; n is the number of spots. Bar plot of cosine similarity across various grid scales from methods SpatialScope and RCTD on gene expression decomposition task for single-slice data (right). Data are presented as mean values $\pm 95\%$ confidence intervals; n is the number of cells.

To further enhance the comparison between SpatialScope and RCTD, we conducted additional experiments in various scenarios and settings to evaluate and compare the performance of SpatialScope with RCTD, at both single-cell levels and spot levels. We selected two datasets (one single-slice dataset and one multiple-slice dataset) and conducted a comparative analysis of our method and RCTD’s performance on these datasets. The first dataset is a single-slice MERFISH dataset (Fig. R1a) and is obtained from the mouse frontal cortex and striatum regions provided by the Allen dataset [1]. After preprocessing, it included expression values from 12,133 single cells and 374 genes. The second dataset was derived from multiple slices of the mouse cortex and hippocampus regions, as provided by Zeng [2] (Fig. R1c). The aligned 3D ST data encompassed 19,231 cells and 2,766 genes.

We systematically varied the grid size to manipulate the number of cells within each spot (Fig. R1a,c). Performance was assessed by computing the error rate at the single-cell level and the Pearson correlation coefficient (PCC) at the spot level for cell type identification task and cosine similarity at single-cell level for gene expression decomposition task. The grid size varies from $50 \times 50 \mu\text{m}$ to $10 \times 10 \mu\text{m}$. The largest size $50 \times 50 \mu\text{m}$, emulates the spot size of the Visium spatial transcriptomic technology, allowing for the presence of one to dozens of cells within a spot. The smallest size $10 \times 10 \mu\text{m}$, emulates the spot size of the Slide-seq spatial transcriptomic technology, allowing for the presence of one to two cells within a spot. It is worth noting that the error rate of the cell type identification in Step 2 and the accuracy of gene expression decomposition in Step 3 are the key focuses of our method. These metrics reflect how the model performs at the single-cell resolution, which is the primary goal of our method - to overcome the limitations and deficiencies of

existing spatial transcriptomics technologies in terms of resolution.

Based on the error rate metric, our method exhibits a distinct superiority over RCTD across almost all grid sizes (Fig. R1b,d left), particularly under large grid sizes, as it effectively exploits the advantages of spatial information. This advantage is further amplified in the multiple-slice dataset due to borrowing information from the adjacent slice (Fig. R1d left). For the inference of single-cell gene expression, SpatialScope exhibits a substantial advantage over RCTD (Fig. R1b,d right), with the magnitude of this advantage becoming more pronounced as the grid size decreases. This evidence demonstrates the major improvement of SpatialScope over RCTD at the single-cell resolution. Regarding cell type deconvolution accuracy at the spot level, SpatialScope achieves comparable performance to RCTD based on the PCC metric in the single-slice dataset (Fig. R1b middle). However, SpatialScope can outperform RCTD in the multiple-slice dataset (Fig. R1d middle) because spatial smoothness offers more advantages in the presence of multiple slices with higher cell density in the data.

In summary, by exploring different scenarios, we can obtain a more comprehensive understanding of the performance and behavior of the SpatialScope and RCTD methods across various settings. In different scenarios, SpatialScope consistently demonstrates superior or, at the very least, comparable performance to RCTD.

2. For Question 7, "The assumption that at most two cell types co-exist within a spot is too strong now.". The answer is confusing. If the spot size matches the size of single cell, why not assume that there is only one cell in spot, or that the spot contains parts of several cells?

Response:

Thank the reviewer for this comment. We sincerely apologize for the previous confused response. We would like to take this opportunity to make clarification based on the Slide-seq data analysis. If the spot size matches the size of a single cell, it is indeed plausible to assume that there is only one cell in a spot (denoted as the "one cell" mode). However, as mentioned by the reviewer, one spot may contain fractions of several cells due to the technique limitation. To demonstrate this, we estimated the number of cell types per Slide-seq V2 cerebellum spot using the cell type deconvolution results. Specifically, we determined the number of cell types within each spot as those with an estimated proportion exceeding 0.2. For example, if a spot exhibited two distinct cell types with proportions larger than 0.2, then the predicted number of cell types for this spot is two. Fig. R2 shows the predicted number of cell types for Slide-seq V2 cerebellum spots. In total, 22.0% and 1.6% of spots were predicted to contain two and three cell types, respectively, consistent with previous estimates [3]. Simply assuming that there is only one cell in these spots may not be appropriate in this case. Consequently, to enhance the flexibility of our model and mitigate the risk of overfitting with regard to cell number estimation, we assume that at most two cell types co-exist within a spot for Slide-seq data (denoted as "two cell" mode).

Next, we show the benefits of this assumption in real data analysis. As shown in Fig. R3a and Fig. 5 of the main text, SpatialScope correctly assigned cell type labels and captured the layered architecture (Oligodendrocytes layer, Granular layer and Purkinje-Bergman layer) of the cerebellum [4; 5]. Notably, Bergmann and Purkinje cells spatially colocalize to the same layer, resulting

in a population of spots exhibiting marker gene expression signatures from both cell types (Fig. R3c). This observation strongly suggests that these spots contain fractional representations of both Purkinje and Bergman cells. If we simply assume that there is only one cell in a spot, then these doublet spots will be incorrectly assigned with one cell type only (Fig. R3c, left panel). In contrast, the flexible “two cell” mode adopted by our method can automatically distinguish doublets from singlets (Fig. R3c, right panel). Consequently, we are able to accurately assign the mixed cell types for these doublet spots, thus yielding a more elucidated and comprehensive depiction of tissue structures. Analogously, for spots in the adjacent region of Oligodendrocytes and Granule layer, we observed the consistent advantage of “two cell” mode over “one cell” mode (Fig. R3b).

Figure R2: Number of cell types per Slide-seq V2 cerebellum spot. In total, 22.0% and 1.6% of spots were predicted to contain two and three cell types, respectively.

Figure R3: Benefits of at most two cell co-exist assumption for Slide-seq dataset.
a, Cell type identification results by SpatialScope correctly captured the layered architecture (Oligodendrocytes layer, Granular layer and Purkinje-Bergman layer) of cerebellum. **b**, Expression of Granule and Oligodendrocytes cell marker genes for spots with Granule, Oligodendrocytes or doublet (mixture of Granule and Oligodendrocytes) cell type assignment when using one-cell (left panel) or two-cell (right panel) modes. **c**, Expression of Bergmann and Purkinje cell marker genes for spots with Bergmann, Purkinje or doublet (mixture of Bergmann and Purkinje) cell type assignment when using one-cell (left panel) or two-cell (right panel) modes.

References

- [1] William E Allen, Timothy R Blosser, Zuri A Sullivan, Catherine Dulac, and Xiaowei Zhuang. Molecular and spatial signatures of mouse brain aging at single-cell resolution. Cell, 186(1):194–208, 2023.
- [2] Hu Zeng, Jiahao Huang, Haowen Zhou, William J Meilandt, Borislav Dejanovic, Yiming Zhou, Christopher J Bohlen, Seung-Hye Lee, Jingyi Ren, Albert Liu, et al. Integrative in situ mapping of single-cell transcriptional states and tissue histopathology in a mouse model of alzheimer’s disease. Nature Neuroscience, 26(3):430–446, 2023.
- [3] Dylan M Cable, Evan Murray, Luli S Zou, Aleksandrina Goeva, Evan Z Macosko, Fei Chen, and Rafael A Irizarry. Robust decomposition of cell type mixtures in spatial transcriptomics. Nature Biotechnology, 40(4):517–526, 2022.
- [5] Francesca Prestori, Lisa Mapelli, and Egidio D’Angelo. Diverse neuron properties and complex network dynamics in the cerebellar cortical inhibitory circuit. Frontiers in Molecular Neuroscience, 12:267, 2019.
- [4] Amanda M Brown, Marife Arancillo, Tao Lin, Daniel R Catt, Joy Zhou, Elizabeth P Lackey, Trace L Stay, Zhongyuan Zuo, Joshua J White, and Roy V Sillitoe. Molecular layer interneurons shape the spike activity of cerebellar purkinje cells. Scientific reports, 9(1):1–19, 2019.